# Distinct phases of eustatic and tectonics forcing for late Quaternary landscape evolution in southwest Crete, Greece

Mouslopoulou, Vasiliki[1], Begg, John[2], Fülling, Alexander[3], Moraetis, Daniel[4], Partsinevelos, Panagiotis[5], Oncken, Onno[1]

[1] GeoForschungsZentrum, Telegrafenberg, D-14473 Potsdam, Germany

[2] GNS Science, PO Box 30368, Lower Hutt, New Zealand

[3] Humboldt University of Berlin, 12489 Berlin, Germany

[4] Sultan Qaboos University, PO Box 36, PC 123, Muscat, Oman

[5] Technical University of Crete, 73100 Chania, Greece

*Correspondence to*: Vasiliki Mouslopoulou (vasso@gfz-potsdam.de)

**Abstract.** The extent to which climate, eustacy and tectonics interact to shape the late Quaternary landscape is poorly known. Alluvial fans often provide useful indexes that allow decoding the information recorded on complex coastal landscapes, such as those of Eastern Mediterranean. In this paper we analyse and date (using luminescence IRSL dating) a double alluvial-fan system on southwest Crete, an island straddling the forearc of the Hellenic subduction margin, in order to constrain the timing and magnitude of its vertical deformation and discuss the contributing factors to its landscape evolution. The studied alluvial system is exceptional because each of its two juxtaposed fans is recording individual phases of alluvial and marine incision, providing, thus, unprecedented resolution in the formation and evolution of its landscape. Specifically, our analysis shows that the fan sequence at Domata developed during Marine Isotope Stage (MIS) 3 due to five distinct stages of marine transgressions and regressions and associated river incision, as a response to sea-level fluctuations and tectonic uplift at averaged rates of ~2.2 mm/yr. Interestingly, comparison of our results with published tectonic uplift rates from western Crete, shows that uplift during 20-50 kyr BP was minimal (or even negative). Thus, most of the uplift recorded at Domata must have been accrued in the last 20 kyr. This implies that eustacy and tectonism impacted on the landscape at Domata over mainly distinct time-intervals (e.g. sequentially and not synchronously), forming and preserving the coastal landforms, respectively.

# 1 Introduction

Sea-level fluctuations relative to modern sea-level are well constrained for the last 0.5 Ma (e.g., Imbrie et al., 1984; Martinson et al., 1987; Bassinot et al., 1994; Chappell et al., 1996; Dickinson, 2001; Siddall et al., 2003; Rabineau et al., 2005; Lambeck and Purcell, 2005; Lisiecki and Raymo, 2005; Antonioli et al., 2007). When these fluctuations are used in conjunction with dating techniques, they provide a powerful tool for interpreting coastal geomorphology and assessing vertical deformation from marine and marginal marine deposits through the middle and late Quaternary (e.g., Pirazzoli et al., 1996; Rabineau et al., 2005; Antonioli et al., 2007; Mouslopoulou et al., 2015a). While there is little debate about the role of tectonic uplift in generating the topographic relief required for the processes of erosion and deposition, uncertainty still exists as to the relative significance of tectonic, eustatic and climatic contributions to deposition and incision of fans of Quaternary age and their variation through time (e.g., Waters et al., 2010).

Alluvial fans are excellent proxies for Quaternary landscape evolution in a climate such as the Mediterranean and their study could potentially place some constraints on the factors that impacted the landscape during its formation and evolution (e.g., Pope et al., 2008, 2016; Zacharias et al., 2009). Overall, alluvial-fan deposition is influenced by rising or relatively high sea-level, by catchment size and sediment supply, by major changes in climatic conditions (such as high rainfall and/or short, intense storms), and by vegetation coverage in the catchment area. These factors regulate stream carrying capacity and sediment supply (e.g. Bull, 1979; Pope et al., 2016) and are responsible for whether deposition or river entrenchment processes predominate at any one time. Alluvial-fan surface abandonment and river entrenchment is favoured by eustatic sea-level fall or tectonic uplift (or a combination of both), reduction in sediment supply due to climatic amelioration, densification of catchment vegetation or reduced rainfall (e.g., Pope et al., 2008, 2016; Waters et al., 2010). Fan aggradation is encouraged by factors such as rising base level (sea-level for Domata), sediment supply, increased stream carrying capacity (rainfall/temporal rainfall distribution) and reduction in catchment vegetation cover.

For example, in southwest Crete (eastern Mediterranean), Nemec and Postma (1993) and Pope et al. (2008, 2016) studied a fan system and showed that fan deposition was associated with all last glacial stadial and interstadial conditions, and that fan entrenchment was governed by the major climatic transitions between MIS5/4 and MIS2/1. On the other hand, Tiberti et al. (2014) and Mouslopoulou et al. (2015b) have shown that tectonic uplift on western Crete is significant, reaching, during late-Quaternary, rates of 7-8 mm/yr. Despite this progress in understanding, we are, however, still unable to precisely appreciate the interplay between, and the relative importance of, climate and tectonics during late Quaternary in the Mediterranean. How do severe uplift rates recorded along the forearc of the Hellenic margin (where Crete lies) reconcile with periods of prevalence of eustatic processes? Here we capitalise on a well-preserved alluvial-fan system at Domata in southern Crete (Fig. 1), to study the late Quaternary (~50 kyr BP) interplay between sea-level fluctuations and tectonics. To our knowledge, the site at Domata is unique on the island of Crete as each alluvial fan-building episode has been followed by distinct

episodes of alluvial incision and subsequent marine trimming (Figs. 2 and 3). Thus, at this site, we have the opportunity to test the idea proposed by Pope et al. (2016) that eustatism controls largely the landscape evolution in southwest Crete.

Using luminescence dating together with the Siddall et al (2003) sea-level curve, we find that the alluvial-fan system at Domata was consecutively affected by: 1) Sea-level fluctuations, triggering building of the fans and subsequent river and marine incision between ~45 and 20 kyr BP, during a period of minimal tectonic activity and 2) Intense tectonic uplift between ~20 kyr and present, at rates that exceeded that of the rising sea-level, resulting in the preservation of the entire fan sequence. These findings are in accord with Pope et al. (2016) in showing that regional tectonics did not necessarily play a key role in fan incision in southern Crete.

## 2 Geological setting of Crete and vertical tectonics

The Mediterranean island of Crete is a mountainous and elongate landmass (~260 km long from west to east, 60 km wide from north to south) that lies within the uplifted forearc section of the Hellenic subduction margin, the most active seismically region in Europe (Fig. 1). The total relative convergence rate between the subducting African Plate and the overriding Eurasian plate is ~35-40 mm/yr (Reilinger et al., 2010). The subduction trench lies ~225 km to the south of Crete (e.g. Ryan et al., 1970; Le Pichon and Angelier, 1979) while the north-dipping subduction interface lies at a depth of ~40 to 65 km beneath Crete (Papazachos et al., 2006; Vernant et al., 2014), with the projection of the down-dip end of the locked zone aligning with the southern coastline of Crete, where the study area is located (Fig. 1) (Meier et al., 2007). The Hellenic Trough, a major bathymetric and tectonic feature within the forearc, lies south of Crete and includes three secondary features which, from west to east, are named as the Ptolemy, Pliny and Strabo troughs (Fig. 1).

Crete has been characterised by a complex history of vertical movements during Cenozoic (e.g., Peters et al., 1985). Onshore sediments record a period of subsidence and basin development through the Middle and Late Miocene (Serravalian to Messinian) with a change to rapid uplift in the Early Pliocene (Zanclean), followed by slower long-term uplift that continues to the present day (e.g., Le Pichon and Angelier, 1981; Angelier et al., 1982; Meulenkamp et al., 1994; Zachariasse et al., 2008; Roberts et al., 2013; Gallen et al., 2014).

Late Quaternary tectonic uplift on Crete is uniform but transient (Tiberti et al., 2014; Mouslopoulou et al., 2015b). Using dated paleoshorelines and numerical models, Tiberti et al. (2014) and Mouslopoulou et al. (2015b) show that the island of Crete experienced, during the last 20 thousand years, periods of severe fluctuations in its vertical deformation (at rates of up to 8 mm/yr) while in the preceding ~30 thousand years, vertical movement on Crete was either minimal or reversed (subsidence of 0-3 mm/yr). High uplift rates (~7-8 mm/yr) are also documented on western Crete since 2 kyr BP, in response to co-seismic uplift (that locally reached up to 10 m) (Pirazzoli et al., 1996; Shaw et al., 2008; Mouslopoulou et al., 2015a).

Uplift rate transients on Crete are thought to result from non-uniform stress accumulation and release on upper-plate reverse faults in the overriding plate (Shaw et al., 2008; Stiros, 2010; Tiberti et al., 2014; Mouslopoulou et al., 2015b).

Historical and archaeological records have been also used to link uplift in western Crete to earthquakes (Pirazzoli et al., 1982, 1996; Stiros, 2001; Papazachos and Papazachou, 2003; Papadimitriou and Karakostas, 2008; Shaw et al., 2008; Stefanakis, 2010; Strasser et al., 2011). In particular, historic accounts of a major earthquake in Crete in ~AD 365 are approximately coincident with historic documents recording tsunami inundation of parts of the Libyan and Egyptian coastlines, particularly Alexandria (Ammianus Marcellinus, translated by C.D. Yonge, 1862; Polonia et al., 2013). A gently tilted paleoshoreline (tidal notch) can be followed along the western shoreline of Crete for ~150 km from the area of maximum uplift near the southwest tip (Elafonisi) to as far east as Agios Georgios (Fig. 1). At Domata, our study site, this notch is at 6 m above sea-level. A number of studies attribute the timing of this prominent paleoshoreline to the AD 365 historic earthquake (e.g., Pirazzoli et al., 1982; 1996; Stiros, 2001; Shaw et al., 2008).

## 3 Data – Methods - Chronology

At Domata a unique sequence of two juxtaposed generations of alluvial fans are documented, each truncated by different episodes of river and marine incision (see Figs. 2, 3 and 4). The discussion that follows gives a detailed account on the materials and the geometry of the alluvial fan system at Domata, establishes the stratigraphic relationships of its key geomorphic features and provides the chronological framework (sequence of events) within which the established stratigraphic relationships developed.

### 3.1 Coastal geomorphic features at Domata

The landmass of the White Mountains (Lefka Ori) dominates the landscape of western Crete (Figs. 1 and 2). At the southern coastline of Crete, and proximal to our study area, the White Mountains drop abruptly by >1800 m to sea-level over a distance of <10 km, forming a steep and rugged landscape, often incised by narrow south-draining gorges (Fig. 2). One such gorge is Klados which reaches the sea at the beach of Domata (Figs. 2 and 3). This steep subaerial landscape extends offshore along most of the southwest coast of Crete, as evidenced by the regional bathymetric slopes which are steeper offshore than onshore (Le Pichon and Angelier, 1979; Mascle et al., 1986). As bedrock crops out along much of the southwestern coastline of Crete, it is clear that bathymetric slopes are also cut in bedrock, implying that the Quaternary sediment sequence recorded at Domata has no significant thickness offshore.

The rivers within the gorges of western Crete are usually ephemeral and scour to bedrock, depositing gravels only locally, commonly where valleys widen at junctions with side valleys, across faults (e.g., Sfakia fans; Pope et al., 2008), or close to the coast. As the rivers approach the sea, gradients shallow and stream carrying capacity decreases resulting in deposition

from bedload of fans grading to the shoreline. The headwaters of the Klados River, only 7 km from the coast, reach an elevation of ~1800 m in the White Mountains, and no significant areas of sediment accumulation exist between its upper reaches and the fans near the coast that are the subject of this paper. In the Klados Gorge area, bedrock comprises mainly crystalline platy limestones with some cherts and platy marbles (Creutzburg and Siedel, 1975; Manutsoglu et al., 2003; Fassoulas et al., 2004). The erosion of these units supplied the Domata area mainly with carbonate clasts and limited chert clasts which explains the abundance of carbonates in the alluvial fans and fluvial terraces.

In order to better interpret the geomorphology at Domata we topographically surveyed and modelled the entire study region (Figs. 3 and 4). The data acquisition was performed with a double precision Real Time Kinematic (RTK) GPS receiver and was corrected to provide coordinates under the Greek Geodetic Reference system (GGRS 87). The topographic dataset includes a total of 4,156 survey points, measured under an excellent geometric dilution of precision (GDOP) and accuracy of <1 cm. Some areas were not surveyed due to dense vegetation; however, values for these regions were interpolated using the 'nearest neighbour method'. A series of breaklines and sparse elevation models from the National Cadastre and Mapping Agency of Greece were incorporated in the model (Fig. 4a) to optimise representation.

Older geomorphic features are present at Domata in the form of two marine benches cut in bedrock at elevations of about 100 and 360 m on the western slopes above the Klados River (Fig. 2). While we cannot assign ages to these benches, their altitude and geomorphic similarity with known and dated (MIS5) late Pleistocene marine benches elsewhere in Crete (e.g., Strasser et al., 2011; Gallen et al., 2014; Strobl et al., 2014), provides some stratigraphic and chronologic context for the age of the alluvial fans at Domata (i.e., because of their lower elevation, the alluvial fan surfaces that are subject of this paper are expected to be younger than 125 kyr).

At Domata, two triangular, elevated fan surfaces (a lower and an upper surface) covering a combined surface area of ~0.1 km$^2$ near the mouth of Klados River, rise to an inland elevation of ~100 m (Figs. 2, 3 and 4). The fan surfaces are derived from a single feeder channel, the Klados River that drains a relatively small catchment (immediately west of the larger and better known Samaria Gorge) and lie at the seaward end of a narrow entrenched gorge. They are unique in south Crete as they are protected from alluvial erosion by a low bedrock ridge (see Fig. 2) which channels the river flow to the western side of its narrow valley. Where the Klados River leaves its bedrock gorge, ~500 m from the coast, its channel is incised into gravels ~40 m below an abandoned fan surface, the lower of the two surfaces (Figs. 2 and 3). Gravel deposits beneath the surface on lap bedrock on both sides of the valley without structural deformation (e.g., faulting). Downstream, ~40 m from the river mouth, the seaward extent of the lower-fan surface is at c. 35 m above sea-level (Fig. 4). Here, both the fan surface and its alluvial entrenchment cliff are trimmed parallel with, and close to, the present shoreline (Figs. 2 and 3). The linearity and parallelism of this cliff to the modern coastline clearly implies that this cliff has been trimmed by the sea. The elevation of the lower-fan surface decreases eastwards along the sea-trimmed cliff to <10 m above sea-level near the east end of

Domata beach (Figs. 2 and 5a). Along this coastal cliff, the highest elevation of the lower-fan surface occurs ~90 m east of the Klados River (Figs. 2 and 5a).

The upper-fan surface lies at ~100 m elevation at its upstream extent, ~60 m above the active river bed and ~20 m above the lower-fan surface (Figs. 4 and 5a). The upper-fan deposits are truncated by an old river incision (trending ~200°) that is older than the lower-fan surface, as the deposits of the lower-fan lap against the buried upper-fan deposits (Fig. 3). Downstream, the seaward extent of the upper-fan surface and its entrenchment cliff are truncated by another marine cliff (trending ~130°) that pre-dates deposition of the lower-fan, as the lower-fan surface also laps against this (Figs. 2 and 3). Where the marine trimming truncates the upper-fan surface and its river entrenchment cliff, the upper-fan surface has an elevation of c. 60 m, and this decreases eastwards to ~30 m at the east-end of the beach (Figs. 2, 4 and 5a). In the east, the upper-fan surface is overlain by silty sand near the eastern end of Domata Beach (Fig. 5a). This deposit is preserved seaward of the only stream gully that crosses the upper fan surface, draining the bedrock area behind that fan; this ephemeral stream is undoubtedly the source of this younger silty sand (UF-2 sample in Table 1).

Lower-fan materials exposed in the sea-cliff are dominantly poorly sorted, matrix-supported gravels, moderately stratified, with coarser beds commonly < 2 m thick and sometimes clast-supported, and finer beds < 1 m thick that display lateral lensing and channelling (see stratigraphic log in Fig. 5b). Along the coastal cliff bedding is convex up, sub-parallel with the lower-fan surface (Fig. 6a). Some individual beds can be traced laterally up to 200-300 m (see thin dashed lines in Fig. 6a). In the coastal cliff, lower-fan materials lap onto a gently undulating, sub-horizontal discontinuity on an underlying older alluvial gravel (e.g. remnants of the upper-fan) that is coarser and more commonly clast-supported (Fig. 5b), and has a higher fine-grained content (Figs. 2, 3, 5b and 6a). The contact surface between the two fan units is very clear and extends along the length of the beach (Figs. 5 and 6a) and also up the Klados River for ~100 m (Fig. 3). In places, the contact is locally obscured by fallen debris, but it is clearly subhorizontal, with low relief and undoubtedly separates the two fan units (Fig. 6a). At each end of the beach, the older gravel materials lap onto older sedimentary rocks (gravel and sand) (Fig. 6c).

Another subtle geomorphic feature of interest is a bioerosion notch indicating an uplifted paleoshoreline at ~6 m above present sea-level at the west end of Domata beach (Fig. 6b and c). This notch continues west and east from Domata and has been mapped around the coastline of western Crete and attributed to a seismically uplifted paleoshoreline dated at ~365 AD (e.g., Pirazzoli et al. 1982, 1996; Shaw et al. 2008). Figure 6c illustrates the relationship between the unconformity at the base of the fan sequence and the AD365 bioerosional notch. Our interpretation of the stratigraphy is as follows: a dissected erosional surface on older sediments pre-dates the fan deposits, while the AD 365 bioerosional notch post-dates both the older surface and the fan deposits that rest on it. Here it is evident that the unconformity does not represent the same feature as the AD 365 bioerosional notch. Instead, Figure 6c clearly shows that the AD 365 bioerosional notch post-dates both the deeply dissected erosional surface (of older deposits) and the fan deposits that rest on it. In other words, the apparent

coincidence of the "wave-cut bench" and the AD 365 bioerosional notch that may appear locally at Domata, cannot mean that the entire Domata fan deposits are Holocene in age (at it was suggested in the discussion associated with this article). Further, the presence of a small terrace on the west side of the Klados River (Fig. 6b) at approximately the same elevation (6m) with the bioerosion notch represents an alluvial terrace stranded by that uplift. As with the lower-fan surface, this terrace has been trimmed by the sea. This late-stage uplift resulted in a readjustment of the Klados River bed and incision near the mouth of the stream (Fig. 6b).

## 3.2 Luminescence dating of alluvial fans

To place chronologic constrains on the series of geomorphic features described here from Domata, we collected in steel tubes five samples for OSL dating from depths ranging from 0.24 to 1.1 m below the ground surface (Table 1). One sample was collected from close to the surface of the upper-fan (UF-1) to constrain the end of the upper-fan aggradation (surface abandonment) and the initiation of incision (Figs. 3 and 5a). A further sample (RB-1) was collected from the upper-fan deposits exposed in the lower reaches of the Domata stream cliff to constrain the age of deposition of the early upper-fan deposits (Figs. 3 and 5a). Two samples collected from close to the lower-fan surface (LF-1a/b and LF-2a/b) were to provide constraints on the timing of lower-fan abandonment and initiation of incision (Figs. 3 and 5a). A further sample (UF-2) was collected from deposits (silty sand) mantling both the lower and upper-fan surfaces, near the east-end of the Domata beach, to test its age relative to UF-1 and UF-2. The results of the luminescence analysis are presented in Table 1 and Figure 7.

### 3.2.1 Sample preparation and measurements

All samples were dated in the luminescence lab at Humboldt University of Berlin (Germany), where they were prepared under subdued red light according to standard procedures. After separating the wanted grain size fractions by wet sieving (38-63 µm and 90-200 µm), carbonates and organic material were removed using 10% hydrochloric acid and 10% hydrogen peroxide. Quartz and potassium feldspar were extracted from the coarser grain fraction by density separation using heteropolytungstate heavy liquid (LST) of 2.75, 2.62 and 2.58 g/cm$^3$.The subsequent etching of the separated quartz with hydrofluoric acid (40%, 60 min) eliminated any potential feldspar contamination and removed the alpha irradiated outer grain layer. From the finer fraction of 38-63µm quartz was isolated by a two-week treatment with 38% hexafluorosilicic acid. After renewed sieving small multiple grain aliquots (2mm) were prepared of etched quartz (90-200 and 38-63µm) and potassium feldspar (90-200µm).

Quartz OSL (optical stimulated luminescence) measurements were performed on a Risø TL/OSL-DA 15 reader (blue LED stimulation at 470 nm and detection through a Hoya U340 filter with transmission centered on 330 nm) and on a Lexsyg luminescence measurement system (green LED stimulation at 525 nm and detection through a Schott BG3 Delta-BP365/50

EX-Interference filter combination at 380 nm). Feldspar IRSL (infrared stimulated luminescence) measurements were conducted on a Lexsyg luminescence measurement system (IR-LD stimulation at 850 nm and detection through a Schott BG39 AHF-BrightLine HC 414/46-Interference filter combination at 410 nm). Quartz paleodoses were measured using a SAR (single aliquot regenerative) protocol according Murray and Wintle (2000, 2003) with preheat temperature set to 240°C

(10s) and test dose cutheat to 160°C. Wallinga et al. (2000) introduced the SAR protocol to the IRSL dating of potassium feldspar. It was here modified following Blair et al. (2005) applying equal preheat procedures after every irradiation step (250°C, 60s). The appropriate preheat temperatures and durations were identified conducting dose recovery tests on samples RB-1 and LF-2a/b (SAR equivalent dose determinations of known lab doses with varying preheat temperatures). Quartz was stimulated at 125°C for 40 s, feldspar at 50°C for 300 s. The built-in beta sources (Sr-90) emitted 0.068 Gy/s (Lexsyg) and

0.093 Gy/s (Risø) respectively. The sediment dose rates were estimated by measuring the contents of uranium, thorium and potassium on a high resolution gamma spectrometer. The cosmic-ray dose rates were estimated from geographic position, elevation and burial depth (Prescott and Hutton, 1994). The internal potassium content of the measured feldspar was assumed to be 12.5 ± 0.5 % according to Huntley and Baril (1997).

### 3.2.2 Luminescence results

Quartz OSL dating results reported in Pope et al. (2008) and OSL and U series dating of Pope et al. (2016) proved the suitability of standard quartz SAR protocols for dating fan sediments along the nearby Sfakia piedmont in southern Crete (Fig. 1). In contrast, the investigated quartz from Domata showed poor luminescence properties: the OSL signals were dim, dose recovery tests yielded unsatisfactory results, the highly scattering paleodoses produced positively skewed broad distributions and the resulting quartz ages showed no relationship with stratigraphy (underestimation of true age). This led to

the conclusion that quartz is not the appropriate material for dating the alluvial fans at Domata. The most likely explanation for the non-suitability of the quartz (weak, or even missing, fast OSL signal component) is the dominance in our samples of fresh insensitive quartz, which had undergone few sedimentation cycles (Preusser et al. 2006; Steffen et al., 2009). Thus, potassium-feldspar (IRSL) was used instead to date the landforms at Domata.

The feldspar (IRSL) dating produced reliable age ranges (Fig. 7). Best results for dose recovery tests on laboratory-bleached feldspar samples from Domata were obtained without applying any sensitivity correction. Thus, a simplified SAR protocol without testdose measurement was used for the paleodose determination of the natural potassium-feldspar samples. No fading tests were made to correct for any potential age underestimation. Sensitivity changes were assessed by repeating the first irradiation step at the end of each SAR cycle assuming that the luminescence intensities should coincide (recycling ratio

close to 1.0). Here, a recycling ratio between 0.85 and 1.15 was tolerated. The a-value for assessing the alpha particle contribution to the paleodose was set to 1.5 ± 0.5 (Balescu and Lamothe, 1994). Basic statistical values are presented in Table 1. Under perfect conditions the arithmetic mean and the median should coincide. But here the mean value is always larger compared to the median, typical for positively skewed age distributions. This can indicate insufficient exposure of the

sediment to daylight during the last sedimentation cycle. But also post-depositional mixing, contamination with younger grains from the surface (low sampling depth, bioturbation) or microdosimetric inhomogeneities are possible reasons for skewed age distributions. Compared to the mean, the median is less sensitive to large outliers (RB-1, LF-1a/b). The Central Age Model (CAM) and the Minimum Age Model (MAM) according Galbraith et al. (1999) were used to further describe the

5 age distributions (Fig. 7 and Table 1). Minimum age models are recommended when dating mixed-age sediments yielding broad age distributions to better estimate the population of well bleached grains (Galbraith and Roberts, 2012).

Collectively, our potassium-feldspar measurements suggest that the ages of the landforms at Domata range between ~55 and 25 kyr BP (Table 1). All Mean values indicate a last glacial age for all samples and appear in stratigraphic order. Median

values are all in stratigraphic sequence (except for LF-2a/b) although they are significantly younger than the Mean and the CAM values. The CAM ages are consistent with the Mean and Median values in indicating a last glacial age (MIS3) for all samples. The CAM series data are all in stratigraphic order (except for the LF-2a/b sample), and the absolute values are younger than the means. The MAM ages "lean" towards the younger end of the timescale, which is to be expected, because they are the minimum possible ages, not the most likely ages.

Thus, the chronostratigraphy of the majority of the geomorphic landforms formed during MIS 3 (29-57 kyr). When used in conjunction with the sequence of events dictated by the stratigraphy presented in Section 3.1 and the Siddall et al. (2003) sea-level curve, the chronology of the geomorphic events that formed the landscape at Domata can be established reasonably well, despite the significant errors in the luminescence measurements (Fig. 7). The landscape evolution, including its chronology, is discussed in Section 4.

### 3.3 Soil development

We have performed a macroscopic soil-profile characterization for the soil-horizons that develop on the two fan surfaces (Fig. 8a). The fan surfaces are gently dipping, with a maximum gradient of ~7 to 8° and incision restricted to few dry and shallow (<1m) creek courses. Thus, erosion on these surfaces is expected to be minimal. Nevertheless, to minimise the effect

of soil erosion, we located our soil profiles away from creek incisions.

The soils at Domata are categorized as Leptosols (Soil Atlas of Europe, 2005; FAO, 2006) and comprise a shallow (<0.5 m) soil cover over coarse sediment of highly calcareous material (Fig. 8). Soils on both the upper (UF) and lower (LF) fans have common parent material (fan gravels), mainly consisting of limestone pebbles and cobbles, and the fans are covered by pine

trees (Pinus brutia) (Fig. 8a). However, the soils of the upper (UF) and lower (LF) fan surfaces differ macroscopically and in their physical, biochemical and geochemical parameters. Soil thickness, averaged from 6 soil profiles, varies from 0.1 m

(Fig. 8b) beneath the lower-fan surface to 0.4 m (Fig. 8c) beneath the upper-fan surface. The UF soil is yellowish-brown with A and (weak) B horizons and texture from subangular to granular while the LF soil is yellowish with granular texture and has no distinct horizons (apart from a very thin horizon A) (Fig. 8b and c).

Comparison of the macroscopic characteristics of the soils at Domata with the soils identified at the piedmonts at the nearby region of Sfakia (Fig. 1) by Pope et al. (2008), provides additional evidence that the alluvial fan system at Domata was formed during MIS 3 (as constrained by the IRSL dating). Specifically, at Domata we find soils the characteristics of which closely resemble the soils that developed at Sfakia during stage 2C (70-16 kyr BP), while there is total absence of older soils that developed during stage 2A (~144 kyr) (Pope et al., 2008). The former (stage 2C) is a brown to yellowish-brown soil

with limited B horizon and subangular texture, characteristics that match the soils at Domata, whereas the latter contains highly crystalline iron oxides with a clear B horizon.

The macroscopic observations (UF soil thicker and redder) also imply that the upper-fan soil is more mature compared to that of the lower-fan. Preliminary geochemical analysis (Moraetis et al., 2015) confirms that the soil in UF is more mature

(e.g., older), as it has lower *specific surface area* and higher content of well-shaped hematite (and less goethite) compared to that in the LF (Wang et al., 2013). This observation is also in agreement with soil analysis at the nearby region of Sfakia (Fig. 1), where Pope et al. (2008) showed that the soil redness and the content of crystalline iron oxide (hematite) increase with increasing alluvial fan age. The greater age of the UF soil compared to the LF soil is also independently demanded by our stratigraphic observations on crosscutting and incision of fans and the luminescence dating that shows that the upper-fan

surface developed at least 5 kyr earlier than the lower fan surface (see discussion in Section 4; Table 1). According to Lair et al. (2009) and Huang et al. (2016), the ~5 kyr of difference in the residence time between the two soil horizons is sufficient to generate the recorded macroscopic and geochemical changes.

## 4 Interpretation of landscape evolution at Domata

According to our luminescence age-range and the relative location of the fan surfaces below the marine terrace (see Fig. 2)

of inferred MIS5 age, the majority of the geomorphic landforms at Domata we discuss formed during MIS 3 (29-57 kyr) (Table 1 and Figs. 2 and 7). The sequence of events that resulted in the development of the landscape at Domata, as dictated by the analysis of the geomorphic landforms and the superposition of the luminescence dating onto the well-established sea-level curve of Siddall et al. (2003)[1] (Fig. 9), is as following (from old to young) (Fig. 10): *a)* deposition of the upper-fan materials (Fig. 10a); *b)* river entrenchment leading to abandonment of the upper-fan surface (Fig. 10b); *c)* marine trimming

---

[1] The shape of the sea-level fluctuations varies slightly globally, so we designate in this work the empirical, high resolution, sea-level curve of Siddall et al. (2003) from the Red Sea that covers the last 128 kyr. Siddall et al. (2003) estimate the error in their sea-level curve at ~12 m, so all sea-level elevations discussed subsequently are subject to this uncertainty.

of the upper-fan surface deposits, the fan deposits and its alluvial entrenchment cliff (Fig. 10c); *d)* deposition of the lower-fan materials against the upper-fan materials in its alluvial entrenchment cliff and the sea-cliff in the upper-fan (Fig. 10d); *e)* river entrenchment leading to abandonment of the lower-fan surface (Fig. 10e); *f)* marine trimming of the lower-fan deposits and the alluvial entrenchment cliff (Fig. 10f); *g)* seismic uplift resulting in a stranded paleoshoreline, the development of a river terrace riser and the oversteepening of the lower river channel (Fig. 10g). In the following discussion we provide evidence in support of each stage of the landscape evolution at Domata and establish its relative chronology.

The initiation of deposition of the upper-fan (Fig. 10a) has occurred post ~50 kyr BP and prior to 45 kyr (RB-1 sample; within MIS 3 of Lisiecki and Raymo 2005; Table 1); this coincides with a period of elevated sea-level (c. -70 to -80 m according to Siddall's sea-level curve; Fig. 9). We argue that upper-fan deposition cannot have started as early as 53.4 kyr (mean of RB-1 sample; Table 1) because there is no geomorphic evidence (e.g., marine cliffs) representing the two subsequent high sea-level stands at c. 45 and 39 kyr. Thus, the deposition of the upper-fan postdates 53 kyr and is completed by ~45 kyr BP, when sea-level rose to reach at c. -72m (Siddall et al., 2003). The period that follows, between 45-41 kyr, reflects the start of a cooling climatic period (possibly with an increase in sediment supply) and falling sea-level to -87 m (Fig. 9). During this period, alluvial-fan entrenchment resulted in fan surface abandonment and development of the alluvial cliff in the upper fan deposits (Fig. 3) and the upper-fan was eventually abandoned (Fig. 10b). This was followed, at ~39 kyr BP, by marine transgression resulting in marine trimming of the upper-fan surface, the alluvial entrenchment cliff and development of a sub-horizontal marine abrasion surface in early upper-fan deposits at about the sea-level of the time (c. -70 m) (see yellow dashed line on the river cliff in Fig. 3; Figs. 9 and 10c). This allowed up to 10,000 years for upper-fan deposition and incision before the fan is trimmed by the sea during the high sea-level peak at c. 39 kyr.

Lower-fan deposition commenced (Fig. 10d) soon after marine trimming of the upper-fan (Fig. 9). The relatively high sea-level at 37 kyr (c. -75 m; see Fig. 9) promoted fan deposition and the possibly deteriorating climatic conditions, involving episodically increased river flow/carrying capacity and diminishing vegetation density in the upper catchment, resulted in, increased sediment supply. Lower-fan deposits lapped against alluvial entrenchment and marine cliffs cut previously in upper-fan deposits by alluvial incision along the Klados River and by the marine trimming sub-parallel to the modern shoreline (Fig. 10d). Lower-fan surface abandonment and river entrenchment through incision commenced at ~36 kyr (LF-2a/b sample; Table 1). We argue that the lower-fan surface was abandoned (Fig. 10e) sometime between ~36 kyr and 29 kyr due to river entrenchment resulting from rapidly falling sea-level (that continued until ~18 kyr) (Fig. 9).

Marine trimming of the lower-fan surface and deposits cannot have occurred between deposition and the last glacial maximum (~18 kyr), as sea-level progressively declined during that period. Following 18 kyr, sea-level rose rapidly by ~100 m in less than 10 kyr (Fig. 9). The marine trimming of the lower-fan surface and deposits occurred during the Holocene high sea-level stand. This is expected to have commenced as sea-level approached roughly the present level c. 4-5 kyr ago (Fig.

10f). Between 18 kyr and 5 kyr BP, while sea-level was rising fast, tectonic uplift at Domata must have outpaced rising sea-level, protecting the entire sequence from marine inundation and destruction. Immediately prior to the co-seismic uplift that affected western Crete at AD 365 (Pirazzoli et al., 1982), the foot of the lower marine cliff would have been within the intertidal zone. Today, due to the 6 m of uplift associated with that earthquake, the prominent stranded paleoshoreline and

the foot of the marine-trimmed cliff are at approximately similar levels and the sea cliff may be isolated from further trimming (Figs. 6b and 10g). We interpret a low terrace riser at about this elevation near the mouth of the Klados River (see lower white dashed line in Fig. 6b) to be relict from the river channel of that time and, in the lower c. 100 m of its course, the river has downcut in response to that first-millennium earthquake uplift.

In summary, deposition of the upper and lower-fan was controlled by a marine base level and, in both cases, fan incision resulted due to falling sea-level. The deposition of the upper-fan was largely completed by ~45 kyr BP, during a period of relative high sea-level (c. -70 m), and fan incision resulting in surface abandonment occurred between c. 45 and 40 kyr. Marine trimming of the upper-fan deposits occurred during a sea-level high (c. -72 m) at c. 39 kyr. Lower-fan deposition was initiated soon afterwards and fan surface abandonment occurred between 36 and 29 kyr. The age of the sandy unit that

mantles both the upper- and the lower-fan surfaces is ~25 kyr (UF-2; Fig. 9 and Table 1), postdating both fans as expected by its stratigraphic relationship with respect to the fans (silty sand that mantles both the lower and upper-fan).

## 5 The importance of tectonic uplift at Domata

The fan sequence at Domata provides a unique opportunity to link terrestrial deposition with sea-level fluctuations and vertical tectonics on southwestern Crete. Geomorphic analysis combined with dating shows that the development of the fan

sequence can be accounted for by eustatic changes coupled with vertical tectonics. The latter can be rationalised if we consider that the landforms at Domata were formed 70 to 90 m (within the known error margins) below the current sea-level, implying that, unless tectonic uplift was significant, the entire sequence would have been inundated, and thus modified or destroyed, by the rising sea-level during the last 20 kyr (Siddall et al., 2003). A requirement of its preservation is that since its deposition, the tectonic uplift rate has outpaced rising sea-level. A question that arises from this reasoning concerns the

rate of the tectonic uplift at Domata and how it relates to the rate of rising sea-level. Dating key terrestrial and marginal marine geomorphological features at Domata, thus, provides an average uplift rate for this part of Crete that can be compared to the rate of rising sea-level and also to other rates of tectonic uplift on western Crete (which have been independently derived). It also provides a means of testing the main finding of Pope et al. (2016) that, during late-Quaternary, the landscape at the nearby site of Sfakia mainly responded to sea-level and climatic changes.

Our data show that the marine trimming episode at c. 39 kyr left a coastal cliff and cut an erosional intertidal abrasion surface in the upper-fan deposits (see yellow dashed line in Figure 3). This surface provides a good datum upon which to

estimate subsequent uplift. Indeed, a total uplift of 86 m (± the 12 m error margin of the sea-level curve) is required to elevate this marine abrasion surface to its current altitude of 14 m a.s.l. Thus, the minimum uplift rate required to accomplish this is c. 2.2 mm/yr. Indeed, the plot in Figure 11 shows that with an average rate of ~2.2 mm/yr since formation (black line), the fan sequence would have escaped the destructive interaction with the wave-zone and, therefore, modification due to erosion (e.g., the black line of uniform uplift rate does not intersect the sea-level curve during the last 39 kyr). Independent support for similar uplift rates comes from published radiocarbon ages on beachrock materials that mantle marine paleoshorelines in nearby localities: a calibrated radiocarbon age of 36,790 - 38,694 yrs BP from reworked rhodoliths in beachrock at an elevation of 10.5 m at Sougia, 9 km to the west of Domata, is within a few thousand years of the proposed timing of marine trimming of the upper-fan, and yields a required average uplift rate of 2.4 mm/yr (Mouslopoulou et al., 2015a). Similarly, beachrock on a marine terrace at 17 m elevation at Palaiochora, 20 km west of Domata, yields a calibrated radiocarbon age of 36,682 - 38,732 yrs BP, producing an average uplift rate of 2.5 mm/yr (Mouslopoulou et al., 2015a). Comparable uplift rates (1.8-2.7 mm/yr) have been independently recorded for the last ~40,000 years at numerous localities on western Crete by Shaw et al. (2008), Strasser et al., (2011) and Tiberti et al. (2014). Thus, the preservation and sub-aerial exposure of the landscape at Domata is due to the sufficient tectonic uplift that southern Crete experienced during late Quaternary.

In order to quantify the relative contribution of tectonics and eustacy on the formation of the landscape at Domata, here we compare published information on incremental uplift rates calculated by Tiberti et al. (2014) and Mouslopoulou et al. (2015b) for western Crete over the last ~40 kyr with the uplift rate calculated for Domata over the last 39 kyr (c. 2.2 mm/yr; this study). Comparison shows that during the time period over which the key features at Domata formed (MIS 3), no significant uplift was accommodated on Crete as the region between ~20-45 kyr was experiencing a tectonically quiet period with no uplift (Mouslopoulou et al., 2015b) or even gentle subsidence (Tiberti et al., 2014). Thus, the shaping of the landscape at Domata during MIS 3 must have been largely achieved by sea-level fluctuations. This comparison also suggests that most of the ~80 m of uplift (subtracting 6m of late Holocene co-seismic uplift), have been accumulated sometime between 5 and 20 kyr BP, resulting in an average uplift rate of 5.3 mm/yr (Fig. 11; dashed red line). However, this uplift rate would have been insufficient to outpace the rising sea-level between 8 and 12 kyr BP (see dashed red line intersecting the sea-level curve in Fig. 11) and, thus, the fan sequence would have been inundated and modified or destroyed by the rising sea-level. This, in turn, implies that the uplift rate at Domata was higher than 5.3 mm/yr. We favour a scenario in which uplift was mostly accommodated by about 9 kyr BP, at an average of ~7 mm/yr (see solid red line in Fig. 11 that does not intersect the sea-level curve). Comparable uplift rates (7-8 mm/yr) have been independently recorded at numerous localities on western and eastern Crete for the last 20,000 years by Tiberti et al. (2014) and Mouslopoulou et al. (2015b) and they result from transient earthquake-slip on upper-plate faults that splay off the plate-interface and extend beneath Crete at relatively steep angles (Mouslopoulou et al., 2015). Such transient rates have been observed on several other margins globally (Mouslopoulou et al., 2016 and references therein).

Thus, the development and evolution (~50-20 kyr BP) of the suite of geomorphic features at Domata can be largely explained by eustatic sea-level fluctuations and sedimentation variations controlled by climatic conditions, without the requirement for significant vertical movements. This conclusion largely supports the main finding of Pope et al. (2016), who dated a fan sequence at Sfakia (ca. 25 km to the east of Domata). However, it is the subsequent tectonic uplift that preserved and exposed sub-aerially the coastal geomorphic features.

Intriguing conclusions of the Pope et al. (2016) high-resolution dating work include also that at Sfakia, three sometimes overlapping phases of fan deposition since the last interglacial are separated by two phases of fan entrenchment, the first close to the MIS 5/4 (c. 70 kyr) boundary, the other close to the MIS 2/1 boundary (c. 14 kyr), triggered by major climatic changes. Fan deposition there has to a large degree persisted through stadial and interstadial periods during the last 125 kyr. Periods of entrenchment at Sfakia do not appear to correlate with the two entrenchment periods at Domata. The Sfakia fan is somewhat different from the Domata fan in catchment size (c. 28 km$^2$ compared with c. 11 km$^2$), fan size (5.3 km$^2$ compared with 0.1 km$^2$), the presence of more than one feeder channel at Sfakia, and in the nature of deposits (primarily clast-supported gravels at Sfakia compared with primarily matrix-supported gravels at Domata). Whether these differences are responsible for differences in depositional and entrenchment histories and in preservation of marine cliffs at Sfakia, or differences in local climatic regimes, or vegetation changes, is uncertain. However, one compatible conclusion of Pope et al.'s work with our own is recognition of the importance of base level (sea-level) changes to the process of incision.

## 6 Conclusions

Alluvial fans often provide a useful index with which to decode the information recorded on the landscape in complex tectonic settings, such as those of Eastern Mediterranean. Herein we use analysis of geomorphic landforms and luminescence dating on an alluvial-fan system with two separate periods of depositional activity in Crete, an island straddling the forearc of the Hellenic subduction margin, to constrain its vertical deformation and discuss the contributing factors responsible for its landscape evolution. Our interpretations suggest that sea-level fluctuations in response to varying climatic conditions formed the landscape at Domata during MIS 3 (~57-29 kyr BP). It is, however, because of the fast tectonic uplift that Crete experienced during the subsequent ~20 thousand years that the entire alluvial sequence escaped destruction and/or modification due to marine inundation and is preserved sub-aerially today. Thus, both eustacy and tectonism impacted on the formation and preservation of the landscape at Domata, but over temporally distinct time periods.

**Acknowledgements**

We are grateful to Nikos Mouslopoulos and late Stavros Sartzetakis for their generous help during fieldwork. We dedicate this work to our beloved Cretan friend Stavros, whose spirit now wings through the gorges and across the mountains

guarding the landscape of his homeland, western Crete. We thank the National Cadastre and Mapping Agency of Greece for providing, free of charge, digital elevation maps and imagery for building the DEM's in Figure 4. We are grateful to M. Tiberti, M. Brandon and an anonymous reviewer for several constructive comments that improved greatly this article. We would also like to thank the Associate Editor V. Vanacker and the Editor F. Herman for handling efficiently this submission.

## Author's contributions

V.M, J.B. and D.M conceived the research idea, pursued all associated fieldwork and analysis of the results. A.F. performed the IRSL dating and P.P. pursued the RTK survey. O.O. provided guidance and contributed to the development of the ideas presented in this article. All authors contributed to the writing of the manuscript.

## Competing interests

The authors declare that they have no conflict of interest

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

**Figure captions**

**Figure 1:** Map illustrating the location of Crete within the forearc of the Hellenic subduction margin. The locations of the Hellenic Trough and its splays (the Ptolemy, Pliny and Strabo troughs) are indicated. Labelled arrows show geodetically-derived site velocities (mm/a) relative to a fixed Nubia plate (Reilinger et al., 2010).. The study area at Domata is indicated

by white filled circle while the regions of Sfakia (S), Elafonisi (E), Palaiochora (P) and Agios Georgios (AG) are marked with blue circles. WM=White Mountains. Hillshade is derived from GeoMapApps.

**Figure 2**: Northward view of the beach at Domata illustrating the two generations of fan surfaces and their separate episodes of marine trimming. The location of the Klados gorge, the two elevated marine benches cut on bedrock and the AD 365 uplifted shoreline are indicated (for a close-up view of the bioerosional AD 365 notch see Figure 6c).

**Figure 3:** The fan sequence at Domata looking obliquely towards southeast. The two fan surfaces and their respective stream-incised cliffs are illustrated. The yellow dashed line in the present stream cliff indicates the benched upper-fan erosional surface, which is overlain by the deposits of the lower-fan. This important marker was used to calculate a long-term (39 kyr) uplift rate at Domata (see text for details). White spots mark the location of IRSL samples.

**Figure 4:** a) Digital elevation model of the study area at Domata as viewed obliquely from the southwest. The model is derived by using the nearest neighbour algorithm along with the GPS measurements marked and colour coded by 10 m elevation bands. Note the upper and lower-fan surfaces, each incised following surface abandonment, and each trimmed by marginal marine processes. b) Digital elevation model with National Greek Cadastre Agency orthophoto draped, along with the GPS measurements. Yellow polygon depicts the area illustrated in the DEM of panel (a). Red polygons indicate the localities of the profiles presented in Figure 5a.

**Figure 5: a)** Profiles of fan surfaces projected onto common planes parallel with the modern Klados River channel (above) and parallel with the modern coastline (below). The extent of the volumes of upper-fan materials (pink) and lower-fan materials (light green) are schematically illustrated beneath each measured profile. The locations of each of the luminescence sample points are annotated by yellow dots. The locations of the stratigraphic columns presented in Figure 5b are also indicated. Horizontal and vertical scales for each profile are similar, with a VE of c. 1.25. Note that the downstream slope of the upper and lower-fan surfaces are about the same, both a little steeper than the slope of the modern stream channel. Also note that the upper-fan surface slopes less to the east than the lower-fan surface, so that the upper-fan marine cliff is higher at the eastern end of the beach than at the west. The highest elevation of the lower-fan surface is close to the foot of the incision of the upper-fan, while the upper-fan surface is highest at its incision point. **b)** Schematic stratigraphic columns for the upper (left) and lower (right) alluvial fan deposits. Note that vertical scale bars indicate elevation in metres above mean sea level for each column. The lower-fan column reflects a stratigraphic section close to the junction between the lower-fan coastal cliff and the river entrenchment. The upper-fan section is located close to the western end of the marine-cliff. Their stratigraphic location is indicated on Figure 5a. Note that the right-hand edge of the lower-fan section represents relative competence of materials.

**Figure 6: a)** The present marine cliff at Domata (above), annotated to highlight various sedimentary relationships (below). The cliff comprises mostly moderately bedded gravels of the lower-fan sequence. The lower contact of the lower-fan gravels (white dashed line) is irregular, but sub-horizontal and lower-fan bedding (white dotted lines) laps onto it. Some individual horizons within the lower-fan deposits can be traced laterally for 100's metres, but channelling, bed-lensing and pinch-outs are present. Note also that the exposed marine cliff beneath the upper-fan surface (behind the forested lower-fan surface). **b)** Looking west across the Klados River mouth (foreground), the uplifted shoreline attributed to the AD 365 earthquake (dashed red line, lower left) aligns well with a low terrace riser on the west side of the river (dashed thick white line, middle). Incision of the modern channel below the surface is attributable to post-earthquake adjustment to new base levels. Note the sea-cliffing of the last interglacial? marine terrace (thick red dashed line at the top of the image) and the upper and lower fan deposits. The upper fan and western parts of the lower fan are overlain by accumulated rockfall debris and the solid thin white line approximates its surface. The lower fan surface is marked with fine dotted white line. **c)** Annotated image of the west-end of Domata beach illustrating the relationship between the unconformity at the base of the fan sequence and the AD 365 bioerosional notch. Specifically, the picture shows a dissected erosional surface on older sediments that pre-dates the fan deposits, and the AD 365 bioerosional notch post-dating both the erosional surface and the fan deposits that rest on it. The picture is taken about 150 meters west of the mouth of the Klados River.

**Figure 7:** Individual ISRL ages for all selected aliquots and resulting kernel density estimates. Mean = arithmetic mean, sd = standard deviation, se = standard error, CAM = Central Age Model, MAM = Minimum Age Model. The box plots, below the main graph, describe the dose distribution as follows: median as bold line, box delimited by the first and third quartile, whiskers defined by the extremes within 1.5 interquartile ranges.

**Figure 8: a)** The two alluvial fans with their tree cover (*Pinus brutia*). Arrows indicate the soil-cover that develops on each of the fan surfaces. **b)** The soil development on the lower-fan (LF) surface is ~12 cm (as indicated by the white arrows). **c)** The soil development on the upper-fan (UF) surface is ~40 cm (see white arrows). The fan gravel in both cases is indicated.

**Figure 9:** The ISRL chronology of the geomorphic features at Domata plotted against a simplified version of the global sea-level curve after Siddall et al. (2003). The shaded zone in the background represents Siddall et al.'s stated error range. Large filled circles represent means of ISRL dates, small filled circles medians and the bars are error bars (at 1σ). The geomorphic event sequence described in the text is shown above the figure and the favoured of three alternative high sea-level stands within MIS 3 to have trimmed the upper-fan surface prior to lower-fan deposition, is identified with a solid vertical blue line at 39 kyr BP.

**Figure 10:** The sequence of events that contributed to the development of the landscape at Domata is schematically illustrated: a) Upper-fan deposition due to sea-level high stand (base level); b) Abandonment of the upper-fan surface due to falling sea-level; c) Sea-level rise resulting in the trimming of the upper-fan deposits and cutting a marine bench; d) Lower-fan deposition starts during relatively high sea-level, and laps against both the river incision cliff and the coastal cliff in the upper-fan; e) As sea-level falls, the lower-fan surface is abandoned through entrenchment; f) A return to high sea-level results in coastal trimming of the lower-fan deposits; g) One or more earthquakes in the first millennium AD resulted in 6 m of uplift at Domata and corresponding adjustments to the lower Klados River geomorphology. The approximate chronology of each stage is annotated.

**Figure 11**: The plot discusses the required uplift rate to elevate the marine cliff/bench of the upper-fan from its elevation at genesis (39 kyr BP) to its present elevation (+14 m). The simplified sea-level curve (after Siddall et al., 2003) is illustrated by thick blue line. The black line represents a constant uplift rate of 2.2 mm/yr (established in this study). The red dashed line represents a minimum uplift rate for Domata of ~5.3 mm/yr tailored to empirical data (Tiberti et al., 2014; Mouslopoulou et al., 2015b). The red solid line represents the uplift rate required for the fan-system to escape marine inundation and destruction/modification (see text for details).

**Table 1:** Luminescence dosimetry measurements and IRSL ages (potassium feldspar).

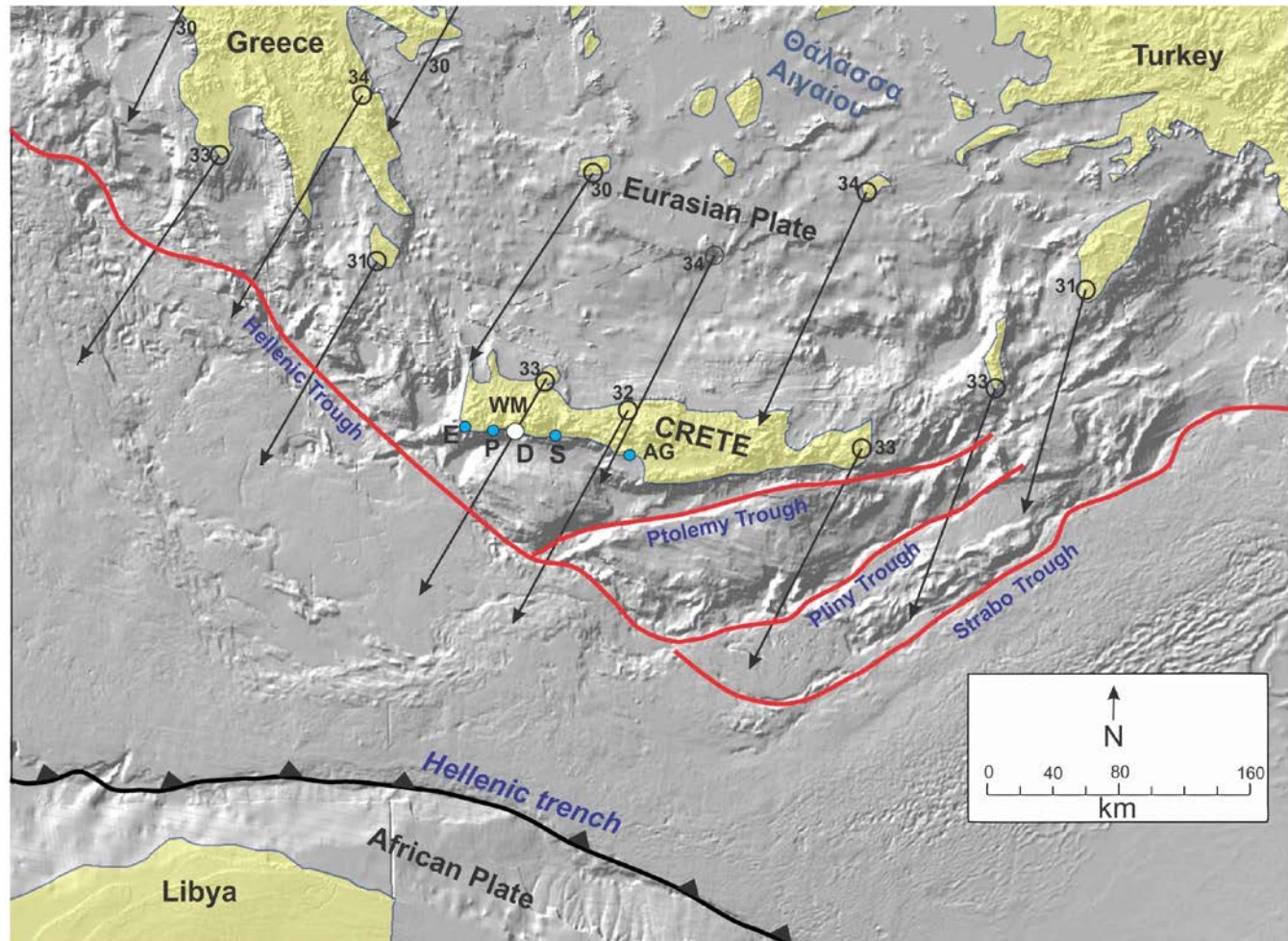

Figure 1

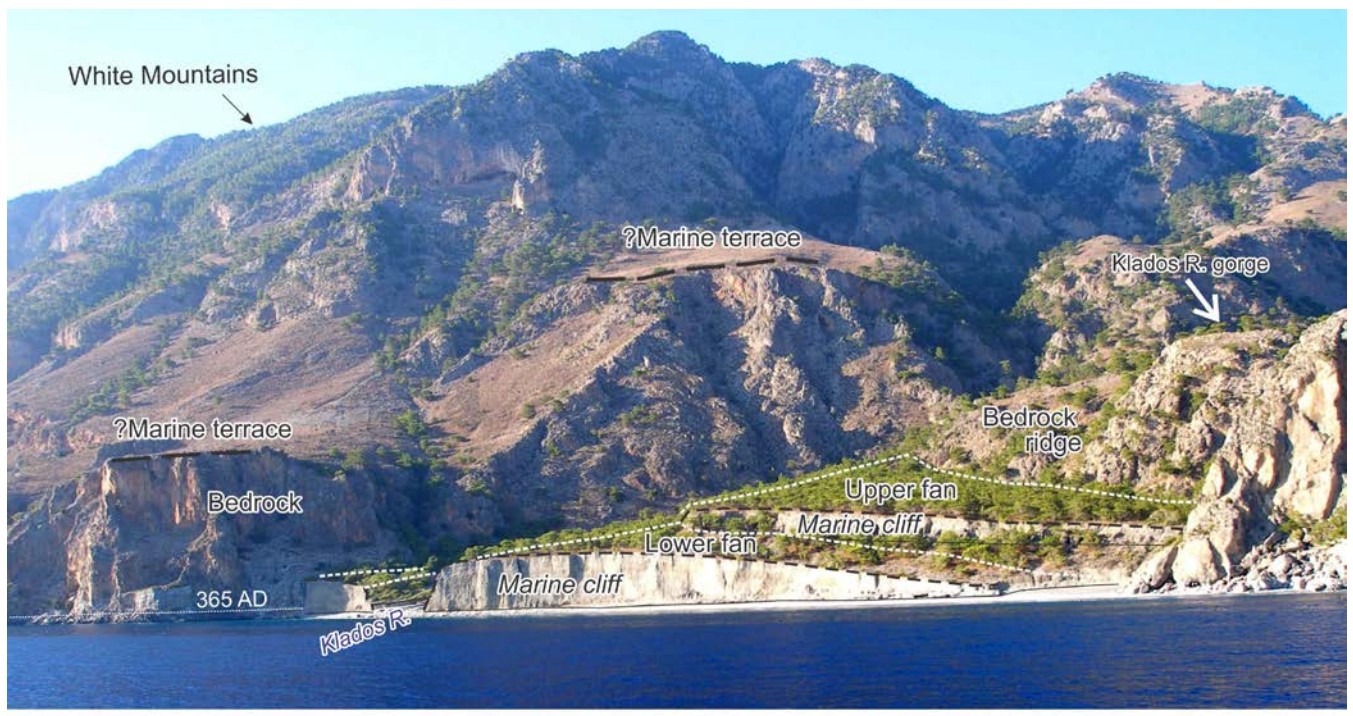

Figure 2

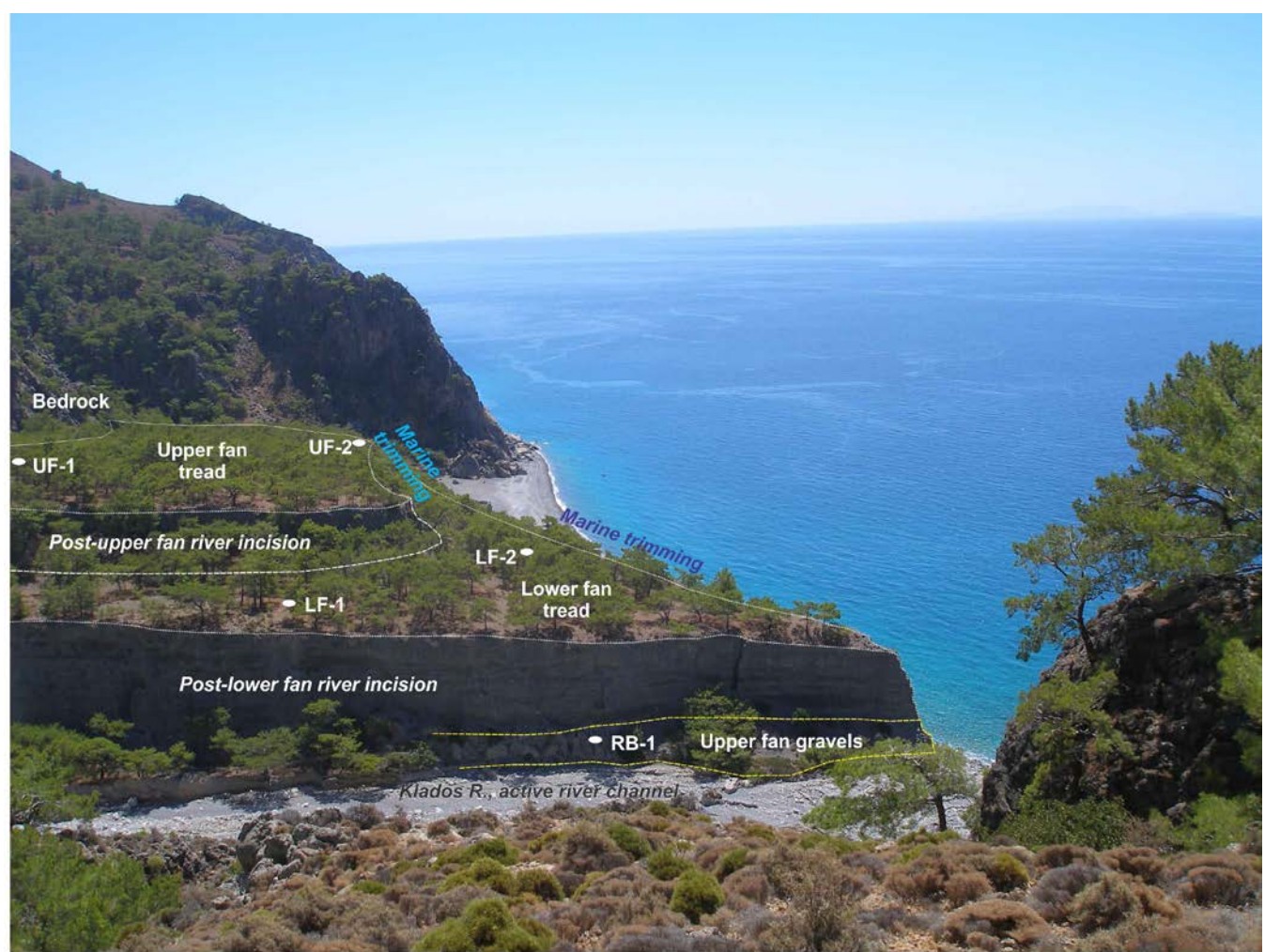

Figure 3

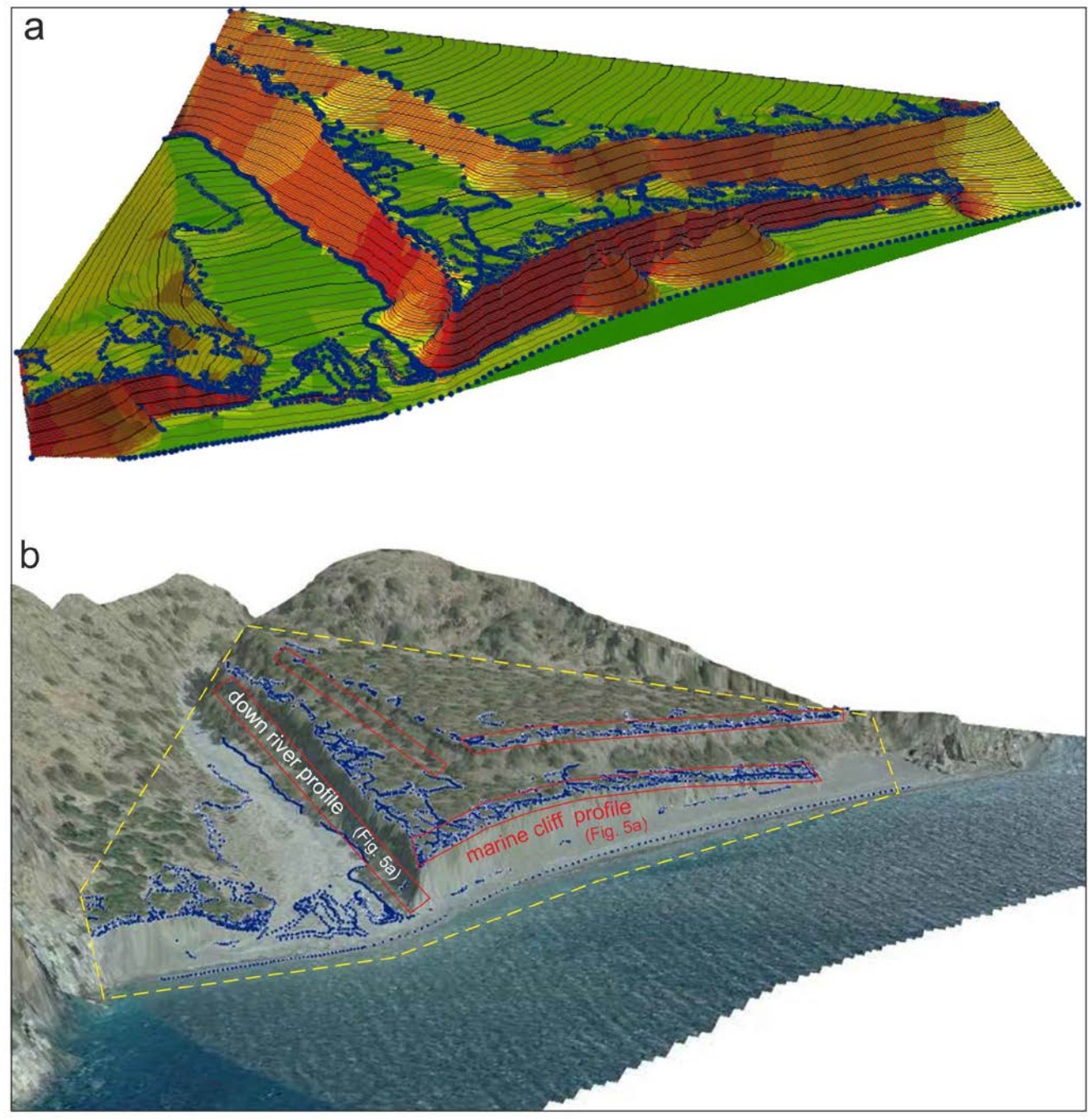

Figure 4

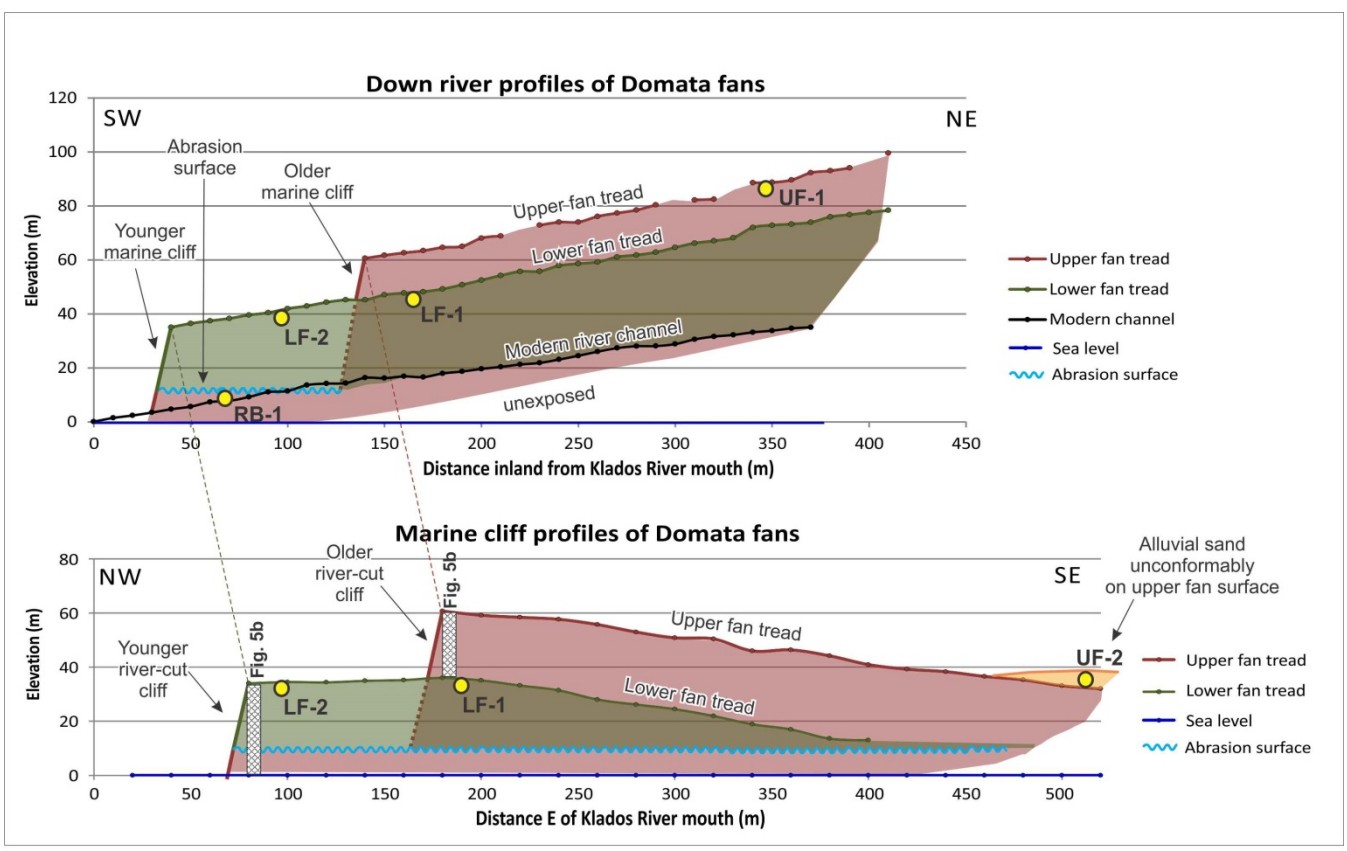

Figure 5a

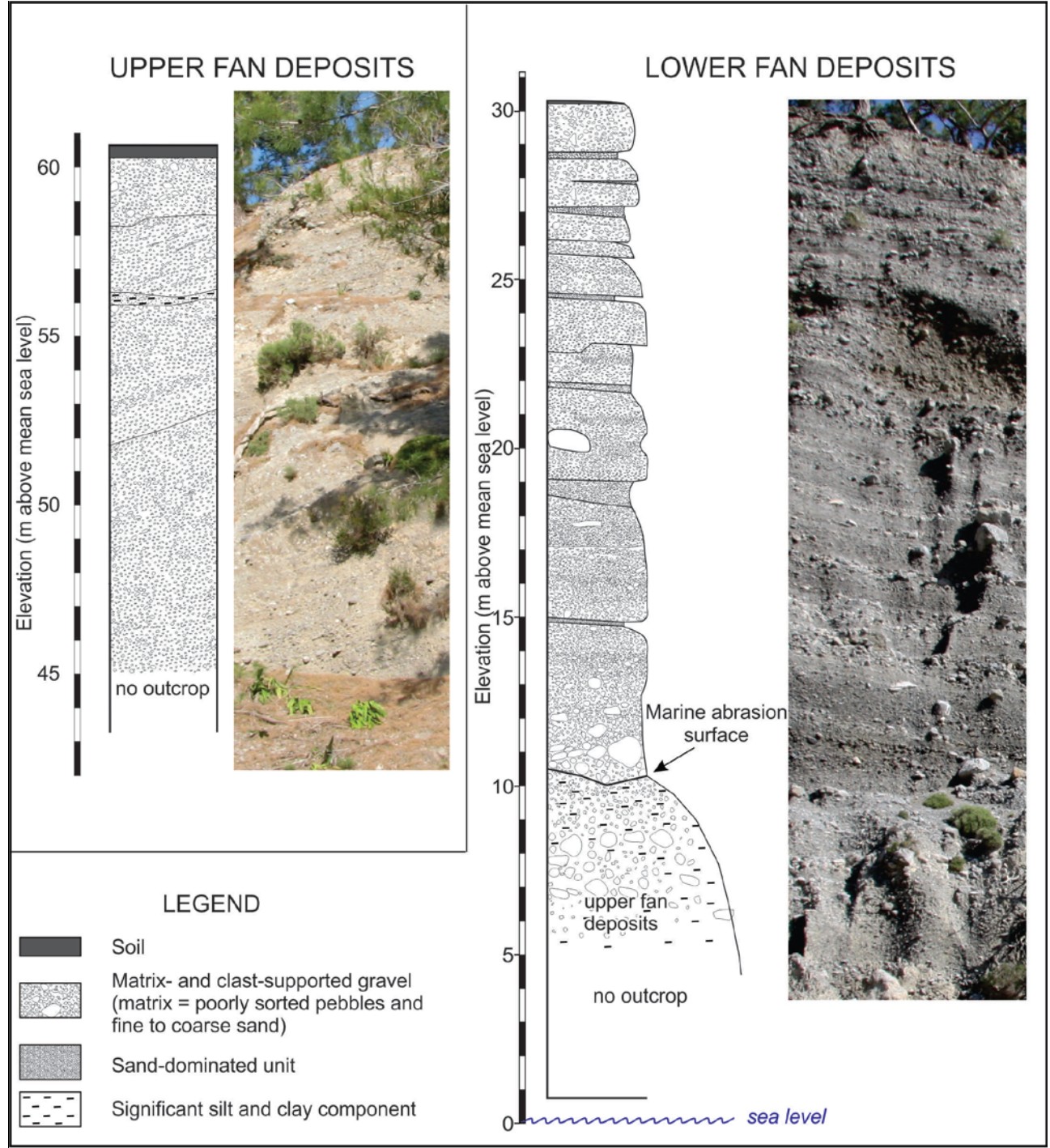

Figure 5b

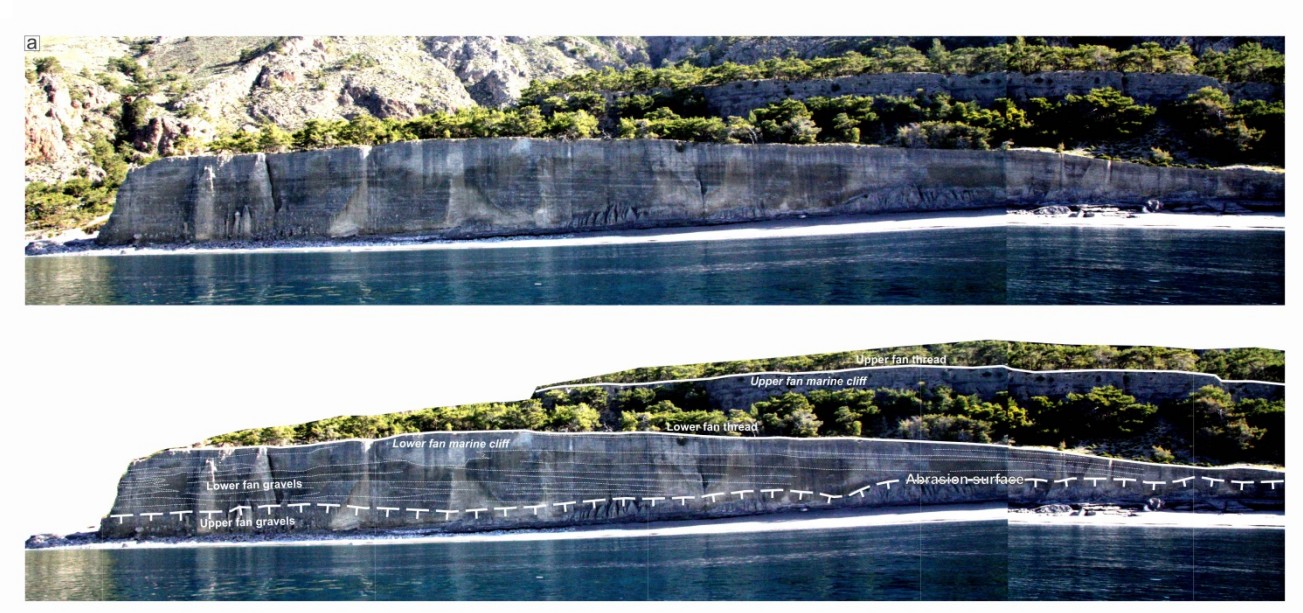

Figure 6a

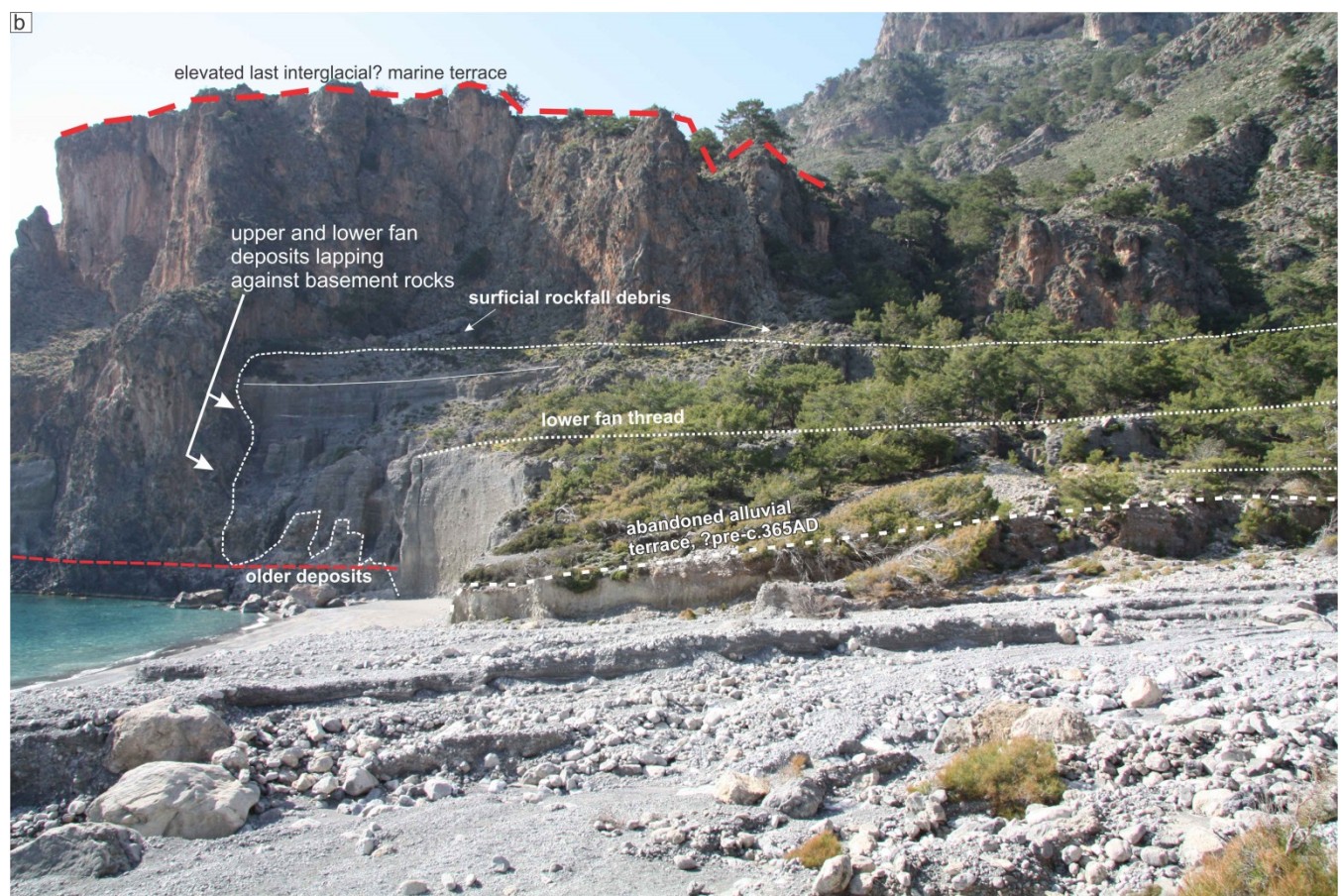

Figure 6b

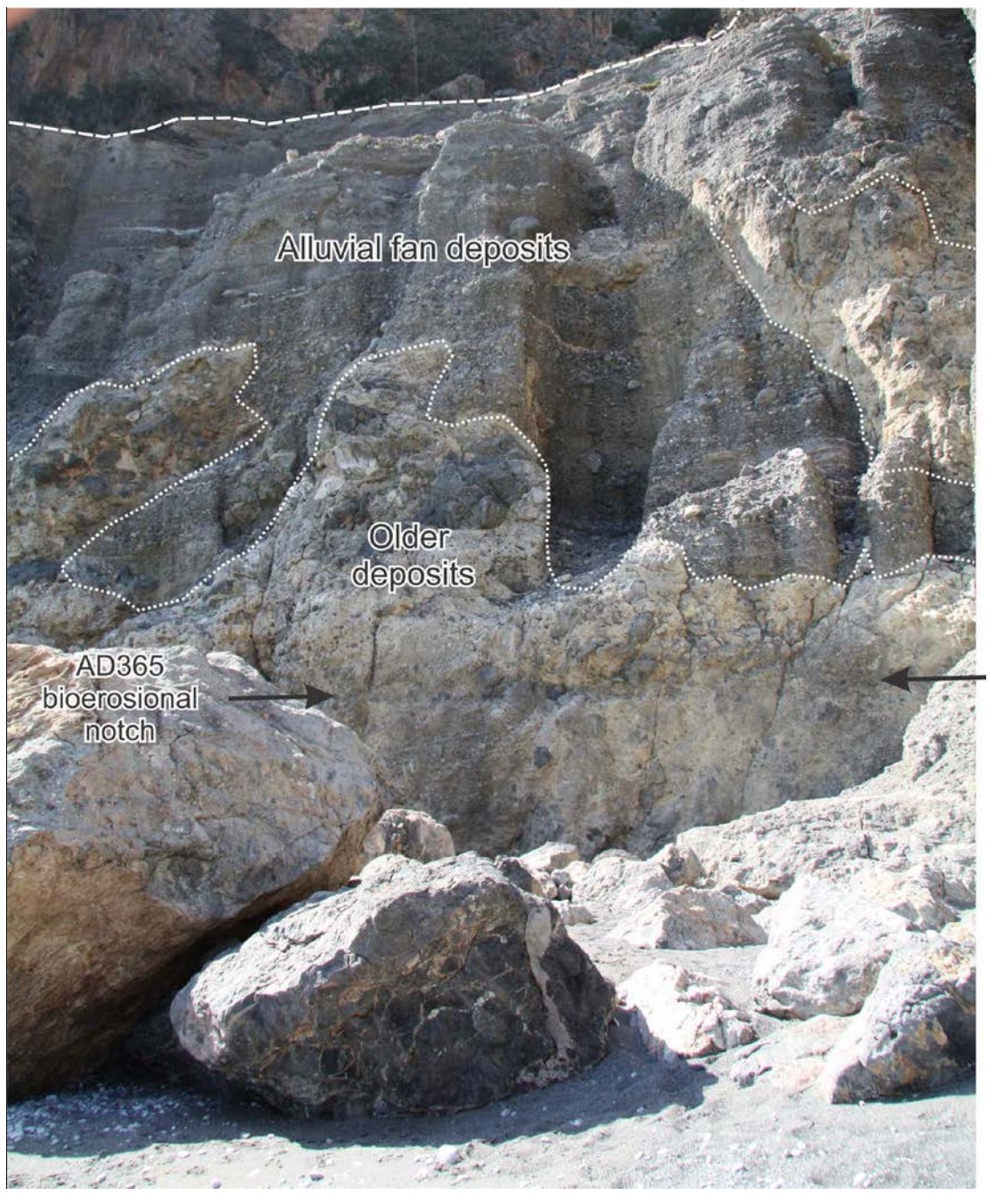

5   Figure 6c

# IRSL age distributions

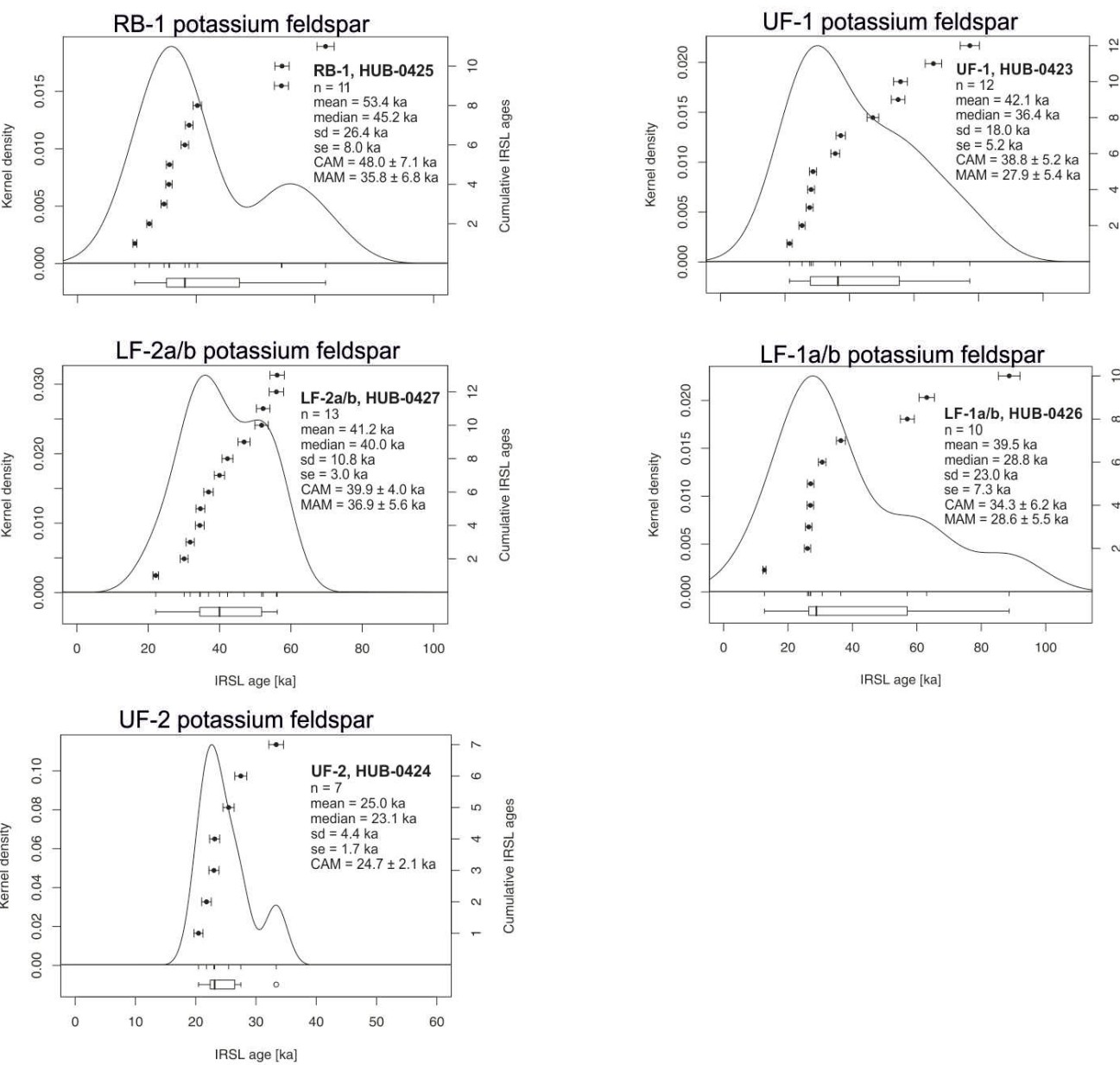

Figure 7

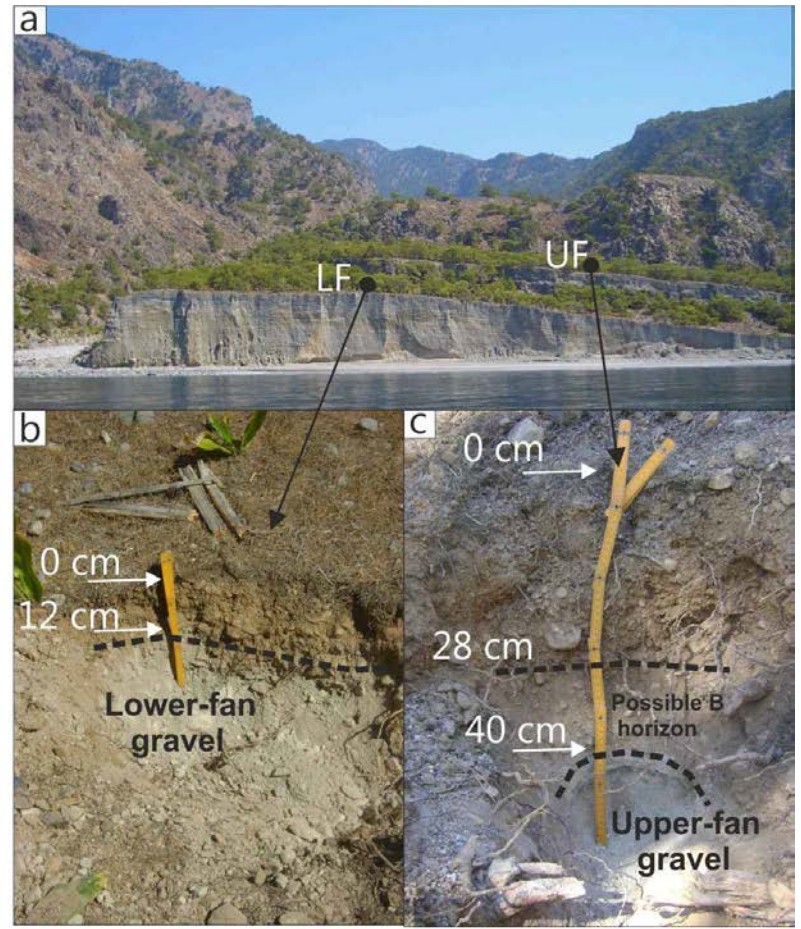

Figure 8

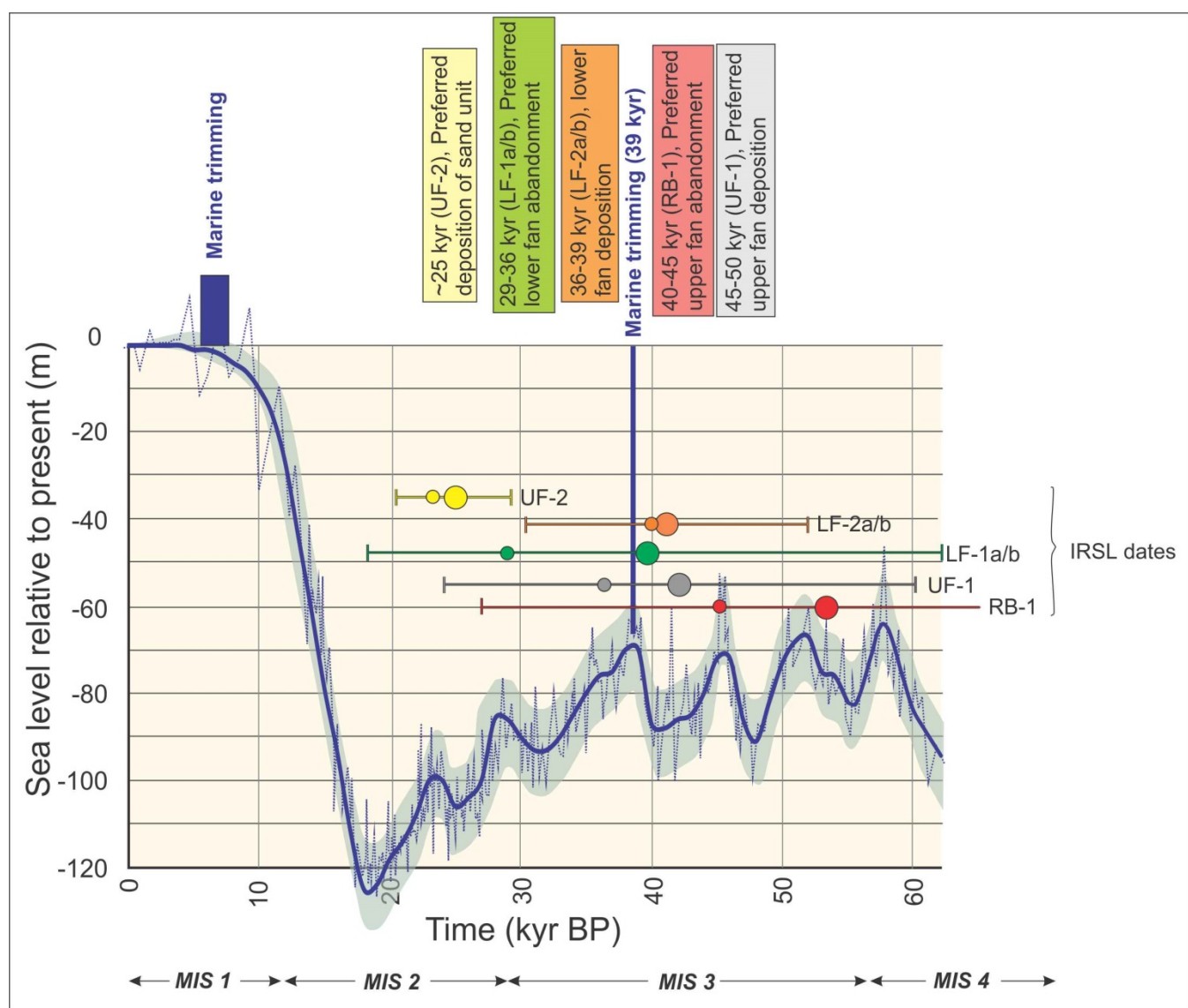

Figure 9

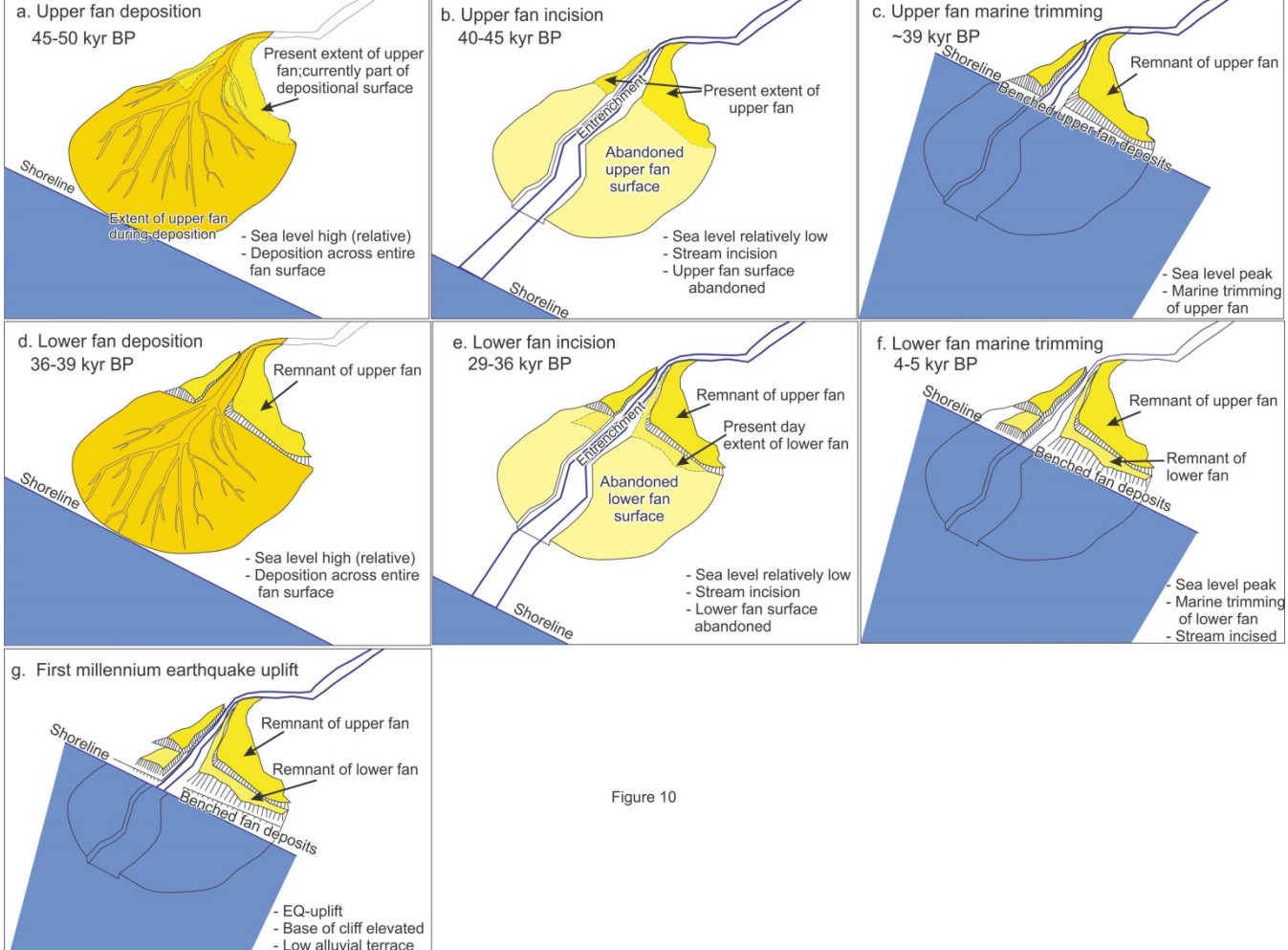

Figure 10

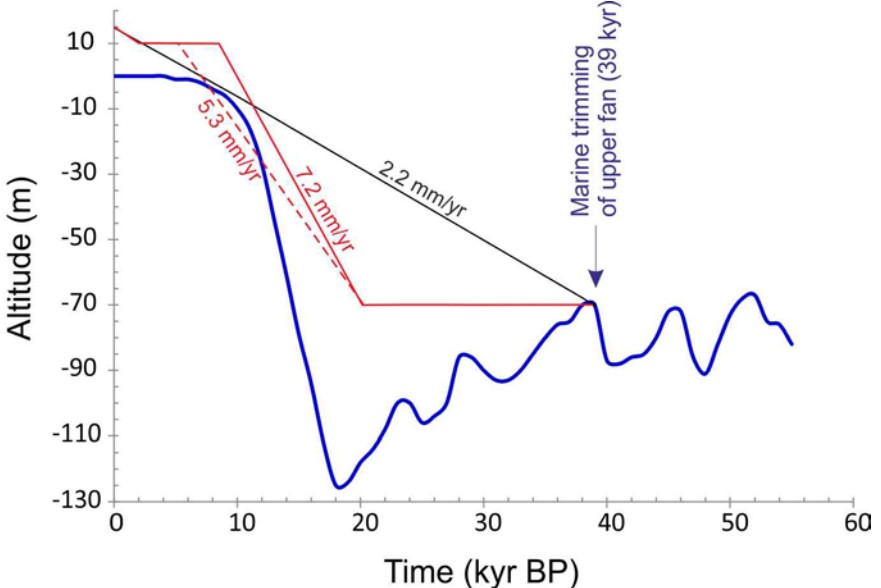

Figure 11

Table 1

| Sample (depth) | Lab. no. (aliquot no.) | U [ppm] (a) | Th [ppm] (a) | K [%] (a) | Cosmic dose rate [mGy/ka] (b) | Water cont. measured [%] (c) | Water cont. estimated [%] (d) | Dose rate (D0) [Gy/ka] (e) | Equivalent dose (De) [Gy] / IRSL age [ka] Mean | Median | CAM (f) | MAM (g) | Standard deviation [%] [ka] | Standard error (h) [%] [ka] | Over-dispersion (i) [%] |
|---|---|---|---|---|---|---|---|---|---|---|---|---|---|---|---|
| UF-1 (0.3 m) | HUB-0423 (12) | 0.46 ± 0.02 | 0.63 ± 0.06 | 0.11 ± 0.01 | 195 ± 20 | 1,1 | 3 ± 2 | 0.99 ± 0.07 | 41.7 Gy / 42.1 ka | 36.0 Gy / 36.4 ka | 38.5 ± 4.4 Gy / 38.8 ± 5.2 ka | 27.7 ± 5.0 Gy / 27.9 ± 5.4 ka | 42.7% / 18.0 ka | 12.3% / 5.2 ka | 39.8% |
| UF-2 (1.1 m) | HUB-0424 (7) | 1.06 ± 0.04 | 2.36 ± 0.13 | 0.53 ± 0.01 | 173 ± 17 | 5,1 | 5 ± 2 | 1.67 ± 0.11 | 41.7 Gy / 25.0 ka | 38.6 Gy / 23.1 ka | 41.2 ± 2.4 Gy / 24.7 ± 2.1 ka | - / - | 17.5% / 4.4 ka | 6,6% / 1.7 ka | 15,0% |
| RB-1 (28.0 m) | HUB-0425 (11) | 0.84 ± 0.02 | 0.62 ± 0.05 | 0.15 ± 0.01 | 17 ± 2 | 1,6 | 3 ± 2 | 0.96 ± 0.06 | 51.3 Gy / 53.4 ka | 43.4 Gy / 45.2 ka | 46.3 ± 6.2 Gy / 48.0 ± 7.1 ka | 34.5 ± 6.2 Gy / 35.8 ± 6.8 ka | 49.4% / 26.4 ka | 14.9% / 8.0 ka | 44.1% |
| LF-1a/b (0.28 m) | HUB-0426 (10) | 0.67 ± 0.02 | 0.4 ± 0.05 | 0.09 ± 0.01 | 194 ± 19 | 0,3 | 3 ± 2 | 1.01 ± 0.07 | 39.9 Gy / 39.5 ka | 29.1 Gy / 28.8 ka | 34.7 ± 5.8 Gy / 34.3 ± 6.2 ka | 29.0 ± 5.2 Gy / 28.6 ± 5.5 ka | 58.2% / 23.0 ka | 18.4% / 7.3 ka | 52,8% |
| LF-2a/b (0.24 m) | HUB-0427 (13) | 0.73 ± 0.02 | 0.67 ± 0.04 | 0.1 ± 0.01 | 195 ± 20 | 1,3 | 3 ± 2 | 1.07 ± 0.07 | 44.0 Gy / 41.2 ka | 42.8 Gy / 40.0 ka | 42.6 ± 3.2 Gy / 39.9 ± 4.0 ka | 39.4 ± 5.4 Gy / 36.9 ± 5.6 ka | 26.2% / 10.8 ka | 7.3% / 3.0 ka | 26.6% |

(a) Uranium, thorium, and potassium contents were determined via high resolution gamma ray spectrometry (HPGe detector).
U-238: U-234 (53.2 keV), Th-234 (63.3 keV), Ra-226 (186.1 keV), Pb-214 (295.2 keV, 351.9 keV), Bi-214 (609.3 keV, 1120.3 keV, 1764.5 keV), Pb-210 (46.5 keV).
Th-232: Ac-228 (338.3 keV, 911.2 keV, 969.0 keV), Pb-212 (238.6 keV), Bi-212 (727.3 keV), Tl-208 (583.2 keV).
K-40: 1461.0 keV.
U-238 and Th-232: The arithmetic means of the activities of the above mentioned natural daughter products were used (± standard error).
The internal K content of the potassium feldspar was set to 12.5 ± 0.5 % (Huntley & Baril 1997).
(b) Cosmic dose rates were estimated regarding geographic position (35°N, 24°E), altitude and sampling depth.
(c) Water content of sediment samples in % of dry mass (oven dried for 24 h at 105°C).
(d) Water content used for dose rate calculation.
(e) For coarse grain potassium feldspar an a-value of 0.15 ± 0.05 was assumed (Balescu & Lamothe 1994).
(f) Central Age Model (CAM) according Galbraith et al. (1999)
(g) Minimum Age Model (MAM) according Galbraith et al. (1999). Sigma b was set to 0.25.
(h) Standard error of the mean: standard deviation devided by the square root of the number of measured aliquots.
(i) The Overdispersion describes the variation of the equivalent dose in addition to the expected error. It is given by the CAM.