# Peer review of "Distinct phases of eustatic and tectonics forcing for late Quaternary landscape evolution in southwest Crete, Greece"

_Earth Surface Dynamics, 2016_

## Short Comment (SC2)

***Clarifying points on response of Mouslopoulou et al. to short comment by Gallen and Wegmann.***

We appreciated the detailed response of Mouslopoulou et al. to our short comment; however, the authors fail to adequately address some of our major concerns. Below we outline several weaknesses of the manuscript that still need to be addressed in the context of the response provided by the authors.

1) **Lack of sedimentologic and stratigraphic context.** In their response, the authors state that their specific objective to use geomorphic markers to assess tectonic signals obviates the need to place the Domata fan sequence in a stratigraphic and sedimentological context. To the contrary, the origin and genesis of the fan sequence is of utmost importance to evaluating their utility as a geomorphic marker. The interpretations forwarded in this manuscript hinge on the assumption that sedimentary deposit at the mouth of the Klados Gorge represents two fan units and that the cliff on the upper fan unit was cut by wave action. However, no evidence is provided to support the interpretation of two fan units. Currently the interpretation of Mouslopoulou et al. that the Domata fan sequence represents two fan units is based solely on morphology. As we pointed out in our short comment, this morphology is not unique to two fans, but can be equally well represented by a fill-cut sequence into a single depositional unit. Essential to the use of the Domata fan sequence is that the cliff in the Upper fan unit was cut by a marine incursion; however no evidence is provided to support this interpretation.

   The geochronology presented in the manuscript is unable to distinguish between two fan units (more on the geochronology below). Therefore, *the only way for the authors' to compelling demonstrate that the Domata fan sequence contains two fan units is by presenting detailed stratigraphic and sedimentological evidence.* The only revision that the authors present is to figure 5, where they now put their *interpretation* of the fan units without providing supporting evidence. Details and observations are required. The stratigraphic information is paramount to the goals of the paper, but the necessary details are not provided, and therefore, the authors' arguments remain unconvincing.

2) **The beach underlying the fan sequence and the 365AD paleo-shoreline.** In their response Mouslopoulou et al. seem to challenge the existence of this beach deposit. The beach deposit underlying the fans is clearly visible in the images that we provided. This is not an interpretation, but a *basic stratigraphic observation* and the photos provided in our previous comment show clear proof that the fan sequence overlies a beach. The existence of the beach is not in question.

   In our short comment we show evidence that portions of the fan sequence overly a wave-cut bench that is associated with the 365 AD paleo-shoreline. This simple, but key observation suggests that the fan sequence is Holocene in age and directly challenges the interpretations made in this manuscript. The authors' fail to address this observation in their response to our comment.

   We reason that because the fan buries a Holocene wave-cut bench, the beach buried by the fan sequence is likely Holocene and by association, was also uplifted in the 365 AD earthquake. Mouslopoulou et al. forward that this interpretation is flawed based on two reasons: (1) they suggest that the 365 AD paleo-shoreline is only represented by a bioerosional notch, and (2) that the elevations of the beach and notch are separated by several meters along the modern coastline. First, the 365 AD paleo-shoreline is not only represented as a bio-erosional notch. Many uplifted

Holocene beaches are preserved all along the western and southern coastlines of Crete (e.g. in Paleochora, Moni Chrisoskalitissis, Damnoni, and numerous other locations). Lateral facies changes are common along the coastlines of Crete and every other coastal environment on Earth. As an example, take the modern Domata shoreline. It is a sediment-rich beach at the mouth of the gorge and is a bioerosional notch on the limestone cliffs to the east and the west of the gorge.

The observation that the beach buried by the fan and the 365 AD notch do not coincide at the modern coastline is not only not surprising, it is an expected observation given the geometry of the different coastal geomorphic markers (paleo beach deposits and bioerosional notches formed in bedrock). Beaches have a topographic slope. The paleo-shoreline (e.g. the elevation where the bedrock notch and beach will meet) will be represented by the inner shoreline angle elevation. This is currently buried by the fan sequence. The 365 AD notch likely continues along the cliff face behind the fan (e.g. a buttress unconformity) and meets up with the inner shoreline angle of the beach upslope from where the beach deposits buried by the fan deposits are visible today.

3) **Luminescence geochronology.** While the authors more thoroughly discuss the luminance results in their response, they fail to demonstrate that the data is reliable. Rather, they explicitly describe problems with the data that more clearly demonstrate that the geochronology is questionable. To quote the authors "yes, there is evidence for incomplete bleaching. Unfortunately, the amount of unbleached signal inherited cannot be quantified clearly." This statement seriously undermines the interpretations presented in the manuscript because they so heavily rely on the geochronology.

4) **Relative age of fan units.** In their response to our comment Mouslopoulou et al. acknowledge that they did not conduct a proper description of the soils in either fan unit. Rather, they use basic visual observations augmented by a "knife penetration". Without quantitative descriptions of the soils it is difficult to compare results to the thorough descriptions of soil profile on other alluvial fans in Crete provided by Pope et al. (2008), Gallen et al. (2014) and Runnels et al. (2014). The soil profiles described by Gallen et al. (2014) do provide a basis for comparison, despite the authors' contention that they do not. Furthermore, as stated in our previous comment, the soils described by Pope et al. (2008) _are not similar_ to the soils shown in figure 8.

One of the key points that we make in our comment is the similar morphologies of the cliff faces on the fan units. In their response the authors' mistake the inferred age of the fan units with the age of the cliffs cut into the fans. In the model presented in figure 10 and from the text in the manuscript the authors' infer that the cliff on the upper fan unit was cut at 39 kyr BP (figure 10, panel c. Upper fan marine trimming) and that the cliff on the lower fan was cut at 4-5 kyr BP (figure 10 panel f. Lower fan marine trimming). There is panels C and F from figure 10.

[Figure]

This interpretation states that the Upper fan cliff is ~ 35 kyrs older than the Lower fan cliff.  So, again we reiterate "*why is the morphology of the cliffs on the Upper and Lower fan units so similar despite an inferred 35 kyr age difference (see figures 5 and 10 in manuscript)? When fault scarps are formed in unconsolidated alluvial fan sediments in places like the Basin and Range of the southwestern United States, they may initially be vertical geomorphic features, but through hillslope erosional processes, the morphology of the scarp changes through time (e.g. McCalpin, 1996). Diffusion of scarps through time has proven to be a useful relative dating tool in studies of both fault scarps (e.g. Nash, 1980) and paleo-shoreline scarps (e.g. Andrews and Bucknam, 1987).  Perhaps this could be attempted in this study.*"

In summary, the authors fail to adequately address our major concerns with the manuscript. The lack of basic stratigraphic observations, questionable geochronology, and poor relative age control on the fans makes it difficult to support their interpretations.

Sincerely,

Sean F. Gallen (sean.gallen@erdw.ethz.ch) and Karl W. Wegmann (karl_wegmann@ncsu.edu)

---

## Short Comment (SC1) · 8 Jan 2017

[supplement omitted: unrelated document]

---

## Referee Comment (RC1) · Anonymous Referee #1 · 18 Jan 2017

**General Comments**

The Mouslopoulou et al. paper deals with a topic that has both environmental and tectonic implications, focusing on a well-expressed sea-side alluvial fan. Compared to the other fans that make up the coastal bajada of south Crete that develops ca 20 km further east, Domata fan is a relatively small one (area c. 0.2 km$^2$), along with a string of other isolated fans that border the steep southern Cretan coast. Nonetheless, its stepped morphology, with an escarpment running roughly parallel to the present-day coastline may be indicative of processes that significantly affect fan-shape evolution, as the 'marine trimming' suggested by the authors. Indeed, Mouslopoulou et al. have tried to link fan evolution to climatic changes, coupled with uplift rate scenarios for the past few kyr and come up with a scenario that ties the evolutionary stages of the fan to successive marine high- and lowstands.

To achieve such a target, one has to resort to high-res geochronology, something that is not always feasible. Her lies, in my opinion, the main drawback of Mouslopoulou et al.'s work, which is found in their sampling strategy and the resulting dating accuracy: while the latter may not be the authors' fault, the former is a weak point in this work. Sampling did not include the body of the 'lower fan': this might clarify —and probably strengthen the authors' distinction of the Domata fan in an 'upper' and a 'lower' one. (Of course, this might not help, either, as the results from the other samples contain significant errors).

Having said these, I acknowledge the fact that the number of dated samples (and possibly the range of dating methods applied) are dictated by research funding; nonetheless, one has to make do with what they have, I'm afraid.

At any rate, the resulting OSL ages contain significant errors (as also acknowledged by the authors: P.8, l. 14); it is also unclear whether standard deviation and standard error refer to 1-σ or 2-σ. Hence, temporal resolution is too poor to support such a detailed evolution scenario, whose resolution is as high as 1kyr, while most ages overlap significantly and the correlation of successive events to KDE$_{max}$ is not satisfactorily constrained.

The lithostratigraphy of the Domata fan is poorly described; lithostratigraphic logs are missing —and these could help place the obtained samples in a coherent geological context. Moreover, it might clarify any probable lithological or other difference between the units described as "upper" and "lower" fan. This could also be aided if appropriate figures (esp. photographs) were included, to show the lithological composition of the fan(s). Panoramic photos are fine, but some close-ups would be very useful for the reader to understand the distinction made between Upper and Lower Fans. Figure 8 (b and c) focus on the soil cover and do not serve this purpose.

Between Domata and Sougia (c. 9 km to the west) there is a number of fans in practically the same a geomorphological and geological environment (the same could also be supported for the bajada the east, with Sfakia fan being its westernmost member). Processes suggested in this paper (i.e. "marine trimming") are not localized ones and affect extended tracts of land. So if such a process was responsible for the modification of the Domata fan, why it is not found elsewhere along this coast?

The authors did not take into account the work by Pope et al., (2016), on the nearby, well-studied Sfakia fan. This may be due to the fact that the m/s postdated the publication date of

this paper, but nonetheless, the authors should take it into consideration in their revised version.

**Specific Comments**

Page 4, l. 20-21. Bedrock geology is grossly misrepresented, both in terms of lithology and age. Klados Gorge runs through platy crystalline limestones with phyllite intercalations and chert-bearing dolomites, chert-nodule-bearing limestones and quartzitic sandstones and shales, not through the platform carbonates of the Tripolis Unit (which is a Mesozoic carbonate platform). Moreover, the aforementioned lithologies (i.e. Kingilos group) belong to the metamorphosed Plattenkalk Unit, also known as Mani Unit. (e.g. Creutzburg and Siedel, 1975; Fassulas et al., 1994; and Jolivet at al., 1996) .

This suffices to explain the occurrence of quartz detritus in the Domata fan; the authors, however, did not seem to wonder why a purely carbonate-fed fan (as they describe it), contains so much quartz!

P.3, l.20. Please use appropriate term instead of "vertical deformation".

P.21, Figure 2. How confident are the authors that the quasi-planar landforms west of the gorge are marine benches? Is there any piece of evidence supporting this suggestion?

P25, Figure 6b. The red dashed line that is suggested to represent a low terrace riser on the west side of the river seems rather ambiguous; it is hard to say from this photo.

**Technical Corrections**

N/A

**References cited**

Creutzburg N., Siedel. E., (1975). Zum stand der Geologie des präneogens auf Kreta. N. Jb. Geol. Paläont. Abh. 149(3): 363-383.

Fassulas, Ch., Kilias, A., Mountrakis, D., (1994). Post-nappe stacking extension and exhumation of the HP/LT rocks in the island of Crete, Greece. Tectonics 13: 127-138.

Jolivet, L., Goffe B., Monie P., Truffert C., Patriat, M., Bonneau M. (1996). Miocene detachment in Crete and exhumation P-T-t paths of high pressure metamorphic rocks. Tectonics 15: 1129-1153.

Pope, R.J.J., Candy, I., Skourtsos, E., (2016). A chronology of alluvial fan response to Late Quaternary sea level and climate change, Crete. Quat. Research 86: 170-183.

---

## Referee Comment (RC2) · M. M. Tiberti (Referee) · 24 Jan 2017

**Review of Mouslopoulou et al. by M. M. Tiberti**

**General comments**

This paper presents the results of measurements and datings on a fan deposition sequence at Domata, on the southern coast of Crete (Greece). Crete Island is a key region in the Mediterranean area, as it is one of the few pieces of emerged land in the forearc domain of the Hellenic subduction zone. Dating geomorphic markers such as alluvial fans and comparing the results with the eustatic curve can contribute to separate the eustatic and tectonic components of apparent coastal uplift, thus constraining vertical tectonic rates. The Authors reconstruct the deposition history of the two alluvial fans and constrain their temporal evolution using IRSL results only (because OSL did not work properly) and stratigraphic considerations. Their results imply that the formation of the two alluvial fan was controlled mainly by climatic and eustatic factors during the general marine regression of MIS 3. No tectonic contribution is necessary to explain their evolution during this period. The state of preservation of the alluvial fans, however, requires that they have been never submerged during the sea-level rise after the last glacial maximum, thus implying that tectonic uplift rates constantly outpaced the eustatic rise during the last 20 ka.

This work presents new data that add an important piece of knowledge to a growing dataset of dated geomorphic markers along the coast of Crete by various authors. These data contribute to constrain the Quaternary geologic evolution of the coast of Crete, helping in separating climatic, eustatic and tectonic components in the Hellenic subduction zone. Hence, the paper address relevant scientific questions within the scope of ESurf, reaching substantial conclusions.

Despite the general good quality of the work, however, the paper lacks clarity and readability. I suggest the Authors to reorganize it in the classical section "Introduction", "Method", "Results" and "Discussion". The Method section should contain the description of how the survey and the datings were carried on, including technical details on OSL and IRSL determinations and soil analysis. The Results section is supposed to contain the Authors' findings, the measurements and age determinations for the alluvial fan deposits and the tentative reconstruction of their evolution. I recommend the Authors to not include other workers' results in this section. Comparison with other researchers' results should be placed in the Discussion section, along with the implications of their own results upon tectonic rates estimates. Considerations on the reliability and accuracy of the datings and their implication should also be included in the Discussion.

In particular, the Author should add the discussion of their results in the general framework of existing data, including those that lead to different interpretations. This important point is at present almost completely disregarded. They should also discuss the intrinsic limitations of the methods used and the consequent implications.

**Specific comments**

There is some confusion about OSL and IRSL datings throughout the text and in tables and pictures. In the text the Authors state that two kind of measurements were performed: quartz OSL and feldspar IRSL. Quartz OSL datings results were not used to constrain the evolution history of the alluvial fans, as they proved to be of poor quality. They are cited in the text, but never appear in tables or pictures. Figure 7 and table 1 apparently report only the results for feldspar IRSL. The Authors should be specific (i.e.: "quartz OSL" and/or "feldspar IRSL") when refer to these measurements, as the use of the general term "OSL" in titles and captions could be somehow confusing. In addition, quartz OSL results should be shown in any case, at least as supplementary material.

In the text, the Authors repeatedly state the "uniqueness" of the Domata site, without discussing it. Are they sure that there is not any similar situation along the southern coast of Crete? What is the difference between the Domata site and the other alluvial fan sequences described in the literature? (e.g. Peterek, et al., 2003; Pope et al., 2008; 2016).

**Technical and other line-by-line comments**

**Abstract**

Line 20: conventionally, the last glaciation corresponds to MIS2 not MIS3.

Line 23: *most* instead of *mot*

**1 Introduction**

Line 30: please specify which sea-level curve are you using. From Figure 9 it turns out to be the one by Siddal et al. (2003). What do you mean with "international"? The curve by Siddall et al. is reconstructed using oxygen isotope records from Red Sea sediment cores.

**2 Geological setting of Crete and vertical tectonics**

Lines 11-16: see also Zachariasse et al., 2008.

Line 18-21: *Using dated paleoshorelines and numerical models, it is shown that the island of Crete experienced, during the last 20 thousand years, periods of severe uplift (at rates of up to 8 mm/yr) while in the preceding ~30 thousand years, the vertical deformation on Crete was minimal (Mouslopoulou et al., 2015b).* Please add also Tiberti et al. (2014; already cited in other parts of the text) to Mouslopoulou et al. (2015b), as they state "*Attaining the S4 to S5 vertical separation thus requires a net subsidence rate of 2.6–3.2 mm/y in the period from ~42 to 23 ky ago. A period of sustained uplift of ~7.7 mm/y should have then followed the S5 abandonment (~23 ky ago) as suggested by the formation of S2, with sea level at -120 m, and S3 and S1-low with sea level at about the same elevation as today*".

Line 24: please add also Tiberti et al. (2014; same reason explained above).

Line 28: Strasser et al., *2011* instead of *2010*

Lines 28-30: Not only historical accounts, however, about the tsunami: see, for instance, Polonia et al., 2013.

**3.2 OSL dating of alluvial fans**

Line 21: *The results of OSL analysis are presented in Table 1 and Figure 7.* Please specify exactly which kind of analysis: both Table 1 and Figure 7 seem to show only IRSL results.

**3.2.2 OSL results**

Page 8, Line 15: *IRSL* instead of *OSL*

**3.3 Soil development**

Line 33: *IRSL* instead of *OSL*

Page 9, Line 14: *IRSL* instead of *OSL*

**4 Landscape evolution at Domata**

Line 18: *international sea-level curve*: please remove "international"

Page 10, Line 31: *deposition of the upper and lower-fan deposits*: please remove "deposits"

**5 The importance of tectonic uplift at Domata**

Lines 24-30: for the sake of completeness, you should mention the other average uplift rates estimates over the last 50 ky based on quantitative datings along the SW coast of Crete:
1.5 mm/y by Wegmann (2008)
2 mm/yr by Shaw et al. (2008)
1-1.5 mm/y by Strasser et al. (2011)
2.5-2.7 mm/y by Tiberti et al. (2014)

Page 12, Lines 5-6: *no significant uplift was accommodated on Crete as the region between ca. 20-45 kyr was experiencing a tectonically quiet period (Mouslopoulou et al., 2015b).* Please notice that for the same period (42 to 23 ky ago), Tiberti et al. (2014) postulated a net subsidence rate of 2.6-3.2 mm/y.

Page 12, Lines 14-15: *Comparable uplift rates have been independently recorded at numerous localities on western and eastern Crete for the last 20,000 years by Shaw et al. (2008), Tiberti et al. (2014) and Mouslopoulou et al. (2015b).* Shaw et al. (2008) never mention an uplift rate of 7 mm/y or similar values over the last 20 ky. They estimate a ca. 2 mm/y uplift rate on the basis of a 20-24 m elevated shoreline dated 41-53 ka.

**6 Conclusions**

Line 23: *IRSL* instead of *OSL*

**Figure 1**

Please add a coordinates reference frame. Use bold for numbers indicating GPS values. Enlarge letters indicating the sites on the southern coast of Crete and use a darker color (e.g. blue instead of yellow) for the circles. In the caption: *WM* instead of *WG* = White Mountains.

**Figure 6**

Caption, Line 5: *100 m* instead of *100's metres*

**Figure 7**

Caption, Line 11: *IRSL* instead of *OSL*. Please change this also in the picture.

**Figure9**

Caption, Line 21: *IRSL* instead of *OSL.* Please change this also in the picture.

**References cited in this review**

Mouslopoulou, V., Nicol, A., Begg, J., Oncken, O., and Moreno, M. (2015) Clusters of mega-earthquakes on upper plate faults control the Eastern Mediterranean hazard, Geophys. Res. Lett., 42, 10282–10289.

Peterek, A., Beneke, K., Schwarze, J. & Spinn, A. (2003) Küste und Küstenformung in Westkreta als Spiegelbild eustaticher, tektonischer und gravitativ-tektonischer Prozesse. Essener Geog. Arb. 35, 39–56.

Polonia, A. et al. (2013) Mediterranean megaturbidite triggered by the AD 365 Crete earthquake and tsunami. Sci. Rep. 3, 1285, doi: 10.1038/srep01285.

Pope, R., Wilkinson. K., Skourtsos, E., Triantaphyllou, M., and Ferrier, G. (2008) Clarifying stages of alluvial-fanevolution along the Sfakian piedmont, southern Crete: New evidence from analysis of post-incisive soils and OSL dating, Geomorphology, 94, 206-225.

Pope, R.J.J., Candy, I., and Skourtsos, E. (2016) A chronology of alluvial fan response to Late Quaternary sea level and climate change, Crete: Quaternary Research, v. 86, p. 170-183.

Shaw, B. et al. (2008) Eastern Mediterranean tectonics and tsunami hazard inferred from the AD 365 earthquake. Nature Geosci. 1, 268–276, doi: 10.1038/ngeo151.

Siddall, M., Rohling, E. J., Almogi-Labin, A., Hemleben, Ch., Meischner, D., Schmelzer, I., and Smeed, D. A. (2003) Sea-level fluctuations during the last glacial cycle, Nature, 423, 853–858.

Strasser, T. F. et al. (2011) Dating Palaeolithic sites in southwestern Crete, Greece. J. Quaternary Sci. 26, 553–560, doi: 10.1002/jqs.1482.

Tiberti, M.M., Basili, R. & Vannoli, P. (2014) Ups and downs in western Crete (Hellenic subduction zone). Sci. Rep. 4, 5677; DOI:10.1038/srep05677.

Wegmann, K. W. Tectonic geomorphology above Mediterranean subduction zones: northeastern Apennines of Italy and Crete, Greece. (2008) Ph.D. Thesis, Bethlehem, Pennsylvania, Lehigh University, 169 p.

Zachariasse, W. J., van Hinsbergen,D. J. J. & Fortuin,A. R. (2008) Mass wasting and uplift on Crete and Karpathos during the early Pliocene related to initiation of south Aegean left-lateral, strike-slip tectonics. Geol. Soc. Am. Bull. 120, 976–993, doi:10.1130/b26175.1.

---

## Author Comment (AC2) · 28 Jan 2017

**Response to the short comment of M. M. Tiberti**

*We thank the reviewer M. Tiberti for her useful comments. Below we respond in detail to her comments.*

**Review of Mouslopoulou et al. by M. M. Tiberti**
**General comments**
This paper presents the results of measurements and datings on a fan deposition sequence at Domata, on the southern coast of Crete (Greece). Crete Island is a key region in the Mediterranean area, as it is one of the few pieces of emerged land in the forearc domain of the Hellenic subduction zone. Dating geomorphic markers such as alluvial fans and comparing the results with the eustatic curve can contribute to separate the eustatic and tectonic components of apparent coastal uplift, thus constraining vertical tectonic rates. The Authors reconstruct the deposition history of the two alluvial fans and constrain their temporal evolution using IRSL results only (because OSL did not work properly) and stratigraphic considerations. Their results imply that the formation of the two alluvial fan was controlled mainly by climatic and eustatic factors during the general marine regression of MIS 3. No tectonic contribution is necessary to explain their evolution during this period. The state of preservation of the alluvial fans, however, requires that they have been never submerged during the sea-level rise after the last glacial maximum, thus implying that tectonic uplift rates constantly outpaced the eustatic rise during the last 20 ka.

This work presents new data that add an important piece of knowledge to a growing dataset of dated geomorphic markers along the coast of Crete by various authors. These data contribute to constrain the Quaternary geologic evolution of the coast of Crete, helping in separating climatic, eustatic and tectonic components in the Hellenic subduction zone. Hence, the paper address relevant scientific questions within the scope of ESurf, reaching substantial conclusions.

Despite the general good quality of the work, however, the paper lacks clarity and readability. I suggest the Authors to reorganize it in the classical section "Introduction", "Method", "Results" and "Discussion". The Method section should contain the description of how the survey and the datings were carried on, including technical details on OSL and IRSL determinations and soil analysis. The Results section is supposed to contain the Authors' findings, the measurements and age determinations for the alluvial fan deposits and the tentative reconstruction of their evolution. I recommend the Authors to not include other workers' results in this section. Comparison with other researchers' results should be placed in the Discussion section, along with the implications of their own results upon tectonic rates estimates. Considerations on the reliability and accuracy of the datings and their implication should also be included in the Discussion. *[We thank the reviewer for this suggestion. However, we will not change the structure of the paper, as the current structure is pretty classical: introduction, geological setting, data-methods, interpretation, and conclusions. What M. Tiberti suggests is not too dissimilar to what it already exists. Also, we don't agree that the comparison with other people's results should be done in an independent section. We feel it is more natural if comparisons are made within each relevant section. From our text, we believe, it is clear which results are our own results and which are published].*

In particular, the Author should add the discussion of their results in the general framework of existing data, including those that lead to different interpretations. *[We interpret our chronology of fan development with existing chronologies of other fans along the southern coastline of Crete (e.g., Nemec & Postma, 1993; Gallen et al., 2014; Pope et al., 2008, 2016 – the latter included in the revised version). We also compare the uplift rate derived here, with other published uplift rates derived from study of marine terraces on western Crete (Shaw et al., 2008; Strasser et al., 2011; Tiberti et al., 2014; Mouslopoulou et al., 2015 – all references included in the submitted version) ].* This important point is at present almost completely disregarded. *We think that this is not the case. The following parts of our ms address this issue:*
 *1) Page 6, lines 6-12*
*2) Page 9, lines 9-11*
*3) Page 11, lines 24-32*
*4) Page 12, lines 14-15*
*The fact that some of these comparisons are made with our own published work is something we cannot avoid. Perhaps the reviewer means that we didn't include their work (Tiberti et. al., 2014) in lines 1-15 of Page 12, and this oversight has been corrected. We should have referred to their work which, for the last 20 kyr, also shows rapid uplift in western Crete whereas for the period 20-40 kyr shows slow subsidence. This, however, won't lead to any different conclusions of the current manuscript. In addition, we have now also included in the revised version a comparison of the results of our work with that published in Pope et al. (2016). The latter article was*

*missing from our submitted version as our work predated the publication of Pope et al., 2016].* They should also discuss the intrinsic limitations of the methods used and the consequent implications. *[We have expanded our discussion of the stratigraphic and geomorphic requirements for development of the Domata fan sequence and comment more specifically on errors in our IRSL dates and the way we use them].*

**Specific comments**
There is some confusion about OSL and IRSL datings throughout the text and in tables and pictures. In the text the Authors state that two kind of measurements were performed: quartz OSL and feldspar IRSL. Quartz OSL datings results were not used to constrain the evolution history of the alluvial fans, as they proved to be of poor quality. *[Please see our detailed response to Gallen and Wegmann comment about this issue].* They are cited in the text, but never appear in tables or pictures. *[We agree. We should use the term IRSL instead of OSL throughout the text].*

Figure 7 and table 1 apparently report only the results for feldspar IRSL. The Authors should be specific (i.e.: "quartz OSL" and/or "feldspar IRSL") when refer to these measurements, as the use of the general term "OSL" in titles and captions could be somehow confusing. In addition, quartz OSL results should be shown in any case, at least as supplementary material. *[See above].*

In the text, the Authors repeatedly state the "uniqueness" of the Domata site, without discussing it. Are they sure that there is not any similar situation along the southern coast of Crete? What is the difference between the Domata site and the other alluvial fan sequences described in the literature? (e.g. Peterek, et al., 2003; Pope et al., 2008; 2016). *[Please see our response to Gallen and Wegmann's short comment where we explain in detail that the geomorphology at Domata, and the Klados River Gorge itself, differ to other gorges in south Crete (For example: The Sfakia fan (Nemec & Postma, 1993; Gallen et al., 2014; Pope et al., 2008, 2016), is significantly different from the Domata fan in catchment area (c. 28 km$^2$ compared with c. 11 km$^2$), fan size (5.3 km$^2$ compared with 0.1 km$^2$), the presence of more than one feeder channel at Sfakia, and in the nature of deposits (primarily clast-supported gravels compared with primarily matrix-supported gravels) ].*

**Technical and other line-by-line comments**

**Abstract**
Line 20: conventionally, the last glaciation corresponds to MIS2 not MIS3.
Line 23: *most* instead of *mot*
**1 Introduction** Line 30: please specify which sea-level curve are you using. *[See response to Gallen & Wegmann review].* From Figure 9 it turns out to be the one by Siddal et al. (2003). What do you mean with "international"? *We have now removed international. Our reasons for using the Siddall et al. curve are explained in our response to the Gallen & Wegmann's review.* The curve by Siddall et al. is reconstructed using oxygen isotope records from Red Sea sediment cores.

**2 Geological setting of Crete and vertical tectonics**
Lines 11-16: see also Zachariasse et al., 2008. *Reference included now.*

Line 18-21: *Using dated paleoshorelines and numerical models, it is shown that the island of Crete experienced, during the last 20 thousand years, periods of severe uplift (at rates of up to 8 mm/yr) while in the preceding ~30 thousand years, the vertical deformation on Crete was minimal (Mouslopoulou et al., 2015b).* Please add also Tiberti et al. (2014; already cited in other parts of the text) to Mouslopoulou et al. (2015b), as they state "*Attaining the S4 to S5 vertical separation thus requires a net subsidence rate of 2.6–3.2 mm/y in the period from ~42 to 23 ky ago. A period of sustained uplift of ~7.7 mm/y should have then followed the S5 abandonment (~23 ky ago) as suggested by the formation of S2, with sea level at -120 m, and S3 and S1-low with sea level at about the same elevation as today*". *Yes, we have added reference to this work now in the revised version – in the following paragraph though, where we talk about late Quaternary vertical tectonics. Thank you.*

Line 24: please add also Tiberti et al. (2014; same reason explained above). *[Yes, see above].*

Line 28: Strasser et al., *2011* instead of *2010 [Thanks].*

Lines 28-30: Not only historical accounts, however, about the tsunami: see, for instance, Polonia et al., 2013. *[Reference included now, thank you].*

**3.2 OSL dating of alluvial fans**

Line 21: *The results of OSL analysis are presented in Table 1 and Figure 7*. Please specify exactly which kind of analysis: both Table 1 and Figure 7 seem to show only IRSL results. *[Yes, see above]*.

**3.2.2 OSL results**
Page 8, Line 15: *IRSL* instead of *OSL* *[Yes]*.

**3.3 Soil development**
Line 33: *IRSL* instead of *OSL* *[Yes]*.

Page 9, Line 14: *IRSL* instead of *OSL* *[Yes]*.

**4 Landscape evolution at Domata**
Line 18: *international sea-level curve*: please remove "international" *We have removed "international" and replaced it with "the Siddall et al. (2003) sea level curve". Our reasons for using the Siddall et al. curve are explained in our response to the Gallen & Wegmann's review.*
Page 10, Line 31: *deposition of the upper and lower-fan deposits*: please remove "deposits" *[Yes, done]*.

**5 The importance of tectonic uplift at Domata**
Lines 24-30: for the sake of completeness, you should mention the other average uplift rates estimates over the last 50 ky based on quantitative datings along the SW coast of Crete:
1.5 mm/y by Wegmann (2008)
2 mm/yr by Shaw et al. (2008)
1-1.5 mm/y by Strasser et al. (2011)
2.5-2.7 mm/y by Tiberti et al. (2014)
*But…. we do say that our results (ca. 2.2 mm/yr) are in very good agreement with the published results above (and we provide the references). Now we have also provided a range of uplift rate values for these studies.*

Page 12, Lines 5-6: *no significant uplift was accommodated on Crete as the region between ca. 20-45 kyr was experiencing a tectonically quiet period (Mouslopoulou et al., 2015b).* Please notice that for the same period (42 to 23 ky ago), Tiberti et al. (2014) postulated a net subsidence rate of 2.6-3.2 mm/y. *[Yes, we have included a statement including reference to your findings - thanks].*

Page 12, Lines 14-15: *Comparable uplift rates have been independently recorded at numerous localities on western and eastern Crete for the last 20,000 years by Shaw et al. (2008), Tiberti et al. (2014) and Mouslopoulou et al. (2015b).* Shaw et al. (2008) never mention an uplift rate of 7 mm/y or similar values over the last 20 ky. They estimate a ca. 2 mm/y uplift rate on the basis of a 20-24 m elevated shoreline dated 41-53 ka. *[For Shaw et al.: we mean the rate over the last ~2 kyr - since the 365 AD event)].*

**6 Conclusions**
Line 23: *IRSL* instead of *OSL* *[Yes]*.

**Figure 1**
Please add a coordinates reference frame. Use bold for numbers indicating GPS values. Enlarge letters indicating the sites on the southern coast of Crete and use a darker color (e.g. blue instead of yellow) for the circles. In the caption: *WM* instead of *WG* = White Mountains.

**Figure 6**
Caption, Line 5: *100 m* instead of *100's metres* – *[No, it is 100's as the Domata beach is more than 500m long]*

**Figure 7**
Caption, Line 11: *IRSL* instead of *OSL*. Please change this also in the picture. *[Yes]*.

**Figure9**
Caption, Line 21: *IRSL* instead of *OSL*. Please change this also in the picture. *[Yes]*.

**References cited in this review**
Mouslopoulou, V., Nicol, A., Begg, J., Oncken, O., and Moreno, M. (2015) Clusters of mega-earthquakes on upper plate faults control the Eastern Mediterranean hazard, Geophys. Res. Lett., 42, 10282–10289.

Peterek, A., Beneke, K., Schwarze, J. & Spinn, A. (2003) Küste und Küstenformung in Westkreta als

Spiegelbild eustaticher, tektonischer und gravitativ-tektonischer Prozesse. Essener Geog. Arb. 35, 39–56.

Polonia, A. et al. (2013) Mediterranean megaturbidite triggered by the AD 365 Crete earthquake and tsunami. Sci. Rep. 3, 1285, doi: 10.1038/srep01285.

Pope, R., Wilkinson. K., Skourtsos, E., Triantaphyllou, M., and Ferrier, G. (2008) Clarifying stages of alluvial-fanevolution along the Sfakian piedmont, southern Crete: New evidence from analysis of post-incisive soils and OSL dating, Geomorphology, 94, 206-225.

Pope, R.J.J., Candy, I., and Skourtsos, E. (2016) A chronology of alluvial fan response to Late Quaternary sea level and climate change, Crete: Quaternary Research, v. 86, p. 170-183.

Shaw, B. et al. (2008) Eastern Mediterranean tectonics and tsunami hazard inferred from the AD 365 earthquake. Nature Geosci. 1, 268–276, doi: 10.1038/ngeo151.

Siddall, M., Rohling, E. J., Almogi-Labin, A., Hemleben, Ch., Meischner, D., Schmelzer, I., and Smeed, D. A. (2003) Sea-level fluctuations during the last glacial cycle, Nature, 423, 853–858.
Strasser, T. F. et al. (2011) Dating Palaeolithic sites in southwestern Crete, Greece. J. Quaternary Sci. 26, 553–560, doi: 10.1002/jqs.1482.

Tiberti, M.M., Basili, R. & Vannoli, P. (2014) Ups and downs in western Crete (Hellenic subduction zone). Sci. Rep. 4, 5677; DOI:10.1038/srep05677.

Wegmann, K. W. Tectonic geomorphology above Mediterranean subduction zones: northeastern Apennines of Italy and Crete, Greece. (2008) Ph.D. Thesis, Bethlehem, Pennsylvania, Lehigh University, 169 p.

Zachariasse, W. J., van Hinsbergen,D. J. J. & Fortuin,A. R. (2008) Mass wasting and uplift on Crete and Karpathos during the early Pliocene related to initiation of south Aegean left-lateral, strike-slip tectonics. Geol. Soc. Am. Bull. 120, 976–993, doi:10.1130/b26175.1

*Thank you!*

*Sincerely,*

*V. Mouslopoulou and co-authors*

---

## Author Comment (AC3) · 18 Feb 2017

**Response to the new short-comment by S. Gallen & K. Wegmann**

Below, we respond again briefly to the additional comments of Gallen. For a more detailed answer, please refer to our first response, dated January 27th, 2017.

1. **Lack of sedimentologic and stratigraphic context**: This is not true. The stratigraphy, and the stratigraphic context of the major geomorphic features described in this work, is the major tool that we use to support our main observations. We copy and paste here text from our first response: *'Our primary interest in our work on Crete is to help constrain its late Quaternary tectonic development. With this work we aim to use first order geomorphic features and stratigraphic relationships to understand the late Quaternary vertical deformation of western Crete. In doing so, we recognised the importance of the Domata fan sequence and the important marine bench cut in upper fan at Domata. This feature allows us to independently derive a Late Quaternary uplift rate from this section of western Crete. The paper is important because of this independence from previously derived uplift rates on western Crete (e.g., Shaw et al., 2008; Strasser et al., 2011; Tiberti et al., 2014; Mouslopoulou et al., 2015 – all references included in the submitted version). To achieve these objectives, we needed to understand the first order fan geomorphology and from it dissect out the sequence of events required in landscape evolution. We derived a basic sequence of events that is demanded by the stratigraphic relationships present at Domata and present these in the paper. We have gone to some effort to place the Domata fan sequence within its stratigraphic and chronologic context so that we fully understand the uplift rate derived. The final piece of the puzzle required to derive our uplift rate was its integration with a high quality sea level curve for the last 125 kyr. We chose Siddall et al.'s (2003) sea level curve because of its precision, relative proximity to the Mediterranean, yet its isolation from the variable tectonic signatures and isostatic problems associated with glacial loading of that region, and for its similarity with the Lisiecki & Raymo (2005) stacked curve. The luminescence dating we undertook simply provides confirmation of the chronological framework for the events that we had deduced with reasonable confidence from other stratigraphic and geomorphic observations. We completely agree with Gallen & Wegmann that our IRSL dating, with its large uncertainties, cannot provide adequate resolution to separate the individual events presented here. Similarly, the soil descriptions provide a supporting understanding for our stratigraphic conclusions, independently confirming that the two fans are built over distinct time-periods. In our revised version we explain more clearly these objectives and the evidence for our interpretation of the sequence of events in our 'landscape evolution' at Domata, so that there can be no misunderstanding'.*

   And we continue (regarding the sedimentological context): *'This objective does not include a comprehensive description of the materials of the Domata fans and their developmental chronology, as has been undertaken for other fans of southern Crete by others (e.g., Nemec & Postma, 1993; Gallen et al., 2014; Pope et al., 2008, 2016). Neither is the objective to compare Domata fan stratigraphy with that of other alluvial fan systems on Crete. We welcome the prospect of additional work that would enhance insight into the processes of fan deposition and a refined chronology that may help better understand relationships between sediment transfer rates and climate in this area.'*

2. **The beach underlying the fan sequence and the 365AD paleo-shoreline**: We repeat that we do not agree with the correlations proposed by Gallen and Wegmann between the surface on the beach deposits, the "wave cut bench" and the AD365 bioerosional notch. The bedrock "wave-cut bench" underlying fan deposits at the west-end of Domata beach shown

in their supplementary figure may locally be at the same elevation as the AD365 notch, but there is no genetic linkage between that bioerosional notch and the "wave-cut bench". At the west-end of the beach, east of the red area highlighted in Gallen & Wegmann's supplementary figure (c), fan deposits sit on a deeply dissected and weathered surface that lies above the AD365 bioerosional notch. This relationship is illustrated and confirmed in the Figure that we provide below, in this response letter, and which we intend to include in the revised version of the manuscript as Figure 6c. If the deposits at the east-end of the beach indeed represent beach deposits (we have limited data on these materials), they may represent shoreface deposits of the marine trimming event. This information is entirely consistent with our interpretation as it stands.

3. **The age of the Domata fan sequence**: Gallen and Wegmann's interpretations are based largely on their correlation of the beach deposits with the AD365 bioerosional notch and their inference that the fan deposits are therefore Holocene in age. Our response to this correlation is discussed above, and our interpretation is corroborated independently by the relative scales of the Sfakia fan and its catchment and those of the Klados catchment and the Domata fans. Despite the contrasting catchment sizes, the area covered by Holocene materials at the Sfakia fan is insignificant compared with the size of the Domata fan deposits. The Sfakia fan Holocene materials rise to a maximum elevation of c. 40 m, while those of the Domata fan rise to close to 100 m. Even under the most extreme local conditions, this incongruence points to an age older than Holocene for the Domata fan deposits. In addition, interpretation of the Domata materials as Holocene in age would contradict our geochronological data. In addition, and further reinforcing our interpretation, the resultant uplift rates conform with previously published rates (including Gallen's) derived independently in areas nearby and on central-eastern Crete. These three strands of independent data, each support our interpretation of a last glacial age for the Domata fan deposits. By far the most rational option is to accept the age constraints provided by the geochronological data, as we do in our paper.

4. **Soil development:** We never stated that we did not perform a proper description of the soils. This statement is from the two commentators who obviously consider macroscopic observations 'improper'. In our previous response, we put significant effort to address all the soil-related comments posed by the two commentators. And while we state openly, in both the submitted manuscript and our response to their comments, that our observations are mainly macroscopic, these macroscopic observations completely support our independent conclusion (based on geomorphic evidence) that the soil in the upper fan is better developed, and thus more mature, compared to the soil of the lower fan. How these macroscopic observations support this conclusion, is explained in our previous response to Gallen & Wegmann and we encourage whoever interested to look at pages 8-9-10 of the uploaded file (27 Jan 2017). It is also explained in the corresponding section (3.3) of the submitted and revised version.

To summarise, we agree that there is additional work that could and should be done at Domata to help answer Gallen & Wegmann's questions and others as well. In this paper, we are reporting specifically on an investigation that provided interesting results that constrain the late Quaternary vertical deformation of western Crete. In our view, and in the view of the other reviewers, this manuscript achieves this objective.

[Figure]

Annotated image illustrating the relationship between the unconformity at the base of the fan sequence and the AD365 bioerosional notch. This picture is taken between the location illustrated by Gallen & Wegmann's (first comment) in his supplementary figure (c) and the mouth of the Klados River. Our interpretation of the stratigraphy is as follows: a dissected erosional surface on older Quaternary sediments pre-dates the fan deposits, while the AD365 bioerosional notch post-dates both the surface and the fan deposits that rest on it. In other words, the coincidence of the "wave-cut bench" and the AD365 bioerosional notch which Gallen & Wegmann illustrate in their supplementary figures cannot mean that the entire Domata fan deposits are Holocene in age. Here it is evident that the unconformity does not represent the same feature as the AD365 bioerosional notch. This Figure will be included in our revised manuscript as Figure 6c.

---

## Author Comment (AC4) · 19 Feb 2017

**Response to the Reviewer**

Ms. Ref. No.: esurf-2016-62

Title: Distinct phases of eustatism and tectonics control the late Quaternary landscape evolution at the southern coastline of Crete

*In the following paragraphs, we carefully address each of the Reviewer's comments and suggestions (which they appear here in* black*). Our responses are in 'red italics'.*

*As the Reviewer did not refer to specific line numbers, we present below answers that correspond to each of his comments. All our changes in the revised version submitted here are tracked and reflect our effort to satisfy the Reviewer of this manuscript as well as the compilers of the short-comments submitted during the online interactive discussion.*

*We thank the Reviewer for his overall positive comments and constructive criticism. From our understanding, his/her main concern is the unsatisfactory results of the luminescence dating. We totally agree, as we clearly state in the manuscript, that the luminescence results have large uncertainties and are, therefore, unsatisfactory. However, our sequence of events presented in this manuscript is not based on the luminescence dating. It is based on, and demanded by, the stratigraphy and the stratigraphic relationships that we recorded at Domata.*

*Before we go addressing in some detail this and other secondary remarks made by the Reviewer, we would like to make a general statement about the objectives of our work and constrain its scope. This may help allay some concerns expressed by the Reviewer.*

*Our primary interest in our work on Crete is to help constrain its late Quaternary tectonic development. With this work we aim to use first-order geomorphic features and stratigraphic relationships to understand the late Quaternary vertical deformation of western Crete. In doing so, we recognised the importance of the Domata fan sequence and the important marine bench cut in upper-fan at Domata. This feature allows us to independently derive a late Quaternary uplift rate from this section of western Crete. The paper is important because of this independence from previously derived uplift rates on western Crete (e.g., Shaw et al., 2008; Strasser et al., 2011; Tiberti et al., 2014; Mouslopoulou et al., 2015 – all references included in the submitted version).*

*To achieve these objectives, we needed to understand the first order fan geomorphology and from it dissect out the sequence of events required in landscape evolution. **We derived a basic sequence of events that is demanded by the stratigraphic relationships present at Domata and present these in the manuscript.** We have gone to some effort to place the Domata fan sequence within its stratigraphic and chronologic context so that we fully understand the uplift rate derived. The final piece of the puzzle required to derive our uplift rate was its integration with a high quality sea level curve for the last 125 kyr. We chose Siddall et al.'s (2003) sea level curve because of its precision, relative proximity to the Mediterranean, yet its isolation from the variable tectonic signatures and isostatic problems associated with glacial loading of that region, and for its similarity with the Lisiecki & Raymo (2005) stacked curve.*

*The luminescence dating we undertook simply provides confirmation of the chronological framework for the events that we had previously deduced with reasonable confidence from other stratigraphic and geomorphic observations. We completely agree with the reviewer that our luminescence dating, with its large uncertainties, cannot provide adequate resolution to separate the individual events presented here. The landscape evolution proposed in Figure 10 is not based on the luminescence*

*dating: it is based on the* **sequence of events** *that is demanded by the stratigraphic relationships present at Domata. The luminescence dating comes only to provide the chronologic frame (MIS 3) within which this sequence of events took place.*

*For detailed discussion on the major issues relating to the IRSL dating we kindly direct the Reviewer to Pages 3-4 of our response to the short comment of Gallen and Wegmann (dated 27 Jan 2017) during our online interactive discussion.*

*In the revised version we have put some effort to:*
*- rephrase the scope of this paper,*
*- more clearly describe the main geomorphic features observed and*
*- more carefully present their interrelations*

*by:*
- *Including additional text (see tracked changes throughout our manuscript),*
- *Modifying our former Figure 5 to include schematic sketch of the volumes of the upper and lower fan materials and the stratigraphic position of each of our IRSL samples within these volumes.*
- *Re-annotating/modifying existing figures (Fig. 6b) or introducing new figures (Fig. 6c).*
- *Including discussion on the important work of Pope et al. (2016), that was missing from our originally submitted version as its publication post-dated our work.*
- *Updating our reference list.*

*In addition, we have modified accordingly the revised version to accommodate all other specific comments of the Reviewer as well as comments from the other two commentators during the online interactive discussion.*

**General Comments**

The Mouslopoulou et al. paper deals with a topic that has both environmental and tectonic implications, focusing on a well-expressed sea-side alluvial fan. Compared to the other fans that make up the coastal bajada of south Crete that develops ca 20 km further east, Domata fan is a relatively small one (area c. 0.2 km2), along with a string of other isolated fans that border the steep southern Cretan coast. Nonetheless, its stepped morphology, with an escarpment running roughly parallel to the present-day coastline may be indicative of processes that significantly affect fan-shape evolution, as the 'marine trimming' suggested by the authors. Indeed, Mouslopoulou et al. have tried to link fan evolution to climatic changes, coupled with uplift rate scenarios for the past few kyr and come up with a scenario that ties the evolutionary stages of the fan to successive marine high- and lowstands.

To achieve such a target, one has to resort to high-res geochronology, something that is not always feasible. Her lies, in my opinion, the main drawback of Mouslopoulou et al.'s work, which is found in their sampling strategy and the resulting dating accuracy: while the latter may not be the authors' fault, the former is a weak point in this work. Sampling did not include the body of the 'lower fan': this might clarify –and probably strengthen the authors' distinction of the Domata fan in an 'upper' and a 'lower' one. (Of course, this might not help, either, as the results from the other samples contain significant errors).

Having said these, I acknowledge the fact that the number of dated samples (and possibly the range of dating methods applied) are dictated by research funding; nonetheless, one has to make do with what they have, I'm afraid. At any rate, the resulting OSL ages contain significant errors (as also

acknowledged by the authors: P.8, l. 14); it is also unclear whether standard deviation and standard error refer to 1-σ or 2-σ *(they refer to 1σ, as it is stated in Figure 9).* Hence, temporal resolution is too poor to support such a detailed evolution scenario, whose resolution is as high as 1kyr, while most ages overlap significantly and the correlation of successive events to KDE$_{max}$ is not satisfactorily constrained. *For this comment, please see our introductory statement above.*

The lithostratigraphy of the Domata fan is poorly described; lithostratigraphic logs are missing – and these could help place the obtained samples in a coherent geological context. Moreover, it might clarify any probable lithological or other difference between the units described as "upper" and "lower" fan. This could also be aided if appropriate figures (esp. photographs) were included, to show the lithological composition of the fan(s). Panoramic photos are fine, but some close-ups would be very useful for the reader to understand the distinction made between Upper and Lower Fans. Figure 8 (b and c) focus on the soil cover and do not serve this purpose.

*The above stated objectives do not include a comprehensive description of the materials of the Domata fans (lithostratigraphic logs) and their developmental chronology, as has been undertaken for other fans of southern Crete by others (e.g., Nemec & Postma, 1993; Gallen et al., 2014; Pope et al., 2008, 2016). Nevertheless, we have rephrased the text in the revised version of our manuscript to better describe these geomorphic features and their interrelations. Besides, to better illustrate the first-order stratigraphic relationships between the main geomorphic features, we have now modified Figure 5 by illustrating schematically the volumes beneath the measured profiles to indicate the likely extent of the volumes of upper fan and lower fan materials. We have also added the locations of each of the luminescence samples. We have further followed the Reviewer's advice and we have modified Figure 6b to better reflect the relationships between lower and upper fan units. In addition, we have also included a close-up image (Fig. 6c) to better reflect the relationship between the unconformity at the base of the fan sequence and the AD365 bioerosional notch.*

Between Domata and Sougia (c. 9 km to the west) there is a number of fans in practically the same a geomorphological and geological environment (the same could also be supported for the bajada the east, with Sfakia fan being its westernmost member). Processes suggested in this paper (i.e. "marine trimming") are not localized ones and affect extended tracts of land. So if such a process was responsible for the modification of the Domata fan, why it is not found elsewhere along this coast?
*It is true that along the southwestern coastline of Crete (e.g., east and west of Domata), there is a number of alluvial fans in roughly the same geomorphological setting. Many of these fans are indeed clearly truncated by the sea (Figure A below, near Trypiti, west of Domata). Although the 'double trimmed' alluvial fan-system at Domata is unique in its kind (as each episode of the two alluvial fan-building episodes has been followed by episodes of alluvial incision and subsequent marine trimming), evidence for 'double marine trimmings' exists elsewhere on western Crete as well. In Figure B below, for example, we provide evidence for a similar geomorphic feature that occurs at Agia Roumeli (about 4 km east of Domata), where an alluvial fan-system is truncated by two distinct marine-trimming events. It is hard to argue that these cliffs (illustrated in Photo B) are not trimmed by marine processes as they are parallel to the coast and on steeply sloping fans. Besides, on the upper cliff, the one with the sea-caves (where the hiking-track from Domata to Agia Roumeli passes through), we have found beachrock with shell-hash.*

[Figure]

*Figure A: Marine-trimmed alluvial fan near Trypiti (west of Domata)*

[Figure]

*Figure B: Double marine trimming of alluvial fan-system at Agia Roumeli (east of Domata).*

The authors did not take into account the work by Pope et al., (2016), on the nearby, wellstudied Sfakia fan. This may be due to the fact that the m/s postdated the publication date of this paper, but nonetheless, the authors should take it into consideration in their revised version.

*That is correct. When our research was compiled the work of Pope et al. (2016) was not yet published; this is why we did not include it in the submitted version. In the revised version, not only have we included reference to Pope et al. but we have added the following extensive comment relating our main findings to those of Pope et al. (2016):*

*'This work was undertaken prior to the publication of the latest results of OSL and U series dating of the Sfakia fan sequence (Pope et al., 2016). Intriguing conclusions of their high resolution dating work include that at Sfakia, three sometimes overlapping phases of fan deposition since the last interglacial are separated by two phases of fan entrenchment, the first close to the MIS 5/4 (c. 70 kyr) boundary, the other close to the MIS 2/1 boundary (c. 14 kyr), triggered by major climatic changes. Fan deposition there has to a large degree persisted through stadial and interstadial periods during the last 125 kyr. Periods of entrenchment at Sfakia do not appear to correlate with the two entrenchment periods at Domata. The Sfakia fan is significantly different from the Domata fan in catchment size (c. 28 $km^2$ compared with c. 11 $km^2$), fan size (5.3 $km^2$ compared with 0.1 $km^2$), the presence of more than one feeder channel at Sfakia, and in the nature of deposits (primarily clast-supported gravels compared with primarily matrix-supported gravels). Whether these differences are responsible for differences in depositional and entrenchment histories and in preservation of marine cliffs at Sfakia, or differences in local climatic regimes or vegetation changes is uncertain. However, one compatible conclusion of their work with our own is recognition of the importance of base level (sea level) change to the process of entrenchment.'*

**Specific Comments**

Page 4, l. 20-21. Bedrock geology is grossly misrepresented, both in terms of lithology and age. Klados Gorge runs through platy crystalline limestones with phyllite intercalations and chertbearing dolomites, chert-nodule-bearing limestones and quartzitic sandstones and shales, not through the platform carbonates of the Tripolis Unit (which is a Mesozoic carbonate platform). Moreover, the aforementioned lithologies (i.e. Kingilos group) belong to the metamorphosed Plattenkalk Unit, also known as Mani Unit. (e.g. Creutzburg and Siedel, 1975; Fassulas et al., 1994; and Jolivet at al., 1996). This suffices to explain the occurrence of quartz detritus in the Domata fan; the authors, however, did not seem to wonder why a purely carbonate-fed fan (as they describe it), contains so much quartz! *We apologise for this – it slipped out of our attention and the correct rock-type is now included in the revised version. We completely agree with the Reviewer that the Plattenkalk unit outcrops in our study area. Specifically, the platy limestones and the platy limestones with cherts are the main units that occupy the largest part of the Klados gorge valley. The Gigilos layers appear in the northern part of the Klados watershed, which drains mainly towards the Omalos plateau and the northern part of the Samaria Gorge (Fassoulas et al., 2004). However, we doubt that the Gigilos beds (shales and quarzitic sandstones and chert bearing limestones) have been contributing in the mass of sediments within the Klados gorge since they are only outcropping in the northern parts of White Mountains (Lefka Ori) and they are north dipping. In contrast, the Klados gorge developed in the overlying platy limestones with cherts which eventually developed in platy marbles with limited chert intercalations (Unit 4d and 4e at Manutsoglu et al., 2003). Thus, Klados gorge has mainly developed in platy limestones with some cherts and platy marbles and the erosion of these units supplied the Domata area mainly with carbonate clasts and limited chert clasts (from the lower units) which explain the abundance of carbonates in the alluvial fans and fluvial terraces.*

1. *Manutsoglu, E., Soujon, A., and Jacobshagen, V., 2003. Tectonic structure and fabric development of the Plattenkalk unit around the Samaria gorge, Western Crete, Greece. Z. Dtsch. Geol. Ges. 154, 85-100.*
2. *Fassoulas, C., Rahl, J.M., Ague, J., and Henderson K., 2004. Patterns and conditions of deformation in the Plattenkalk nappe, Crete, Greece: a preliminary study. Bull. Geol. Soc. Greece, vol. XXXVI, 1626-1635.*

P.3, l.20. Please use appropriate term instead of "vertical deformation".
*We don't understand what the reviewer means. We will change the 'vertical deformation' to 'vertical movement', hoping that this is more appropriate.*

P.21, Figure 2. How confident are the authors that the quasi-planar landforms west of the gorge are marine benches? Is there any piece of evidence supporting this suggestion?
*These quasi-planar surfaces are very common in southwest Crete and are generally well-studied. As we state in the manuscript, 'while we cannot assign ages to these benches, their altitude and geomorphic similarity with known and dated (MIS5) late Pleistocene marine benches elsewhere in Crete (e.g., Strasser et al., 2011; Gallen et al., 2014; Strobl et al., 2014), provides some stratigraphic and chronologic context for the age of the alluvial fans at Domata (i.e., because of their lower elevation, the alluvial fan surfaces that are subject of this paper are expected to be younger than 125 kyr)'.*

*Moreover, marine materials (shell-hash beachrock) have been found by the authors on such a planar surface of similar (~100 m) elevation, 2.5 km west of Domata (towards Sougia). However, we did not date these materials as we expect their age to be outside the radiocarbon age-range.*

P25, Figure 6b. The red dashed line that is suggested to represent a low terrace riser on the west side of the river seems rather ambiguous; it is hard to say from this photo.
*We have now modified and re-annotated 6b to satisfy the Reviewer.*

**Technical Corrections**
N/A

**References cited**

Creutzburg N., Siedel. E., (1975). Zum stand der Geologie des präneogens auf Kreta. N. Jb. Geol. Paläont. Abh. 149(3): 363-383.

Fassulas, Ch., Kilias, A., Mountrakis, D., (1994). Post-nappe stacking extension and exhumation of the HP/LT rocks in the island of Crete, Greece. Tectonics 13: 127-138.

Jolivet, L., Goffe B., Monie P., Truffert C., Patriat, M., Bonneau M. (1996). Miocene detachment in Crete and exhumation P-T-t paths of high pressure metamorphic rocks. Tectonics 15: 1129-1153.

Pope, R.J.J., Candy, I., Skourtsos, E., (2016). A chronology of alluvial fan response to Late Quaternary sea level and climate change, Crete. Quat. Research 86: 170-183.

---

## Editor Comment (EC1) · V. Vanacker (Editor) · 24 Feb 2017

We now have three detailed reviews back on the manuscript, that provide valid and very useful suggestions for the improvement of the final document. The revised version of the paper needs to address all comments raised by the reviewers, and I would like to suggest you to take the following four elements along in your revisions :

1/ The conclusions should be robust and based on facts and data. In the paper, the authors conclude (p14, L 1-7) that "sea-level fluctuations in response to varying climatic conditions formed the landscape at Domata during MIS 3 (∼57-29 kyr BP). It is, however, because of the fast tectonic uplift that Crete experienced during the subsequent ∼20 thousand years that the entire alluvial sequence escaped destruction and/or mod-

ification due to marine inundation and is preserved sub-aerially today". As tectonism, sea level variation and paleo-climate have regional impact, the observations at Domata cannot be isolated from other studies on nearby alluvial fans. A discussion of the findings from the Domata fan, in the light of previous work on alluvial fans in Crete is essential (see comments raised by reviewer#1, reviewer#2). If there exist different hypotheses about the Quaternary evolution of the region, try to give the different models of evolution and discuss alternative interpretations based on facts and data.

2/ The text misses some consistency in the interpretations as outlined by the reviewers. To give one example : p4, Line 5-10 : the interpretation of the tilted paleoshoreline. The authors state that..." A number of studies have constrained the timing of this prominent paleoshoreline at ∼1.5-2 kyr BP, with some attributing it to the AD 365 historic earthquake". However, we see on p6, L24-30 : "bioerosion notch indicating an uplifted paleoshoreline at ∼6 m "..." seismically uplifted paleoshoreline dated at ∼365 AD"..." the AD 365 bioerosional notch " . Also in Figure 6, the authors only refer to the "365 AD paleo-shoreline" not mentioning alternative explanations (Pirazzoli et al., 1982; 1996; Stiros, 2001; Shaw et al., 2008).

3/ As the chronological framework of the sedimentary events is not well constrained (5 samples, with four samples giving a broad time interval of ∼ 30 to 60 kyr BP, Figure 9), one needs to be very careful with the interpretation of landscape evolution (section 4, and Figure 10). Supplementary information (either from soil chronosequences, or lithostratigraphy) would be very helpful. Two reviewers have recommended to include lithostratigraphic logs, or detailed stratigraphic and sedimentological evidence of the fan sequences, and I fully support these recommendations. See details in review#1 and 1-bis and review#2.

4/ Carefully revise and edit the figures 5 and 8. Figure 5 is a new figure showing the topography of the fan surfaces. Can you give the orientation of the profiles (or location on Fig 4), and the unit of the X-axis ? Figure 8 shows two soil profiles. One would typically give the soil depth (depth till C horizon, starting at 0 cm at the surface). Also,

what is meant with "parent rock"?

**ESurfD**

---

## Author Comment (AC1)

**Response to the short comment of S. Gallen and K. Wegmann**

*We take this opportunity to thank S. Gallen and K. Wegmann for their comments on our manuscript.*

*Before we respond in detail to their comments, we would like to make a general statement about the objectives of our work and constrain its scope. This may help allay many of the concerns they expressed.*

**Objectives:**
*Our primary interest in our work on Crete is to help constrain its late Quaternary tectonic development. With this work we aim to use first order geomorphic features and stratigraphic relationships to understand the late Quaternary vertical deformation of western Crete. In doing so, we recognised the importance of the Domata fan sequence and the important marine bench cut in upper fan at Domata. This feature allows us to independently derive a Late Quaternary uplift rate from this section of western Crete. The paper is important because of this independence from previously derived uplift rates on western Crete (e.g., Shaw et al., 2008; Strasser et al., 2011; Tiberti et al., 2014; Mouslopoulou et al., 2015 – all references included in the submitted version).*

*This objective does not include a comprehensive description of the materials of the Domata fans and their developmental chronology, as has been undertaken for other fans of southern Crete by others (e.g., Nemec & Postma, 1993; Gallen et al., 2014; Pope et al., 2008, 2016). Neither is the objective to compare Domata fan stratigraphy with that of other alluvial fan systems on Crete. We welcome the prospect of additional work that would enhance insight into the processes of fan deposition and a refined chronology that may help better understand relationships between sediment transfer rates and climate in this area.*

**Methods:**
*To achieve these objectives, we needed to understand the first order fan geomorphology and from it dissect out the sequence of events required in landscape evolution. We derived a basic sequence of events that was demanded by the stratigraphic relationships present at Domata and present these in the paper. We have gone to some effort to place the Domata fan sequence within its stratigraphic and chronologic context so that we fully understand the uplift rate derived. The final piece of the puzzle required to derive our uplift rate was its integration with a high quality sea level curve for the last 125 kyr. We chose Siddall et al.'s (2003) sea level curve because of its precision, relative proximity to the Mediterranean, yet its isolation from the variable tectonic signatures and isostatic problems associated with glacial loading of that region, and for its similarity with the Lisiecki & Raymo (2005) stacked curve.*

*The luminescence dating we undertook simply provides confirmation of the chronological framework for the events that we had deduced with reasonable confidence from other stratigraphic and geomorphic observations. We completely agree with Gallen & Wegmann that our IRSL dating, with its large uncertainties, cannot provide adequate resolution to separate the individual events presented here. Similarly, the soil descriptions provide a supporting understanding for our stratigraphic conclusions, independently confirming that the two fans are built over distinct time-periods.*

*In our revised version we explain more clearly these objectives and the evidence for our interpretation of the sequence of events in our 'landscape evolution' at Domata, so that there can be no misunderstanding.*

**Our detailed responses are presented below in blue.**

**(1) Lack of a sedimentologic and stratigraphic descriptions of the Domata fan in the context of other alluvial fans on Crete in the current version of the manuscript.**

The Klados River gorge itself is not unique, but one of five similar gorges *[Actually, the Klados R. gorge is significantly shorter and its catchment size is much less than the other gorges of the Lefka Ori (c. 11 km$^2$ compared with 28 km$^2$ for the Sfakia fan).The size of the Domata fan sequence is also very small (0.1 km$^2$ vs 5.3 km$^2$ for Sfakia). Therefore, they are not necessarily similar.]* that drain the Lefka Ori (White Mountains) and there is little difference in the coastal geomorphology at the mouths of each of these gorges *[We disagree with this statement either. Please compare the geomorphology at the mouths of Tripiti, Klados, Samaria and Eligia gorges. They are very different. And the entrenched Aradaina Gorge at Marmara, with its marine benches is*

*very different to the alluvial fan systems that extend from Hora Sfakion to Skaloti. There are significant differences in the coastal geomorphology at the mouth of each of these gorges. In addition, V.M., J.B. and D.M. who have walked down all five of them through the years, can confirm that these gorges also differ internally – that was also reflected in the significantly different degree of effort/difficulty to cross them].* However, having visited this fan sequence and studied numerous other fans on Crete, we can say that the Klados (Domata) fan sequence is sedimentologically and stratigraphically unique among fan deposits on Crete *[Please see the statement preceding this section. Our paper is not focussed on this issue. The authors of this short comment have made these observations and have the opportunity to write such a paper making this comparison.]* Most alluvial fans in Crete are coarse grained, clast supported, and weakly stratified. By contrast the Domata fan sequence is finer grained, having horizons that are variably clast and matrix supported, and better stratified than any other fan that we have seen on the island to date. It is these sedimentological details that will provide the most insight into the origins of the fan unit *[We believe that this is a value judgement. Our paper does not alter the opportunity for these reviewers to prove this point in their own paper].* The unique sedimentology and stratigraphy of this fan relative to other fans in Crete suggests that a different process is responsible for the deposition of this fan sequence *[Presumably to be included in their paper].* Given the short transport distance through the Klados River gorge (~ 5 km) *[transport distance is not necessarily a significant factor in the energy of a depositional environment, a high base level is]* and the relatively fine grained nature of the fan deposit, the Domata fan sequence is suggestive of a high-energy process *[Our interpretation differs – we believe the depositional environment is lower energy, influenced by high sea-level].* Stratigraphic and sedimentological descriptions of the fan sequence and discussion of how these observations compare with other studies of fans on Crete would substantially improve the manuscript *[As explained above, these descriptions and comparisons can still be made by these reviewers or by others; they are not the subject of this paper. See our objectives statement above. Further, it is inappropriate to suggest modification of our manuscript on the basis of their interpretation of their data, when we don't even have these data].* Based on the data presented current version of the manuscript it is difficult to evaluate whether or not the authors' argument that the fan sequence represents two fan *[We have now included a specific sentence stating unequivocally that the relict fan sequence at Domata represents two distinct phases of fan deposition within a single channel feeder fan. We have also added a sentence clarifying that both fans are sourced from the entrenched Klados River gorge, the feeder channel].* An alternative interpretation is that the Domata fan sequence represents a single depositional phase followed by unsteady incision, as is common in alluvial fill-cut terrace sequences (also known as complex-response fill terraces of Bull, 1990) *[The presence of a marine cliff at the shoreward side of the upper fan that also truncates an alluvial entrenchment of the upper fan surface indicates clearly that the depositional phase present during upper fan deposition is interrupted, and pre-dates deposition of the lower fan].*

Importantly, our own observations indicate that the entire Domata fan sequence overlies a beach deposit that is the lateral continuation of the Holocene bioerosional notch (see figure below). If correct, such a stratigraphic relationship demands that the Domata fan sequence is Holocene, rather than Pleistocene, which is in direct challenge to the geochronology presented in this manuscript. *[We firmly believe that this interpretation is seriously flawed. The "bioerosional notch" in their figure (c), with its interpretation clearly points to the 6 m a.s.l. AD 365 earthquake bioerosional notch that we can follow westward all the way to Sougia and beyond to Palaiochora and Elafonisi and from there around the corner all the way to Falasarna. At the point indicated in their figure, the elevation of this notch coincides with the elevation of the contact between the lower fan deposits and bedrock. The bench indicated in the supplementary figure has been preferentially eroded because it relict from an earlier erosion surface (upon which the lower fan deposits sit here.) The same bioerosional notch is also present in bedrock immediately to the east of the indicated location, 1 m and more below the bedrock/lower fan deposits contact, the latter represented by an uneven morphology (see our Fig. 6b, newly annotated to illustrate this point and the onlapping relationship between the fan deposits and bedrock). The contention therefore, that beach deposits underlie the lower fan deposits on the basis of the reviewers' illustration, is stratigraphically unfounded. The lower fan deposits must pre-date the bioerosional notch. To interpret this bioerosional notch as pre-dating the lower fan deposits as an older Holocene feature requires the entire lower fan to post-date c. AD365. This bioerosional notch is cut in bedrock, has been eroded from fan gravels (note the rockfall deposits at the base of the cliff in the lower fan behind the main beach, illustrating that the loose*

*gravels of the fan deposits erode very differently to the bedrock), but is further represented by the young, abandoned terraces at the Klados River mouth].*

Furthermore, we are curious as to why luminance dating was only attempted on a fan sequence that is almost entirely comprised of carbonate detritus? Why not also try to date the underlying beach deposit that contains more material suitable for luminance dating and is less likely to suffer from incomplete bleaching?

*[The reason is clear: no "beach deposit" as interpreted by the commentators underlies the fan. Dating the "bioerosional notch" they refer to would provide an incorrect estimation for the age of the fans. The reviewers appear to be suggesting that entire >60 m thickness of these fans was deposited in <10kyr. Compare this thickness to those of other fans in Crete, such as those recorded at Sfakia region or even those recorded in central-eastern Crete by Gallen (2013). In addition, no beach deposit was identified on the marine bench cut in upper fan deposits, presumably because if deposited, it was subsequently eroded early in the deposition of the lower fan].*

**(2) The luminescence geochronology and the lack of tests for, or detailed discussion of the potential for and implications of incomplete bleaching.**

The proper tests needed to confirm or reject whether or not incomplete bleaching has occurred were not reported. Without these key tests of samples from a depositional environment that is notorious for incomplete bleaching (Rhodes et al., 2010), it is difficult to interpret the luminescence data as being a trustworthy chronometer reflecting a true burial age. Every other study that has used luminescence dating to constrain the timing of alluvial fan deposition on Crete has successfully used quartz OSL (Pope et al., 2008; 2015; Gallen et al., 2014; Runnels et al., 2014). The fact that this method did not work for this study is of significance provided that the setting is geologically, tectonically and climatically similar to the locations of all of the aforementioned studies. The only thing that makes the Domata fan sequence unique in its sedimentology and stratigraphy, which suggests that a different process is responsible for its deposition (see comment above). Our suspicion is that the unique origin of the Domata fan sequence is why quartz OSL was unsuccessful. Furthermore, we question the reliability of the feldspar IRSL data without bleaching tests. It is acknowledged in the text that incomplete bleaching can explain the noisy IRSL data (P. 8, Lines 4-5), but the significance of this signal and the proper tests for incomplete bleaching are not present in the manuscript. Provided the unique problems with quartz OSL signals in this deposit, coupled with the known problems of incomplete bleaching in alluvial fans, and results that are difficult to explain, one is hard pressed to interpret this data at face value without prior proper vetting of the luminescence signals.

*We summarise the reviewer's concerns in three main groups:*
*1) Quartz seemed not to have worked at Domata but has worked in other studies nearby on Crete (Pope et al., 2008, 2016; Gallen et al., 2014)*
*2) Feldspar dating at Domata yields imprecise IRSL ages with high uncertainties*
*3) Feldspar IRSL dating was not carried out following standard procedures (no testdose correction during SAR measurement, no fading test to correct for potential age underestimation).*

*1: Why was quartz OSL so dim? For readers of the above mentioned studies it's hard to believe that quartz didn't work at Domata, a few kilometres away from the Sfakia fan system with comparable geological settings. But it is an empirical fact, that the OSL signal intensities were very low and that linear modulation (LM) measurements showed that there was almost no "fast component" in the natural OSL signals. Thus, quartz dating is wrong. A possible explanation is that the dated quartz was not sensitized before sedimentation. Quartz needs repeated cycles of daylight bleaching and radioactive irradiation to emit any analysable OSL signal. Dim quartz not suited for OSL dating is often reported from geologically young environments (e. g. the Alps). But, of course, this does not explain why the behaviour of Domata quartz is different from quartz of other locations on Crete.*

*2: Why are our error in age so large? Our samples consisted almost entirely of carbonate detritus with only few quartz and feldspar grains of a suitable size (exception: UF-2) and the samples were collected mostly near the surface. This implies some problems leading to high age uncertainties: 1) In such inhomogenous and coarse grained samples one can expect radioactive inhomogeneities. 2) Near the surface, any kind of postdepositional mixing is possible (roots, bioturbation, penetration of bleached material into voids). 3) The*

*dominance of carbonate causes an extremely low dose rate of roughly 0.5 Gy/ka for quartz and 1.0 Gy/ka for potassium feldspar. Hence, the contribution of the cosmic dose rate to the total dose rate is relatively high (ca. 40% for quartz, 20% for potassium feldspar). But the cosmic dose rate is only estimated from the sample position and cannot be measured. 4) Incomplete bleaching (insufficient daylight exposure) is always a risk in fluvial environments. The coarse grain sizes suggest runoff events of high energy, where incomplete bleaching is more likely. The studies of Pope et al. (2008, 2016) suffer less from the mentioned problems because they sampled homogenous sandy material from sand layers/sand lenses within the fan body.*

*3: Problems appeared with potassium feldspar IRSL dating? In addition to the problems discussed above (large errors), IRSL dating failed using standard SAR procedures. For instance, it was not possible to recover a known laboratory dose with standard SAR protocols. It was deduced that the testdose correction did not work, but rather caused a systematic underestimation of known laboratory doses. Thus, testdose correction was omitted, but all aliquots affected by sensitivity changes were rejected. Another constraint is the so-called anomalous fading, which affects almost all potassium feldspar samples. Anomalous fading is the unwanted signal loss during burial leading to age underestimation. Often, age underestimation up to 30% of the true age is observed. It is possible to determine fading rates to correct the resulting IRSL ages, but this is very time consuming and there is currently discussion questioning whether this technique is reliable. Nevertheless, such fading tests reveal whether anomalous fading plays an important role or not. Fading correction was not carried out here, because fading is the "antagonist" of incomplete bleaching. Fading causes age underestimation, incomplete bleaching causes age overestimation. It is probable, that our feldspar samples suffer both from fading and incomplete bleaching and it's hard or even impossible to distinguish the two effects.*

*Bleaching tests were not conducted. This is to test, whether the IRSL signal is bleachable in an appropriate time. For this test, aliquots are bleached stepwise in a solar simulator and the remaining signals are measured. This could be easily done. Tests for incomplete bleaching (possible age overestimation) are important and we did it by analysing the OSL and IRSL data (e.g. analysis of age distributions or plotting the signal intensities vs. dose). And yes, **there is evidence for incomplete bleaching**. Unfortunately, the amount of unbleached signal inherited cannot be quantified clearly. For quartz OSL there are statistical models to select the best bleached grain population when measuring single grains or "small aliquots" containing only few grains. These models are less reliable for feldspar.*

**(3) Incomplete review of pertinent literature.**
Much of the literature on 1) Cretan alluvial fans and 2) alternative models for the tectonics of the Hellenic forearc are missing from the manuscript. *[We agree that there is some missing literature, mainly because this work was compiled before Pope et al. (2016) was published. We will include this work in the revised version. Also, we will include discussion on alternative tectonic interpretations, although whether or not there are uplift transients on Crete will not impact at all on the results of this study.]* In addition to the excellent work of Pope et al. (2008), Pope et al. (2016), Runnels et al. (2014), and Gallen et al. (2014) employ luminescence geochronology to date alluvial fans on Crete. Pope et al. (2016) and Runnels et al. (2014) are absent from the current version of the manuscript *[For Pope et al., (2016) the reason is explained above. For Runnels et al. (2014): this publication was published in the journal of European prehistory, and it escaped our attention. We thank the reviewers for pointing this article out].* While Gallen et al. (2014) is cited, no acknowledgement is made for this studies contributions to understanding the Quaternary coastal stratigraphy of Crete *[We have amended the ms to include Gallen et al.'s studies, indicating that they have recognised elevated benches in central-eastern Crete dated as last Interglacial in age. This supports our interpretations on relative age of the Domata fan sequence. Our objective in this ms was to provide a post-Last Interglacial temporal context for the Domata fans].* In addition to successfully dating alluvial fans with quartz OSL, the Gallen et al. (2014) study dates marine terrace deposits with OSL that are buried by alluvial fans. The authors of the above cited studies, and especially Gallen et al. (2014) use detailed mapping, stratigraphy and sedimentology of the deposits, pedology, OSL geochronology and a global sea level curve to derive a model for the coastal stratigraphy in southern Crete that relates interactions between tectonics, climate and eustacy. Discussion of the findings and interpretations presented by Mouslopoulou et al. in the context of other, similar studies from Crete would greatly improve the manuscript *[See our objectives statement above].*

The review of the Quaternary tectonics of Crete is incomplete. In the background section and again in the discussion, alternative models for the Quaternary vertical tectonics of the island are not discussed *[See above]*. Section 2 reads as though consensus has been reached regarding "Late Quaternary uplift transients". However, there is an ongoing scientific debate in the literature about whether or not these Late Quaternary uplift transients actually exist or if there are problems with the geochronology used to derive this model *[The geochronology used in our ms is consistent with the Gallen et al. (2014) OSL geochronology for the Last Interglacial. Furthermore, it contradicts the suggestion that the entire Domata fan sequence is Holocene in age]*. While this may be the favored interpretation of the authors of this manuscript, other interpretations should be acknowledged and the ongoing controversy in the literature noted *[We will make this change in the revised version – but it is unrelated to our conclusions]*.

*To answer to some of the reviewer's concerns, we will add the following text in our revised manuscript:*
*This work was undertaken prior to the publication of the latest results of OSL and U series dating of the Sfakia fan sequence (Pope et al., 2016). Intriguing conclusions of their high resolution dating work include that at Sfakia, three sometimes overlapping phases of fan deposition since the last interglacial are separated by two phases of fan entrenchment, the first close to the MIS 5/4 (c. 70 kyr) boundary, the other close to the MIS 2/1 boundary (c. 14 kyr), triggered by major climatic change. Fan deposition there has to a large degree persisted through stadial and interstadial periods during the last 125 kyr. Periods of entrenchment at Sfakia do not appear to correlate with the two entrenchment periods at Domata. The Sfakia fan is significantly different from the Domata fan in catchment size (c. 28 km² compared with c. 11 km²), fan size (5.3 km² compared with 0.1 km²), the presence of more than one feeder channel at Sfakia, and in the nature of deposits (primarily clast-supported gravels compared with primarily matrix-supported gravels). Whether these differences are responsible for differences in depositional and entrenchment histories and in preservation of marine cliffs at Sfakia, or differences in local climatic regimes or vegetation changes is uncertain. However, one compatible conclusion of their work with our own is recognition of the importance of base level (sea level) change to the process of entrenchment.*

**Line-by-line comments:**

**Introduction:**

P. 2, Line 14-16: There is little mention of sediment supply here. The interplay between sediment supply and discharge is an important factor controlling alluvial fan deposition and may have little to do with changes in base level (e.g. rising sea-level). Furthermore, enhanced rainfall does not necessarily translate into alluvial fan deposition, as implied. Enhanced rainfall may favor increased discharge at the expense of reduced hillslope sediment supply because the hillslopes are vegetated more during times of increased annual precipitation and thus, the alluvial fan experiences an episode of incision. The interplay between climate and tectonics, deposition and incision is not straightforward. We would also like to point the authors to alternative models for channel aggradation that might be relevant to this study. In particular, the recent work of Scherler et al. (2016) documents that Late Pleistocene fill terraces in southern California (a region climatically similar to Crete), which were traditionally interpreted as the result of climate change, are more likely the result of changes in sediment supply due to a large landslide in the catchment. This research is also relevant because they use luminance dating of the alluvial fill and discuss at length the geochronologic problems associated with incomplete bleaching.
*[We have modified these sentences on the revised version to accommodate some of the reviewer's concerns.]*

P. 2, Line 20-24: These types of interpretations are difficult to discern from field data alone as the drivers of aggradation and incision reflect the interplay between sediment supply and discharge. What seems to be implied by this review is that deposition is driven solely by enhanced precipitation and incision by tectonic uplift. Yes, ultimately, the accommodation space needed for alluvial fan deposition is a result of tectonic processes, but at the time scale of the Late Pleistocene, the amount of tectonic uplift is insignificant in comparison to variations in climate-driven discharge and hillslope sediment supply from the mountainous catchment to the alluvial fan system. Furthermore, precipitation, temperature, and thus, vegetation co-vary in ways that make it difficult to predict how changes in precipitation relate to variations in catchment sediment supply and discharge. Depending on the climate and vegetative response, increased precipitation can lead to a reduction in sediment supply and incision, rather than aggradation.

The authors also appear unaware of a critical new body of research by Pope et al. (2016). *[See above explanation].* In this paper, Pope and colleagues 30 present 32 new OSL and U-series dates for what is undoubtedly the best dated alluvial fan sequence on the south coast of Crete, the Sfakia fan. Importantly, Pope et al. (2016) conclude that over the entirety of the late Quaternary, the Sfakia fan only experienced two episodes of entrenchment (incision), during the transition between Marine Isotope Stages (MIS) 5a/4 and MIS 2/1. They propose that the MIS 5a/4 period of fan incision was driven by sea level-induced base level fall; whereas the MIS 2/1 interval of incision (during a time of rapid eustatic sea level rise) was the result of reduced hillslope sediment supply to the fan resulting from landscape stabilization (re-vegetation) during the onset of the current interglacial (Holocene). If their data is correct – and they have lots of reliable geochronology to support their conclusions – the most 5 recent episode of fan incision, for example had little if anything to do with base level fall or tectonic uplift. Pope and colleagues conclude that, with the exception of the above mentioned intervals of fan entrenchment (incision), fan aggradation occurred across the entire last interglacial/glacial cycle in all climatic settings (i.e. interglacials, interstadials, and stadials). The Domata fan is at the same latitude and only 25 km west of the Sfakia fan studied by Pope and colleagues (2016). It would be surprising if two nearby fan sequences on the south coast of Crete had markedly different aggradation-incision histories if the driving processes were climate change and/or eustatic variations, as both of these factors should almost certainly be nearly identical for the two sites. If there are real differences in the timing of fan aggradation and incision episodes between Domata and Sfakia they likely are the result of internal stochastic variations in catchment hillslope sediment supply to the channels feeding the alluvial fans.
*[We have modified these sentences on the revised version to accommodate the reviewer's concerns]*

P. 2, Line 26-29: It is an inference, based solely upon a morphogenetic interpretation of the topography of the Domata fan that the sequence represents two episodes of fan building as no stratigraphic evidence is provided. An alternative interpretation is that the Domata fan sequence represents a single depositional phase followed by unsteady incision, as is common in alluvial fill-cut terrace sequences (also known as complex-response fill terraces of Bull, 1990). For the former interpretation to be convincing, stratigraphic data delineating two distinct fan depositional units needs to be provided and would substantially improve the manuscript.
*[We believe that the stratigraphic evidence inherent in the presence of a marine cliff that truncates both the upper fan surface and underlying deposits, and the alluvial entrenchment wall of the Klados River suffice as convincing stratigraphic data.]*

**Geological setting of Crete and Vertical tectonics:**
P. 3, Lines 18-24: We think that it is important to qualify these statements. The way that it is written herein is that there is scientific consensus on this topic, which is not the case. The debate is ongoing about uplift transients in the Hellenic forearc and it is important to acknowledge that this only presents one side of the argument. Many researchers favor a slow, mostly steady (at least at time scales greater than several earthquake events) Quaternary history of uplift for the island.
*[We will modify this sentence on the revised version but actually, the only two studies that have derived a series of uplift rates through time on western Crete are Tiberti et al. (2014) and Mouslopoulou et al.(2015) show transient uplift. But in reality, whether there is transient uplift or steady-state uplift doesn't change our conclusions regarding Domata].*

P. 3-4, Lines 32, 1-2: The Holocene notch is buried by the fan at Domata (see supporting data). Basic stratigraphic principles demand that if an extensive coastal geomorphic (geodetic) marker is locally buried by a sedimentary deposit, the deposit must by younger than the geomorphic marker, in this case the Holocene notch. This single field geomorphic observation places the geochronologic results and subsequent conclusions of this manuscript into doubt.
*[In our view, the reviewers' interpretation is incorrect. The notch is cut in the surface of the bedrock, but eroded from the alluvial fan gravel surface due to its susceptibility to erosion. The notch is not preserved behind the beach in the gravel cliffs at Domata for this reason. However, its presence is represented by the low, relict alluvial terraces at the mouth of the Klados River. Between the reviewers' illustration and the Klados River mouth, the bioerosion notch (which we correlate with the c. 365AD paleoshoreline) underlies the base of the fan gravels and is preserved as a notch in bedrock. It is therefore not possible that this feature underlies the alluvial fan sequence].*

**Data – methods – Chronology:**
P. 4, Lines 4-5: This is an interpretation that requires supporting data. *[The supporting data are presented in the following paragraphs, as we state in the following sentence of the manuscript].* The morphology of the fan

might equally well be represented by a single filling episode followed by unsteady incision into the fan deposit *[see above].*

**Coastal geomorphic features at Domata:**

P. 4, lines 20-21: Where does the quartz and feldspar in the fan come from if the bedrock in the Klados River catchment is mostly carbonate? *[This is now incorporated in the text through better description of the bedrock units].*

P. 5, Lines 16-17: This is a key observation, but from figures 2-3 there is no evidence that the "lower fan" onlaps the "upper fan". The lower fan surface could simply be a fill-cut terrace into the maximum aggradational surface of the "upper fan". Please provide stratigraphic observations to support this interpretation. *[Upper fan deposits are cut by alluvial entrenchment and subsequent marine cliffing. Alluvial deposits of the lower fan lap against the cut alluvial entrenchment cliff and the lower fan surface is seamless between the alluvial cliff and the marine cliff. This stratigraphic relationship is clear and there is no other explanation needed, or indeed possible.]*

P. 5, Line 23-24: It is difficult to see these details in Figure 6a. Is it possible to add some close-up photos of what the deposit looks like in detail with examples of the features provided? It would help readers' understanding of the stratigraphy if they could "see" what the fan deposit looks like. All of the overview photographs are great, but readers will be left wondering what the deposits looks like up close.
Also, what lithology makes up the fan deposits? We assume that it is carbonate, but no details are provided. If the deposit is mostly carbonate, where is the quartz and feldspar used for OSL coming from? *[See above]*

P. 5, Lines 23-25: Details on the stratigraphy for the "lower-fan" are great! How does the "upper-fan" stratigraphy differ? In other words, how does one distinguish between the lower and upper fan units as illustrated in figure 6b? These details are essential to the interpretation of two distinct fan units. Perhaps a composite stratigraphic column of the fan sequence would help. *[We have added some more descriptions in the revised version and we have replaced old Figure 5 with a new Figure 5 (attached below) that illustrates clearer the stratigraphic features we discuss as well as the location of the luminescence samples with respect to these features. The two deposits are clearly distinguishable using surface geomorphology. Figure 6b is at the west end of the Domata fan exposures and far removed from the luminescence age sampling sites. The geomorphic and stratigraphic context of these sites are now illustrated schematically in the revised Fig. 5. Lithological characterisation of the two fans units is not a key part of our study].*

P. 5, Lines 25-31: This is a key observation, but the level of detail in Figure 6 is insufficient for the reader to be able to see this relationship. *[See comment above].*

P. 5, Lines 33-34 & Page 6, Lines 1-2: From the way that this section of the text is written, it is unclear if the paleoshoreline (marine bench) is cut into, or buried by the fan. Our interpretation of Figure 6 is that it appear as though the paleoshoreline is buried by the fan. Our field observations from this area suggest that this Holocene shoreline is buried by the fan (see supporting figure).
*[We firmly disagree with this interpretation. This is a surficial bioerosional notch cut into the fan (now removed through erosion) and the bedrock (where applicable). This notch can be nicely followed westward along the bedrock all the way to Sougia and relates to the 365 AD earthquake that uplifted western Crete up to 10m (Pirazzoli et al., 1982, 1996; Shaw et al., 2008; Stiros, 2010; Mouslopoulou et al., 2015)].*

P. 6, Line 10: Wegmann, (2008) and Gallen et al. (2014) also studied Pleistocene terraces on Crete and interpreted them in the context of stratigraphic relationships with interfingered alluvial fan deposits. Furthermore, Gallen et al. (2014) and Runnels et al. (2014) dated several alluvial fans in southern Crete with OSL, in addition to dating marine terraces with the same technique. *[We now include reference to Gallen et al. (2014) in this statement].*

P. 6, Line 10-13: Based on stratigraphic relationships, pedology and OSL geochronology, Gallen et al., 2014 suggest a stratigraphic model for the genesis of marine terraces and alluvial fans based on tectonic, climatic and eustatic considerations in which marine abrasion platforms are cut and marine terrace deposits emplaced during eustatic transgressive-to-highstand phases , whereas Pleistocene alluvial fans are deposited during cooler (and drier) periods associated with relative sea-level low stands when sediment supply presumably is elevated relative to discharge. In addition to the geochronologic constraints on alluvial fan age, the other observation that implies deposition during cool periods is that the surface gradient of coastal alluvial fans on the south coast of

are steep and prograde to a base level far lower than modern day sea level. This observation suggests that the Pleistocene fans are deposited when relative sea level is lower than the present day. This stratigraphic model is relevant because, if preservation potential were not a problem, a fan at lower elevation might be older than a marine terrace found at a high elevation relative to modern sea level. *[We agree that MIS3 and MIS2 represent relative sea level low stands compared to the present day. We make no assumptions of sediment supply or river carrying capacity. We base our ms on clear stratigraphic and geomorphic data that require explanation by a certain sequence of events. We recognise that the Domata fan deposits are MIS3 in age, and the older fan is benched by marine trimming. Given the well-established sea level curve, our objective is to calculate an uplift rate from this observation. We propose no model, but are familiar with those of Gallen et al. (2014) and Pope et al. (2016). To propose a model or even support one existing model on the basis of our work at Domata would be presumptuous.]*

**OSL dating of alluvial fans:**

P. 6, Line 13: Perhaps consider changing the heading of this section and all subsections. OSL stands for optically stimulated luminescence and IRSL stands for infra-red stimulated luminescence. They are different techniques and should be treated as such in the section headings. Similarly, figure 7 shows IRSL results, rather than OSL results as is indicated by the caption and the labels on the x-axis of the figures. Perhaps use "Luminescence dating of alluvial fans"?
*[We agree it may be better to differentiate OSL from IRSL and now use the general term "luminescence dating"].*

P. 6, Line 14-21: The sampling strategy is well thought out; however, why are there no samples from the base of fan unit 2? *[As stated previously, the reason is simply because the deposits below the lower fan are not part of the fan.]* Also what did the sampled horizons look like? The only information on this is for UF-2. Our field observations of this deposit suggest that it is composed primarily of carbonate sediment. Were there individual fine sand-to-silt lenses that were sampled, or simply stratigraphic horizons that were soft enough to hammer a tube into? *[We have added some more descriptions in the revised version and we have replaced old Figure 5 with a new Figure that illustrates clearer the stratigraphic features we discuss as well as the location of the luminescence samples with respect to this features].*

**OSL results:**
P. 7, Line 19-20: Gallen et al. (2014) and Runnels et al. (2014) also dated alluvial fans in southern Crete with Quartz OSL. There is also a new paper by Pope et al. (2016) that has an abundance of OSL data on the Sfakia fan sequence (see comment above). *[See comments above and elsewhere].*
P. 7, Line 19-27: This might be better reserved for the discussion, but what makes Domata unique in that quartz OSL of the fans there does not work? Quartz OSL has worked fine for multiple other studies where geologic conditions are similar 20 (Pope et al., 2008, 2016; Gallen et al., 2014; Runnels et al., 2014). *[See general comments on luminescence dating in our first set of answers].*

P. 7, Line 29: What about the ages makes them reliable? *[Reliable in terms of the sampling and measurement methodology regardless of problems like incomplete bleaching].*

P. 8, Line 4-5: In alluvial fans, incomplete bleaching is a problem (e.g., Rhodes, 2010). It appears that the Domata fan samples suffer from incomplete bleaching. What tests, if any, were preformed to rule-out incomplete bleaching? For quartz OSL results that deviate from every other published study that has been performed on alluvial fans on Crete it is worth 25 investigating why the signals are so different. Provided the short transport distance of the Klados River gorge (~ 5 km) and the high-energy nature of the Domata fan sequence, incomplete bleaching is potentially a major concern.
Taken at face value, the IRSL ages reported in Table 1 imply that the lower (supposedly younger) fan unit was emplaced before the end of deposition of the Upper (older) Fan unit. This is difficult to reconcile with the stratigraphic arguments advanced in this manuscript. Furthermore, the observation that the fan buries a presumably Holocene age paleoshoreline is problematic (see supplemental figure). Some experiments to test for incomplete bleaching or at least a detailed explanation of why incomplete bleaching isn't an issue should be added. *[See general comments on luminescence dating in our first set of answers].*

**Soil development:**
P. 8, Line 28-29: Were these colors derived from a Munsell color chart? *[A Munsell chart was not used. The described colour is the obvious macroscopically difference in colour which is presented also clearly in figure 8b*

*and 8c. However, the difference in colour was obvious and we consider it as in dry conditions (sampling performed in summer, after prolonged period without rainfall)].* If so, the hue, value and chroma values should be provided as they represent a semi-quantitative measure of color and several combinations of hue, value and chroma have the 5 same color name. It would be useful to point out where the weak B horizon is on the Upper fan soil in figure 8C and the soil texture evidence used to support this interpretation. *[The B horizon in the upper fan is already indicated with a dashed line in Figure 8c. It is not clear to us why the reviewer made this comment.]* Having worked on soil profiles on Crete, and based on the photo presented in Fig. 8C, it looks like this profile could be characterized as a thin A horizon over a C horizon. *[As it is stated in our study, a macroscopic approach in the field was used in order to describe the soils, thus the possible B horizon in the upper fan was identified by the technique of knife penetration in the excavated profile. The lower knife penetration (less friable), the change in the texture (accumulation of finer material clay+calcite) and the absence of organic material (aggregates due to organic accumulation are obviously less also in figure 8c) lead us to the conclusion of possible B horizon in the upper fan. However, ongoing work in the soil horizons in the special case of Domata (as it appears from the spectacular alluvial fan development) will reveal further evidence on what we describe as a "(weak) B horizon". In addition, the authors believe that whether the horizon is B or A/B (A/B: in the initiation stage of B horizon), doesn't impact on our basic argument that the upper fan soil is better developed compared to the lower fan soil].*

P. 8-9, Line 31-34, 1-4: Gallen (2013) provides detailed descriptions of alluvial fan soil profiles with OSL geochronology in Chapter 2. Gallen et al., 2014 and Runnels et al., 2014 describe the pedology of alluvial fans of Crete in conjunction with OSL geochronology. These studies should be discussed in the context of this study, as soils formed on alluvial fans of reportedly the same age are distinctly different. Furthermore, aside from color, observations supporting the notion that the soil profiles shown in figure 8B and C are consistent with the descriptions of stage 2C soils from Pope et al., 2008, particularly soil textures consistent with an increase in clay content (e.g. Bt horizon), are not provided. Based on observations presented in Gallen (2013) and Gallen et al. (2014), the soils on these fan surfaces are less mature than fans dated to ca. 40 ka that have evidence of pedogenic alteration to > 1.5 m below the present-day surface (see figure 4 of Gallen et al., 2014 and Appendix of Chapter 2 in Gallen, 2013).

*[The reviewers imply that the soil in our study area is less mature compared to the thicker and better developed soils of the same (or older) ages presented by Gallen (2013) and Gallen et al. (2014). However, we believe that several parameters can dramatically change the soil cover thickness in areas of similar ages, especially when these regions span several 10's of kilometres. Rainfall on the south flanks of Lefkai Ori varies radically from the crest of the 2450 m of mountain range to the south coast (i.e. Domata). The elevation change in the 6.5 km north from the coast is 2100 m, (a gradient of 32%) while in Tsoutsoros area where the Gallen et al. 2014 studied the 'marine terrace soil profiles, has a relief of 0 to 480 m within 3.44 km (13% gradient). There is clearly a difference in orographic rainfall. Overcoming the local rainfall variability, another parameter which is decisive in the soil development is the texture and lithology of the parent rock. Gallen et al. 2014 investigated terraces developed in both Neogene formations and in Mesozoic limestones-sandstones-mudstones of Pindos nappe which are absent from our site. Thus, the results of the soil profiles which are presented by Pope et al. 2008 are the best available "benchmark" where we could interpolate our findings. As already mentioned, the soil depth and the physical characteristics presented by Pope et al. (2008) showed several similarities with those profiles presented in our work. Finally, chronosequence studies on soils are those which are explicitly giving answers in the above questions, however, constrains and limitations of a chronosequence study make them rare especially on the island of Crete where landscape is dictated by intense tectonic evolution. Lin (2011) nicely summarised soil development as an archive of the cumulative effects of tiny repeatedly cyclic changes where even a small difference in soil parameters can create a remarkable difference in pedogenesis].*

P. 9, Line 11-13: This isn't supported by the geochronology (keep in mind the difference between OSL and IRSL). The data suggests that the alluvial fan units are synchronous. The Upper fan unit is bracketed between ~54 and 23 ka and the lower fan unit ceased deposition ~40-28 ka.

*[The authors thoroughly understand the meaning of the data. The IRSL data (with inherent errors) presented in our article support the chronology demanded by the stratigraphy that we have established. The soil development independently supports our stratigraphic interpretation. We conclude that the stratigraphy, luminescence dating and soil analysis independently point to the same interpretation: that the upper fan is older than the lower fan and that they are last glacial (MIS4 to MIS2) in deposition].*

P. 9, Line 13-15: Aside from the problem that the geochronology suggests that the Upper fan surface was abandoned ~5-15 ka *after* the lower fan surface *[this is wrong and we don't understand why the reviewers have included this comment; they appear to be taking the geochronology out of the stratigraphic context and we regard this as inappropriate, particularly given the error margins on IRSL ages]*, is the Liar et al. (2009) reference relevant in this context? Liar et al. (2009) work on Holocene soils. This study, according to the authors is about Pleistocene soils. The effects of a 5 ka difference in geomorphic surface age for surfaces that are presumably an order of magnitude older than that would need to be shown. One would expect that soils forming on different aged surfaces would become more similar over time. For example, a 5 ka time gap for the initiation of soil formation might be negligible after a 30-40 ka shared history of soil development.

*[We acknowledge that the study of Liar et al. (2009) presents Holocene soils and that the soils at Domata are upper Pleistocene. So in the revised version we complete our argument by presenting the study from Huang et al (2016), where they found a clear difference in amorphous and crystalline Fe oxides concentration (hematite) in the soils from 135 ka and 125 ka. In our study, soil age differences of between 4 ka (minimum) and 14 ka (maximum) could easily have left a significant weathering imprint if this time span was characterized by more intense rainfall resulting in faster soil development than today (e.g. see our evolution model, Figure 10). We disagree with their argument that over greater time periods, soils always progressively assume a similar appearance. Soil development is related to climatic (and other) parameters. Our interpretation (Figure 9) is that the upper fan was deposited within the MIS3 and the soil development started after surface abandonment at 40-45 ka; the time that followed was climatically warm and wet in Greece (intense weathering; Tzedakis et al., 2004). The lower fan was deposited from 36 to 29 ka, the start of a dry and cold period. Thus, the soil beneath the lower fan surface developed under cold and dry conditions, while the upper fan soils developed under warm and wet conditions. It is difficult to conceive that soil development has converged during MIS2 and MIS1, "erasing" the past pedogenic processes].*

**Landscape evolution at Domata:**
P. 9, Line 29-30: Why would falling sea-level promote deposition? *[This has been corrected in the revised version].*

P. 10, Line 10-11: Why does relatively high sea-level now promote deposition? What are "deteriorating climate conditions", what evidence is there to support them and why do they promote fan deposition? *[Fan deposition is promoted by a high base level, in this case, sea level of the time. When climatic conditions deteriorated, resulting in a falling sea level, the balance between sediment supply, gradient, rainfall and catchment vegetation re-adjusts, but with on-going sea level fall, entrenchment is the likely outcome (e.g., see Pope et al. 2016)].*

P. 10, line 27-29: But isn't the fan burying this Holocene deposit (see supplemental figure)? *[We firmly believe that the reviewers' interpretation is wrong].*

P. 9-11, Line 17 – 32, 1 – 34, 1-3: Provided that the geochronology is correct, any number of interpretations could be argued to be equally valid within the uncertainty of the data. For a given date the authors utilize either the mean or the median as preferred to provide an older or younger age estimate, respectively. Furthermore, taken at face value the data indicate that the upper fan surface was active until 25 ka. *[We don't believe this is true. The dates are single samples, and they have errors that are presented and documented. Deposition could be anywhere within the error limits. Stratigraphy determines the sequence of events, but dated events provide some chronological information (within error)].* It isn't until lines 1-3 on page 11 that the reader is told that this sandy horizon is a fine-grained cap on the entire deposit. The position of the luminescence samples in a composite stratigraphy of the fan sequence is needed in order to understand the context of the data. *[We have replaced old Figure 5 with a new Figure that illustrates clearer the stratigraphic features we discuss as well as the location of the OSL samples with respect to this features].*

Another observation which should be addressed in the context of the stratigraphic model is how an unvegetated near-vertical cliff of unconsolidated gravel can remain between the Upper and Lower fan units for ~ 39 ka? Furthermore, why is the morphology of the cliffs on the Upper and Lower fan units so similar despite an inferred 35 kyr age difference (see figures 5 and 10 in manuscript)? *[We don't understand how the reviewers get a 35 kyr age difference. The time period between abandonment of the upper fan surface (c.45 kyr) and abandonment of the lower fan surface (36 kyr) is a 9 kyr age difference].* When fault scarps are formed in unconsolidated alluvial fan sediments in places like the Basin and 10 Range of the southwestern United States, they may initially be vertical geomorphic features, but through hillslope erosional processes, the morphology of the scarp changes through time (e.g. McCalpin, 1996). Diffusion of scarps through time has proven to be a

useful relative dating tool in studies of both fault scarps (e.g. Nash, 1980) and paleo-shoreline scarps (e.g. Andrews and Bucknam, 1987). Perhaps this could be attempted in this study.

Why aren't similar fan deposits observed in the five gorges that drain southward off the Lefka Ori (White Mountains)? The Klados River gorge is the only one that preserves such spectacular fans. Despite the contention that Domata beach is geomorphically unique along the southwest coast of Crete, it is not. *[We don't agree with this statement. Catchment sizes vary, river lengths vary, gradients vary, vegetation varies, rainfall varies, and none of the others have the bedrock ridge that forms a gorge close to present day sea level (see Fig. 1 in our manuscript)].* All the major gorge outlets to the ocean along the Lefka Ori are morphologically similar, yet none host similar fan deposits. Each gorge should preserve similar features if, as it is implied the forcing that generated the fans is a coupled climate-tectonic-eustasy signal that should affect the island regionally. *[Again, we disagree. Each gorge is different as indicated above and in lots of other ways, too. Regional signals are always complicated by local factors].*

**The importance of tectonic uplift at Domata:**
P. 11 Line 17-18: Where is the marine terrace that cut this cliff? Based on the stratigraphic model, one world expect marine deposits between the two fans. *[While it would be useful if this was the case, it is not necessarily true. Cliff collapse, as well as the 365AD uplift, keep the sea at bay from the foot of the Holocene sea cliff. In an erosional setting, marine deposition is not necessarily likely, and in addition, early phase fluvial erosion associated with lower fan deposition may have removed such marine deposits, if they were ever deposited].* An uncertainty analysis on the elevation of the inner shoreline and some evidence that this cliff corresponds with a marine abrasion platform would be beneficial for readers. Furthermore, if it is assumed that the base of the fan lies on a marine abrasion platform (e.g. marine terrace), the difference in age between the marine platform and the overlying alluvial fan can be substantial (see Gallen et al., 2014 for examples from southern Crete). *[This appears to contradict the reviewers' suggestion that the fans were both deposited within the Holocene. Given the nature of the Siddall et al. (2003) MIS3 and MIS2 sea-level curve and other high resolution sea-level curves that have been published to date, where sea-level declines from 55 kyr to the last glacial maximum, the time period between platform cutting and renewed alluvial deposition cannot be long. Any tectonic uplift that occurred during this period would merely exacerbate a tendency for rapid entrenchment].*

P. 11 Line 11-32: There is reason to suspect that Pleistocene radiocarbon ages might suffer from alteration of primary material, shifting radiocarbon dead ages to ones that are younger. This is nicely discussed by Wegmann, 2008 and can be noted in $\delta O_{18}$ and $\delta C_{13}$ shifted to more negative (terrestrial) values, relative to marine standards, in Triberti et al., 2014. *[The ages presented in Mouslopoulou et al. (2015) are only of secondary importance in this manuscript. Nevertheless, the results of the current study, using completely different methods, provide an independent test on the results of the Mouslopoulou et al. (2015) study. In both cases uplift rates, averaged over the last ~40 kyr, are similar to one another and are also similar to those derived by dating other marine terraces in western Crete (Shaw et al., 2008; Strasser et al., 2011; Tiberti et al., 2014].*

P. 12 Lines 1 – 15: Again, this is only the opinion of a few and does not represent the view of many other researchers that study the tectonic geomorphology of Crete. It would be useful to include a more thorough discussion of all the relevant literature.

P. 12, Line 17-20: No evidence is provided for the climatic link. The geochronology is simply not precise enough to permit such interpretations. *[Sea-level fluctuations are controlled by (and reflect) climate fluctuations].*

**Conclusions:**
P. 12, Line 25-26: This is an inference based on uncertain geochronology. Perhaps it is better to say "One interpretation of the data" rather than "Data analysis shows". *[In fact, we use the geochronology presented here as independent support for the hard geomorphic and stratigraphic data demanded to develop this sequence of events].*

P. 12, Line 28-29: It's not entirely clear to us how this interpretation is supported by the data.

**References cited in review that are not cited in the manuscript:**

*The references below that are indicated in red are ones we find relevant to our study and will be included in the revised version.*

Andrews, D. J., and Bucknam, R. C., 1987, Fitting degradation of shoreline scarps by a nonlinear diffusion model: Journal of Geophysical Research: Solid Earth, v. 92, no. B12, p. 12857-12867.

*Bull, W.B., 1990, Stream-terrace genesis: implications for soil development: Geomorphology, v. 3, p. 351-367.*

Gallen, S.F., 2013. The Development of Topography in Ancient and Active Orogens: Case Studies of Landscape Evolution in the Southern Appalachians, USA and Crete, Greece. Ph.D. Dissertation, North Carolina State University, Raleigh, NC, 171 pp.

McCalpin, J., 1996, Paleoseismology: San Diego, Academic Press, p. 583.

Nash, D. B., 1980, Morphologic Dating of Degraded Normal Fault Scarps: The Journal of Geology, v. 88, no. 3, p. 353-360. 15

*Pope, R.J.J., Candy, I., and Skourtsos, E., 2016, A chronology of alluvial fan response to Late Quaternary sea level and climate change, Crete: Quaternary Research, v. 86, p. 170-183.*

Rhodes, E.J., 2011. Optically Stimulated Luminescence Dating of Sediments over the Past 200,000 Years, *Ann. Rev. Earth planet Sci.,* **39,** 461-488, doi:10.1146/annurev-earth-040610-133425.

Runnels, C., DiGregorio, C., Wegmann, K.W., Gallen, S.F., Strasser, T.F., and Panagopoulou, E., 2014, Lower Palaeolithic 20 artifacts from Plakias, Crete: Implications for Hominin Dispersals: Eurasian Prehistory, v. 11, p. 129-152.

Scherler, D., Lamb, M. P., Rhodes, E. J., and Avouac, J.-P., 2016, Climate-change versus landslide origin of fill terraces in a rapidly eroding bedrock landscape: San Gabriel River, California: Geological Society of America Bulletin.

Wegmann, K.W., 2008. Tectonic geomorphology above Mediterranean subduction zones; northern Apennines of Italy and Crete, Greece, Ph.D. Dissertation, Lehigh University, Bethlehem, PA, 169 pp. 25.

*We have also used the following additional references in response to the reviewers' comments:*

*Tzedakis, P.C., Frogley, M.R., Lawson, I.T., Preece, R.C., Cacho, I., de Abreu, L., 2004. Ecological thresholds and patterns of millennialscale climate variability: the response of vegetation in Greece during the last glacial period. Geology 32 (2), 109–112.*

*Wen-Shu Huang, Shih-Hao Jien, Heng Tsai, Zeng-Yi Hseu, Shiuh-Tsuen Huang, 2016 Soil evolution in a tropical climate: An example from a chronosequence on marine terraces in Taiwan. Catena, 139, 61–72.*

*Lin Henry 2011 Three Principles of Soil Change and Pedogenesis in Time and Space. Soil Sci. Soc. Am. J. 75:2049–2070.*

*Sincerely,*

*Vasiliki Mouslopoulou and all co-authors*

[Figure]

***New Figure 5:*** *Profiles of fan surfaces projected onto common planes parallel with the modern Klados River channel (above) and parallel with the modern coastline (below). Horizontal and vertical scales for each profile are similar, with a VE of c. 1.25. Note that the downstream slope of the upper and lower-fan surfaces are about the same, both a little steeper than the slope of the modern stream channel. Also note that the upper-fan surface slopes less to the east than the lower-fan surface, so that the upper-fan marine cliff is higher at the eastern end of the beach than at the west. The highest elevation of the lower-fan surface is close to the foot of the incision of the upper-fan, while the upper-fan surface is highest at its incision point. We schematically illustrate the volumes beneath the measured profiles to indicate the likely extent of the volumes of upper fan materials (pink) and lower fan materials (green) and have added the locations of each of the luminescence sample points (annotated yellow dots)."*

[Figure]

*Updated Figure 6b: Looking west across the Klados River mouth (foreground), the uplifted shoreline attributed to the 365 AD earthquake (dashed red line, lower left) aligns well with a low terrace riser on the west side of the river (dashed thick white line, middle). Incision of the modern channel below the surface is attributable to post-earthquake adjustment to new base levels. Note the sea-cliffing of the last interglacial? marine bench (thick red dashed line at the top of the image) and the upper and lower fan deposits. The upper fan and western parts of the lower fan are overlain by accumulated rockfall debris and the solid thin white line approximates its surface. The lower fan surface is marked with fine dotted white line.*

---

## Author Response (AR1)

**Detailed Response to the Reviewers & Associate Editor**

**Ms. Ref. No.: esurf-2016-62**

**Title: Distinct phases of eustatism and tectonics control the late Quaternary landscape evolution at the southern coastline of Crete**

**Dear Associate Editor,**

On behalf of my co-authors, I am now in the pleasant position to return to you the revised version of our manuscript, together with the detailed answers to all the comments raised by the reviewers/commentators of this manuscript. As you may see below, we have taken very seriously all concerns expressed here and we have devoted some time in trying to address each comment (regardless whether we agreed or disagreed).

Overall, the whole review process has been interesting and beneficial and we firmly believe that our work improved for all the useful comments provided by the reviewers. Having said this, at the same time, we felt that there was an overall pressure to make our work what it clearly isn't: a detailed sedimentological study, mainly because of the detailed sedimentological studies provided by Pope et al. (2008 & 2016) and Nemec & Postma (1993) at a nearby alluvial fan system in western Crete.

At this point, we would like to make clear that although we have all worked with alluvial fans, our motivation and objectives are markedly different to those of Pope et al. (2008 & 2016) or Nemec & Postma (1993). Our primary interest in our work on Crete is to help constrain its late Quaternary tectonic development. With the submitted work we aim to use first-order geomorphic features and stratigraphic relationships to understand the late Quaternary vertical deformation of western Crete. This objective does not include a comprehensive description of the materials of the Domata fans and their developmental chronology, as has been undertaken for other fans of southern Crete by others (Nemec & Postma, 1993; Pope et al., 2008, 2016). Neither is the objective to compare Domata fan stratigraphy with that of other alluvial fan systems on Crete.

For addressing our objective, we recognised the importance of the Domata fan sequence and the important marine bench cut in upper-fan at Domata. This feature allows us to independently derive a late Quaternary uplift rate from this section of western Crete. The paper is important because of this independence from previously derived uplift rates on western Crete (e.g., Shaw et al., 2008; Strasser et al., 2011; Tiberti et al., 2014; Mouslopoulou et al., 2015). The latter is something that our work achieves – and something that, for example, the work of Pope et al. (2008 and 2016) or Nemec and Postma (1993) has not achieved (exactly because the scopes of our manuscripts are different).

Nevertheless, we have taken on the advice of most reviewers to present a detailed stratigraphic log of the two fan deposits in the revised version and we have added extensive discussion comparing the first-order conclusions of our study to that of Pope et al. (2016) and others studies as well. And we do welcome the prospect of additional work that would enhance insight into the processes of fan deposition and a refined chronology that may help better understand relationships between sediment transfer rates and climate in this area.

In summary, we do hope that this revised version, together with our detailed response to the numerous review comments, will convince you that our work warrants publication in the Journal of Earth Surface Dynamics.

In the following paragraphs, we carefully address the comments and suggestions of:

- the Associate Editor (Dr. Veerle Vanacker),
- Each of the two reviewers of our manuscript (the Anonymous Reviewer ≠1 and M. M. Tiberti)
- Gallen and Wegmann's two short-comments, uploaded during the online interactive discussion (one posted on January 27th, 2017 and the second on January 30th, 2017).

*Our responses appear in 'red italics' while the comments of the Reviewers/Asocciate Editor appear in* black.

At places where the reviewers refer to specific pages and line-numbers we answer point-by-point; in circumstances where the reviewers do not refer to specific line numbers, we present answers below each of their comments.

In the revised version we have put effort to rephrase the scope of this paper, to more clearly describe the main geomorphic features observed and their interrelations, to present the lithostratigraphic logs of the two studied alluvial fans and to compare our work with the work undertaken by Pope et al (2016) and others on western Crete by:

- Including additional text (see tracked changes throughout our manuscript),
- Modifying our former Figure 5 (now Figure 5a) to include schematic sketch of the volumes of the upper and lower fan materials and the stratigraphic position of each of our IRSL samples within these volumes.
- Including a new figure (Figure 5b) that presents the lithostratigraphic logs of the two alluvial fans examined.
- *Re-annotating/modifying the existing Figures 6b*
- Introducing a new figure (Figure 6c) that shows the details of the relationship between the bioerosional AD 365 notch and the unconformity at the base of the fan sequence.
- Including discussion on the important work of Pope et al. (2016), that was missing from our originally submitted version as its publication post-dated our work.
- Expanding on comparing our work with other published studies on Crete.
- Updating our reference list.

In addition, we have modified accordingly the revised version to accommodate all other specific secondary comments of the reviewers/commentators.

**Response to the comments of Associate Editor, Dr Veerle Vanacker**

We now have three detailed reviews back on the manuscript, that provide valid and very useful suggestions for the improvement of the final document. The revised version of the paper needs to address all comments raised by the reviewers, and I would like to suggest you to take the following four elements along in your revisions:

**1.** The conclusions should be robust and based on facts and data. In the paper, the authors conclude (p14, L 1-7) that "sea-level fluctuations in response to varying climatic conditions formed the landscape at Domata during MIS 3 (57-29 kyr BP). It is, however, because of the fast tectonic uplift that Crete experienced during the subsequent 20 thousand years that the entire alluvial sequence escaped destruction and/or modification due to marine inundation and is preserved sub-aerially today". As tectonism, sea level variation and paleo climate have regional impact, the observations at Domata cannot be isolated from other studies on

nearby alluvial fans. A discussion of the findings from the Domata fan, in the light of previous work on alluvial fans in Crete is essential (see comments raised by reviewer#1, reviewer#2). If there exist different hypotheses about the Quaternary evolution of the region, try to give the different models of evolution and discuss alternative interpretations based on facts and data.

We agree that there should be some discussion comparing our results with other published results on nearby alluvial fan systems. The only such work in the vicinity of our study area is that of Pope et al. (2016) on the alluvial fan system of Sfakia [the latter study builds on previous work undertaken on the same fan-system by Pope et al. (2008) and Nemec & Postma (1993)]. Although the two studies have very different motivations and objectives (see introduction of our response), we agree that there is a reason for comparison. This is why now, in the revised version, we have added discussion at the end of Section 5 (Page 13, lines 17-30) to compare the results of these two studies and also provide possible explanations as to why aspects of 'theirs' and 'our' findings may differ. Please note that despite these differences, both studies have come up to the same conclusion: the recognition of the importance of base level (sea-level) changes to the process of incision (in addition to Section 5, we also state this in Page 3, lines 5-6). Besides, reference to these and other studies (on western and eastern Crete), especially those related to tectonic information and how they may compare to our findings, exist now throughout our revised manuscript:

- Page 2, lines 24-31
- Page 3, lines 5-6
- Page 3, lines 24-31
- Page 4, lines 1-10
- Page 5, lines 14-19
- Page 6, lines 25-31
- Page 8, lines 7-9
- Page 9, lines 20-26 and 30-33
- Page 10, lines 1-5
- Page 12, lines 25-33
- Page 13, lines 1-4 and lines 10-11.
- Page 13, lines 18-30.

**2.** The text misses some consistency in the interpretations as outlined by the reviewers. To give one example: p4, Line 5-10: the interpretation of the tilted paleoshoreline. The authors state that..." A number of studies have constrained the timing of this prominent paleoshoreline at 1.5-2 kyr BP, with some attributing it to the AD 365 historic earthquake". However, we see on p6, L24-30 : "bioerosion notch indicating an uplifted paleoshoreline at 6 m "..." seismically uplifted paleoshoreline dated at 365 AD"..." the AD 365 bioerosional notch ". Also in Figure 6, the authors only refer to the "365 AD paleo-shoreline" not mentioning alternative explanations (Pirazzoli et al., 1982; 1996; Stiros, 2001; Shaw et al., 2008).

There is no alternative explanation. Both of the above statements are correct. The prominent paleoshoreline that we discuss has been dated all around western Crete (more than 65 radiocarbon ages from 5 different studies) and all ages range between 1500-2000 yrs BP, with a strong clustering of ages around the AD 365. Well, this is not a surprise as the dawn of July 21st, 365 AD there was a massive (M~8.3) earthquake that hit eastern Mediterranean, triggering a mega tsunami that affected numerous cities in middle east and north Africa, as well as Cyprus, Turkey, south Italy/Sicely and, of course, Greece (as its epicentre was offshore western Crete).

Nevertheless, to ease the Associate Editor's concern we have 'corrected' the statement in Page 4 (line 5-10) in order to read as following: 'A number of studies attribute the timing of this prominent paleoshoreline to the AD 365 historic earthquake'. Thus now the age of this prominent paleoshoreline on Crete is stated, throughout the manuscript, as AD 365.

**3.** As the chronological framework of the sedimentary events is not well constrained (5 samples, with four samples giving a broad time interval of 30 to 60 kyr BP, Figure 9), one needs to be very careful with the interpretation of landscape evolution (section 4, and Figure 10). Supplementary information (either from soil chronosequences, or lithostratigraphy) would be very helpful. Two reviewers have recommended to include lithostratigraphic logs, or detailed stratigraphic and sedimentological evidence of the fan sequences, and I fully support these recommendations. See details in review#1 and 1-bis and review#2.

To address the reviewers and Associate Editors concern we have done our best to produce detailed lithostratigraphic logs of each of the two fans preserved in the alluvial system. These new logs are now presented in a brand new figure: Figure 5b. Reference to this figure has been added throughout Section 3.1. Their locality is shown now on Figure 5a.

We firmly believe that the insertion of the new Figure 5a, the insertion of new Figure 5b, the serious revision of Figure 6b and the insertion of new Figure 6c, together with our revised text where we more explicitly describe the stratigraphic evidence present in the study area, **leave no space for misunderstanding regarding the stratigraphy encountered at Domata**. And it is this stratigraphy, that is established on a concrete base, that dictates the sequence of events presented ultimately in Figure 10. The IRSL dating is only supplementary and helps placing **this sequence of events**, for which we feel much confident, within a chronologic context (and that is MIS 3).

**4.** Carefully revise and edit the figures 5 and 8. Figure 5 is a new figure showing the topography of the fan surfaces. Can you give the orientation of the profiles (or location on Fig 4), and the unit of the X-axis? Figure 8 shows two soil profiles. One would typically give the soil depth (depth till C horizon, starting at 0 cm at the surface). Also, what is meant with "parent rock"?

We have now carefully revise Figure 5 and 8 as indicated by the Associate Editor. Specifically, in the old Figure 5 (now Figure 5a) we have included the orientations of the profiles (their location is also indicated on Figure 4) and we have added units (m) on both x-axes. Regarding Figure 8, we have reversed the log, so that now zero being the ground-surface (as requested). We have also replaced the phrase 'parent rock' with 'fan-gravel'.

**Reviewer≠1 (anonymous)**

From our understanding, his/her main concern is the unsatisfactory results of the luminescence dating. We totally agree, as we clearly state in the manuscript, that the luminescence results have large uncertainties and are, therefore, unsatisfactory. However, our sequence of events presented in this manuscript is not based on the luminescence dating. It is based on, and demanded by, the stratigraphy and the stratigraphic relationships that we recorded at Domata.

Before we go addressing in some detail this and other secondary remarks made by the Reviewer, we would like to make a general statement about the objectives of our work and constrain its scope. This may help allay some concerns expressed by the Reviewer.

Our primary interest in our work on Crete is to help constrain its late Quaternary tectonic development. With this work we aim to use first-order geomorphic features and stratigraphic relationships to understand the late Quaternary vertical deformation of western Crete. In doing so, we recognised the importance of the Domata fan sequence and the important marine bench cut in upper-fan at Domata. This feature allows us to independently derive a late Quaternary uplift rate

from this section of western Crete. The paper is important because of this independence from previously derived uplift rates on western Crete (e.g., Shaw et al., 2008; Strasser et al., 2011; Tiberti et al., 2014; Mouslopoulou et al., 2015 – all references included in the submitted version).

To achieve these objectives, we needed to understand the first order fan geomorphology and from it dissect out the sequence of events required in landscape evolution. We derived a basic sequence of events that is demanded by the stratigraphic relationships present at Domata and present these in the manuscript. We have gone to some effort to place the Domata fan sequence within its stratigraphic and chronologic context so that we fully understand the uplift rate derived. The final piece of the puzzle required to derive our uplift rate was its integration with a high quality sea level curve for the last 125 kyr. We chose Siddall et al.'s (2003) sea level curve because of its precision, relative proximity to the Mediterranean, yet its isolation from the variable tectonic signatures and isostatic problems associated with glacial loading of that region, and for its similarity with the Lisiecki & Raymo (2005) stacked curve.

The luminescence dating we undertook simply provides confirmation of the chronological framework for the events that we had previously deduced with reasonable confidence from other stratigraphic and geomorphic observations. We completely agree with the reviewer that our luminescence dating, with its large uncertainties, cannot provide adequate resolution to separate the individual events presented here. The landscape evolution proposed in Figure 10 is not based on the luminescence dating: it is based on the sequence of events that is demanded by the stratigraphic relationships present at Domata. The luminescence dating comes only to provide the chronologic frame (MIS 3) within which this sequence of events took place.

For detailed discussion on the major issues relating to the IRSL dating we kindly direct the Reviewer to Pages 3-4 of our response to the short comment of Gallen and Wegmann (dated 27 Jan 2017) during our online interactive discussion.

**General Comments (Reviewer ≠1)**

The Mouslopoulou et al. paper deals with a topic that has both environmental and tectonic implications, focusing on a well-expressed sea-side alluvial fan. Compared to the other fans that make up the coastal bajada of south Crete that develops ca 20 km further east, Domata fan is a relatively small one (area c. 0.2 km2), along with a string of other isolated fans that border the steep southern Cretan coast. Nonetheless, its stepped morphology, with an escarpment running roughly parallel to the present-day coastline may be indicative of processes that significantly affect fanshape evolution, as the 'marine trimming' suggested by the authors. Indeed, Mouslopoulou et al. have tried to link fan evolution to climatic changes, coupled with uplift rate scenarios for the past few kyr and come up with a scenario that ties the evolutionary stages of the fan to successive marine high- and lowstands.

To achieve such a target, one has to resort to high-res geochronology, something that is not always feasible. Her lies, in my opinion, the main drawback of Mouslopoulou et al.'s work, which is found in their sampling strategy and the resulting dating accuracy: while the latter may not be the authors' fault, the former is a weak point in this work. Sampling did not include the body of the 'lower fan': this might clarify –and probably strengthen the authors' distinction of the Domata fan in an 'upper' and a 'lower' one. (Of course, this might not help, either, as the results from the other samples contain significant errors).

Having said these, I acknowledge the fact that the number of dated samples (and possibly the range of dating methods applied) are dictated by research funding; nonetheless, one has to make do with what they have, I'm afraid. At any rate, the resulting OSL ages contain significant errors (as also

acknowledged by the authors: P.8, I. 14); it is also unclear whether standard deviation and standard error refer to  $1-\sigma$  or  $2-\sigma$  (*they refer to 1\sigma, as it is stated in Figure 9*). Hence, temporal resolution is too poor to support such a detailed evolution scenario, whose resolution is as high as 1kyr, while most ages overlap significantly and the correlation of successive events to KDEmax is not satisfactorily constrained. *For this comment, please see our introductory statement above.*

The lithostratigraphy of the Domata fan is poorly described; lithostratigraphic logs are missing – and these could help place the obtained samples in a coherent geological context. *We have now added detailed lithostratigraphic logs (Fig. 5b) and have also modified the old Figure 5 (now Fig. 5a) to illustrate the relationship of the samples and the geomorphic units.* Moreover, it might clarify any probable lithological or other difference between the units described as "upper" and "lower" fan. *Yes, we have now added detailed lithostratigraphic logs (Fig. 5b).* This could also be aided if appropriate figures (esp. photographs) were included, to show the lithological composition of the fan(s) *Yes, we have now added logs and associated photograps (Fig. 5b).* Panoramic photos are fine, but some close-ups would be very useful for the reader to understand the distinction made between Upper and Lower Fans *Yes, we have now added close ups, see Figure 5b and Figure 6c.* Figure 8 (b and c) focus on the soil cover and do not serve this purpose.

The above stated objectives do not include a comprehensive description of the materials of the Domata fans (lithostratigraphic logs) and their developmental chronology, as has been undertaken for other fans of southern Crete by others (e.g., Nemec & Postma, 1993; Gallen et al., 2014; Pope et al., 2008, 2016). Nevertheless, we have rephrased the text in the revised version of our manuscript to better describe these geomorphic features and their interrelations. Besides, to better illustrate the first-order stratigraphic relationships between the main geomorphic features, we have now modified Figure 5 by illustrating schematically the volumes beneath the measured profiles to indicate the likely extent of the volumes of upper fan and lower fan materials. We have also added the locations of each of the luminescence samples. We have also now included stratigraphic logs (new Figure 5b) of the two fans, and associated close-up photographs, to show the detailed stratigraphy (and the differences) on the fans examined. We have further followed the Reviewer's advice and we have modified Figure 6b to better reflect the relationships between lower and upper fan units. In addition, we have also included a close-up image (Fig. 6c) to better reflect the relationship between the unconformity at the base of the fan sequence and the AD365 bioerosional notch.

Between Domata and Sougia (c. 9 km to the west) there is a number of fans in practically the same a geomorphological and geological environment (the same could also be supported for the bajada the east, with Sfakia fan being its westernmost member). Processes suggested in this paper (i.e. "marine trimming") are not localized ones and affect extended tracts of land. So if such a process was responsible for the modification of the Domata fan, why it is not found elsewhere along this coast?

It is true that along the southwestern coastline of Crete (e.g., east and west of Domata), there is a number of alluvial fans in roughly the same geomorphological setting. Many of these fans are indeed clearly truncated by the sea (Figure A below, near Trypiti, west of Domata). Although the 'double trimmed' alluvial fan-system at Domata is unique in its kind (as each episode of the two alluvial fan-building episodes has been followed by episodes of alluvial incision and subsequent marine trimming), evidence for 'double marine trimmings' exists elsewhere on western Crete as well. In Figure B below, for example, we provide evidence for a similar geomorphic feature that occurs at Agia Roumeli (about 4 km east of Domata), where an alluvial fan-system is truncated by two distinct marine-trimming events. It is hard to argue that these cliffs (illustrated in Photo B) are not trimmed by marine processes as they are parallel to the coast and on steeply sloping fans. Besides, on the upper cliff, the one with the sea-caves (where the hiking-track from Domata to Agia Roumeli passes through), we have found beachrock with shell-hash.

---

## Referee Report (RR1)

**Review of Mouslopoulou et al. (revised version) by M. M. Tiberti**

In my opinion, the Authors succeed in improving the clarity and accuracy of the data presentation, that was my main concern. They also added some necessary elements of discussion.

I would have preferred a complete reorganization of the manuscript, as previously stated, but I think this is mostly the Editor's choice.

I suggest just some more technical corrections:

Page 2 Lines 30-31: too many repetition of the word "*episodes*" in this sentence, please correct

Page 4 Line 11: "*alluvial fans*" instead of "*an alluvial fan*"

Figure 7: remove "*OSL*" from x-axis label

---

## Referee Report (RR2)

Review of revised manuscript for Earth Surface Dynamics, esurf 2016-62: Mouslopoulou et al., "Distinct phases of eustatism and tectonics control the late Quaternary landscape evolution at the southern coastline of Crete"

Reviewer: Mark Brandon, Yale University June 6, 2017

Overview

I have read through all of the available reviews, short comments, and replies, and the current revised manuscript and associated summary about the revisions. My conclusion is that this manuscript makes a useful contribution about the stratigraphy and age of alluvial units in an interesting locality in SW Crete. The data are original, well documented, and very nicely illustrated, and they will be useful for future geomorphic and tectonic studies. Crete is also widely recognized as an important active tectonic setting. Thus, I strongly recommend that the manuscript be "accepted", which I guess means moving it from discussion to publication status.

A shortcoming, however, is that the manuscript, in its revised form, fails to establish an intellectual framework for the work. Why should the reader be interested in the paper? What are the specific competing ideas or hypotheses? The introduction focuses on eustasy versus tectonics, but there are no explanation about why this issue is important. In their reply to comments, the authors indicate that they are mainly focused on tectonic not geomorphic issues. But if the focus is on tectonics, then I would expect an introduction that summarized competing tectonic ideas, and a discussion that showed how the new data tested those ideas. At present, the main contribution is that the rock-uplift rate is about 2.5 mm/a in a small location in Crete. There are a couple of vague sentences about steady versus unsteady uplift. One would think that Crete, which lies above an active subduction zone, would be an ideal setting for testing tectonic interpretations.

I follow here with some specific feedback, which the authors might find useful for sharpening their presentation.

Specific Comments

1) The core contribution of this manuscript is strongly challenged by the short comments from Gallen and Wegmann. Their most significant challenge is to suggest that the Domana fan units are not 20 to 30 ka, as proposed by the authors, but rather <10 ka. The authors need to directly address this issue in the manuscript. It is important to remember that the short comments and reviews will be published along with the paper, so they can be directly addressed and cited in the paper.

2) Based on my reading, I side with the authors for their interpretation of the age of the Domata fan units. Gallen and Wegmann argue for a Holocene age based on relationships relative to a "Holocene" (?365 AD) sealevel notch. The authors clearly show that the notch is probably not equivalent to the unconformity

beneath the lower fan unit. I am also convinced by the IRSL Kspar ages. These ages are messy, but each of the five ages has a well-defined minimum-age component. The clustering of a minimum-age component in each sample suggests a similar bleaching history for all of these grains. That suggests to me full bleaching for this component. In addition, the IRSL ages appear to be consistent with the stratigraphic ordering of the samples. (It would help to organize the data plots in Figure 7 in stratigraphic order, and to discuss this relationship in the text.)

3) Rex Galbraith would blow a gasket if he read this citation to the Galbraith and Roberts (2012) paper (p. 8, line 30), in that citation is used to justify using the mode in the density plot (as determined by the KDE method) as an estimate of the age. Galbraith has spent his career lecturing against the use of probability density plots. In fact, the Galbraith and Roberts review recommends estimating "minimum ages" for mixed OSL grain-age distributions. If the authors want to use the mode of the probability density plot as an estimate for the age, then they should remove the Galbraith and Roberts' citation. Alternatively, they could calculate minimum ages, and, in that case, the Galbraith and Roberts citation would be appropriate. My recommendation is to use the minimum age estimate. That approach provides a clean way to deal with mixed grain ages.

4) The manuscript would be stronger if it had a more critically developed interpretation (see section 4). As presently written, the interpretation read as a "just-so" story (i.e. a self-consistent narrative). The reader is left to wonder if there might be other interpretations that fit the data. In addition, I would expect that if the issue of steady versus unsteady uplift is important, then the interpretation should address this issue as well.

5) The manuscript contains many cases where an argument is advanced on the basis of what might be "likely" or "unlikely" to have occurred (see below for partial list). My sense is that these terms are used to indicate what the authors view as reasonable and unreasonable aspects of their interpretations. Unfortunately, this phrasing tends to make the argumentation sound weak. I would recommend stating the interpretation in direct terms and to avoid personal assessments of whether the ideas are likely or unlikely. That judgment is probably best left to the reader.
   p. 10, line 20: The initiation of deposition of the upper-fan (Fig. 10a) is likely to have occurred post ~50 kyr BP and prior to 45 kyr
   p. 10, line 23: We argue that upper-fan deposition is unlikely to have started as early
   p. 11, line 14: "Marine trimming of the lower-fan surface and deposits is unlikely to have occurred
   p. 11, line 16: "The Holocene high sea-level stand is the most likely candidate period

6) The term casual "marine bench" is widely used throughout the paper. The term "marine terrace" or "marine abrasional surface" is probably more precise. Also, it would be useful to know if there is direct evidence that these "flats" were cut in the surf zone, as opposed to being fluvial straths. I suspect that this distinction was made by considering the context of the flat relative to the coast and adjacent river channels. Nonetheless, it would probably help to know if the marine versus fluvial origin of these flats included evidence based on fossils or deposits.

7) The title includes the noun "eustatism" as a modifier. I would suggest using the adjective, "eustatic". For example, "Distinct phases of eustatic and tectonics forcing for late Quaternary landscape evolution in southwest Crete".

Recommendations for Figures

Caption for Figure 1: "Numbered arrows show geodetically-derived convergence rates between the African and Eurasian plates and their azimuths at selected sites (after Reilinger et al., 2010)." >>> "Labeled arrows show geodetically-derived site velocities (mm/a) relative to a fixed Nubia plate (Reilinger et al., 2010)."

Figures 3, 5, 6: Change "upper fan surface" and "lower fan surface" to "upper fan tread" and "lower fan tread". Surface is vague. Tread clearly indicates the uppermost limit of an inset geomorphic unit.

Figure 4:
1) Change labels as such: profiles > profile (Fig. 5 shows only one profile for
2) The label "marine cliff profile" should refer to Fig. 5b (not 5a).

Figure 6a:
1) My understanding is that the "upper fan gravels" are inset into the "lower fan gravels". It might help to use a different contact line to indicate an inset relationship. Perhaps something like TTTTTTTTT (like a normal fault contact) with the barbs pointing to the inset unit.
2) In the lower panel, the "marine bench" is shown as grading into the "upper fan gravels". This relationship is inconsistent with the description in the text. The line work needs to be modified (or the text rewritten) to resolve this conflict.
3) The term "marine bench" is a bit confusing. Does it refer to the wave-cut unconformity (marine erosion) or does it refer to a depositional unit below the unconformity? The position of the label suggests that it is depositional unit. Is there evidence that this unit was a marine deposit?

Figure 6b: This figure is a bit confusing and incomplete.
1) Unit labeled "rockfall debris post-dating upper fan deposition" lies within the dashed line that encircles the upper and lower fan deposits. I recommend

placing the dashed line below this "rockfall debris" unit, to make it clear that it is not part of the two fan units.

2) I recommend showing here the "Older Quaternary deposits" unit in the lower left side of this image (see Figure 6c).

3) The unconformity at the base of the fan units is not highlighted, and the "beach bench" unit is not labeled.

Figure 6c: The "Older Quaternary deposits" are not dated, so it is best to call them "older deposits" or "older sediments".

Figure 7
1) What are the numbers on the right axis of these plots? I suspect that they show the cumulative distribution in terms of "Cumulative Grains". This right axis need a label.

2) Left axis is labeled "kernel density estimate" but the units are not specified. Of course, density plots do not require any units since they are only used to indicate a relative sense where the location of the probability mass of the dated grains. If scaling is needed, then I recommend, for simplicity, to leave this axis blank. If the authors do insist on labeling this axis, then it should be called "probability density" and the units should be shown (dP/dTau where P is probability and tau is age; could use density units of grains/ka or %/ka).

Table 1.
1) The columns showing locations should be labeled "Longitude" and "Latitude". Easting and northing refer distances, as used for UTM coordinates.

2) The term "mode" (in the sense of mode of the probability density distribution) should be used instead of "KDE max".

---

## Author Response (AR2)

**Detailed Response to the Reviewers & Associate Editor**

**Ms. Ref. No.: esurf-2016-62**

**Title: Distinct phases of eustatism and tectonics control the late Quaternary landscape evolution at the southern coastline of Crete**

*Dear Associate Editor,*                                    *6 July 2017*

*We have now completed the revision of the revised version. The comments of Reviewer-1 were easily addressed and the comments made by Reviewer-2 were straight forward and very useful. The most important change is the calculation of minimum ages, as recommended by Prof. Brandon.*

*In the following paragraphs, we carefully address the comments and suggestions of:*
- *the Associate Editor (Prof. Vanacker),*
- *Each of the two reviewers of the revised version of our manuscript (Dr. Tiberti and Prof. Brandon)*

*Our responses appear in 'red italics' while the comments of the Reviewers/Associate Editor appear in* black.

*In the uploaded revised-revised version we have:*
- *Included additional text in the introduction, end of section 3.1 and section 5 (all tracked)*
- *Modified Figure 2 (replaced 'marine bench' with 'marine terrace').*
- *Modified Figure 3 (replaced 'upper/lower fan surface' with 'upper/lower fan thread').*
- *Modified Figure 4 (removed 's' from the word profile).*
- *Modified Figure 5a (replaced 'upper/lower fan surface' with 'upper/lower fan thred' and 'sea-level' with 'abrasion surface').*
- *Modified Figure 5b (replaced 'wave cut bench' with 'marine abrasion surface').*
- *Modified Figure 6a (replaced 'marine bench' with 'abrasion surface'. Also modified the white bedding in the lower panel).*
- *Modified Figure 6b (modified the white and red lines according to the reviewer's request).*
- *Modified Figure 6c (replaced 'older Quaternary deposits' with 'older deposits').*
- *Provided a new Figure 7, including also Minimum ages, as requested by Prof. Brandon.*
- *Modified Table 1.*
- *Recalculated minimum ages.*
- *Answered all minor comments of both reviewers (see below).*

*In summary, we do hope that this revised-revised version, together with our detailed response to the review comments, will convince you that our work warrants publication in the Journal of Earth Surface Dynamics.*

*Dr. Mouslopoulou and co-authors.*

**Response to the comments of Associate Editor, Prof.* Veerle Vanacker**

Based on the comments, I find that your manuscript may be suitable for publication after revisions.

1. To quote reviewer 2: "the manuscript fails to establish an intellectual framework for the work". This comment was already raised in the discussion phase, and reviewer#2 suggests a possible way to sharpen the focus of the paper in the introduction.
*Addressed. Please see response to Prof. Brandon's introductory comment.*

2. The chronology of the Domata fan units as presented in the paper is challenged by the comments from Gallen and Wegmann. Their viewpoint and arguments need to be addressed in the paper.
*We were satisfied to see that Prof. Brandon is also convinced, as is Dr. Tiberti, of the non-Holocene age of the Domata fan (that was the main 'challenge' posed to our work by G&W). We have taken the challenge from G&W that these fans may be of Holocene age seriously and considered all angles of available evidence. We remain unconvinced that this is a possibility. In this revised-version, we have followed Prof. Brandon's, as well as your, advice to directly deal with G&W's comments and have added a paragraph at the end of section 3.1 that we believe leaves no doubt that their argument has some serious flaws.*

3. Given the uncertainty on the age estimates, reviewer#2 suggests to "calculate minimum ages". The minimum age components can then be used in the discussion.
*Done. We have calculated minimum ages as requested by the reviewer (see response to Prof. Brandon's comments below).*

**Reviewer≠1:* Dr M.M. Tiberti, INGV, Italy**
*We thank the reviewer Dr. Tiberti for her positive feedback. Below we show that we corrected all three points that she made.*

I suggest just some more technical corrections:
- Page 2 Lines 30-31: too many repetition of the word "*episodes*" in this sentence, please correct
  *Corrected.*
- Page 4 Line 11: "*alluvial fans*" instead of "*an alluvial fan*"
  *Corrected.*
- Figure 7: remove "*OSL*" from x-axis label
  *Corrected and entire Figure 7 is now replaced.*

**Reviewer≠2: Prof.* Mark Brandon, Yale University, USA**
*We thank the reviewer Prof. Brandon for his very useful comments which improved our manuscript. Below we answer to all of his points and remarks. Note that we have accommodated all of his comments/recommendations.*

Review of revised manuscript for Earth Surface Dynamics, esurf 2016-62: Mouslopoulou et al., "Distinct phases of eustatism and tectonics control the late Quaternary landscape evolution at the southern coastline of Crete"

Overview

I have read through all of the available reviews, short comments, and replies, and the current revised manuscript and associated summary about the revisions. My conclusion is that this manuscript makes a useful contribution about the stratigraphy and age of alluvial units in an interesting locality in SW Crete. The data are original, well documented, and very nicely illustrated, and they will be useful for future geomorphic and tectonic studies. Crete is also widely recognized as an important active tectonic setting. Thus, I strongly recommend that the manuscript be "accepted", which I guess means moving it from discussion to publication status.

*Thank you.*

A shortcoming, however, is that the manuscript, in its revised form, fails to establish an intellectual framework for the work. Why should the reader be interested in the paper? What are the specific competing ideas or hypotheses? The introduction focuses on eustasy versus tectonics, but there are no explanation about why this issue is important. In their reply to comments, the authors indicate that they are mainly focused on tectonic not geomorphic issues. But if the focus is on tectonics, then I would expect an introduction that summarized competing tectonic ideas, and a discussion that showed how the new data tested those ideas. At present, the main contribution is that the rock-uplift rate is about 2.5 mm/a in a small location in Crete. There are a couple of vague sentences about steady versus unsteady uplift. One would think that Crete, which lies above an active subduction zone, would be an ideal setting for testing tectonic interpretations.

*We completely agree with Prof. Brandon. We have now modified accordingly the introduction and Section 5 (see details below) but we would also like to add the following remark as part of our response to Prof. Brandon's comment:*

*An island that lies above an active subduction margin is, indeed, a natural 'seismograph', capable of recording vertical motion due to large-magnitude earthquakes. This is why, inspired specifically by Crete, we recently published an article revealing transient uplift-rates along the Hellenic margin due to clusters of mega-earthquakes on upper-plate faults (Mouslopoulou et al., 2015; GRL) and later tested successfully this idea on several other margins globally (Mouslopoulou et al., 2016; Tectonics). This is to clarify that we have performed already a nearly exhaustive analysis on the subduction seismogenesis (including stable vs. transient uplift) of the Hellenic margin.*

*And that was possible because we had collected uplift rate data through time from individual localities along and across the margin.*

*Here, at Domata, we only have data for one uplift-rate, deriving from a single time-period (ca. 39 kyr BP). This is still very valuable, for the reasons stated in Section 5 of the current manuscript, but by having only one data-point, we can't independently test different models (e.g. steady-state uplift vs. transient uplift), as proposed by Prof. Brandon (and we did in our GRL paper).*

*However, what we have done in this current article, that adds value to this Esurf work, is to combine the tectonostratigraphic information that derive from Domata with published*

*tectonic uplift-rates in order to place constraints on the interaction between eustacy and tectonics and test the idea put forward by Pope et al. (2016) that tectonics do not play a major role in the landscape evolution in southwest Crete. This is now stated clearer in the introduction [Page 2, lines 25-31] and Section 5 [P. 12, lines 28-30; P. 13, lines 32-35; P. 14, lines 4-5 ] of the revised manuscript. We hope that these additions give a more generic flavour to our work.*

*References cited above:*

*Mouslopoulou, V., Nicol, A., Begg, J.,Oncken, O., Moreno, M., 2015.Clusters of mega-earthquakes on upper plate faults control the Eastern Mediterranean hazard. Geophysical Research Letters, 42, 10,282–10,289, doi:10.1002/2015GL066371.*

*Mouslopoulou, V., Oncken, O., Hainzl, S., Nicol, A., 2016. Uplift rate transients at subduction margins due to earthquake clustering. Tectonics, 35, 2370–2384, doi:10.1002/2016TC004248.*

*Pope, R. J. J., Candy, I., and Skourtsos, E.: A chronology of alluvial fan response to Late Quaternary sea level and climate change, Crete, Quatern. Res., 86, 170-183, 2016.*

*New Reference added to the revised version:*

*Mouslopoulou, V., Oncken, O., Hainzl, S., Nicol, A., 2016. Uplift rate transients at subduction margins due to earthquake clustering. Tectonics, 35, 2370–2384, doi:10.1002/2016TC004248.*

I follow here with some specific feedback, which the authors might find useful for sharpening their presentation.

Specific Comments
1) The core contribution of this manuscript is strongly challenged by the short comments from Gallen and Wegmann. Their most significant challenge is to suggest that the Domana fan units are not 20 to 30 ka, as proposed by the authors, but rather <10 ka. The authors need to directly address this issue in the manuscript. It is important to remember that the short comments and reviews will be published along with the paper, so they can be directly addressed and cited in the paper.
*Yes, in this revised version, we have followed Prof. Brandon's advice to directly deal with Gallen and Wegmann's comment and have added a paragraph at the end of section 3.1. We trust that now everything is clear.*

2) Based on my reading, I side with the authors for their interpretation of the age of the Domata fan units *[thank you].* Gallen and Wegmann argue for a Holocene age based on relationships relative to a "Holocene" (?365 AD) sealevel notch. The authors clearly show that the notch is probably not equivalent to the unconformity beneath the lower fan unit. *[Yes, we use Figure 6c to make this point even clearer now in the revised manuscript].* I am also convinced by the IRSL Kspar ages *[thank you]*. These ages are messy, but each of the five ages has a well-defined minimum-age component. The clustering of a minimum-age component in each sample suggests a similar bleaching history for all of these grains *[exactly]*. That suggests to me full bleaching for this component. In addition, the IRSL ages appear to be consistent with the stratigraphic ordering of the samples *[exactly!]*. (It would help to organize the data plots in Figure 7 in stratigraphic order, and to discuss this relationship in the text.) *Done.*

3) Rex Galbraith would blow a gasket if he read this citation to the Galbraith and Roberts (2012) paper (p. 8, line 30), in that citation is used to justify using the mode in the density plot (as determined by the KDE method) as an estimate of the age. Galbraith has spent his career lecturing against the use of probability density plots. In fact, the Galbraith and Roberts review recommends estimating "minimum ages" for mixed OSL grain-age distributions. If the authors want to use the mode of the probability density plot as an estimate for the age, then they should remove the Galbraith and Roberts' citation. Alternatively, they could calculate minimum ages, and, in that case, the Galbraith and Roberts citation would be appropriate. My recommendation is to use the minimum age estimate. That approach provides a clean way to deal with mixed grain ages.

*The reviewer proposes to calculating minimum ages. Indeed, we have now calculated two new age models: the Central Age Model and the Minimum Age Model (see updated Table 1). We removed the KDEmax column. We agree that this value is not distinct as it depends on the band-width used.*

*The 'new' ages are consistent with the 'old' in indicating a last glacial age (MIS3) for all samples and they conform to the geological sequence established.*

*We have added the following text at the upper-half of Page 9:*

*'The Central Age Model (CAM) and the Minimum Age Model (MAM) according Galbraith et al. (1999) were used to further describe the age distributions. Minimum age models are recommended when dating mixed-age sediments yielding broad age distributions to better estimate the population of well bleached grains (Galbraith and Roberts, 2012)'.*
*and...*

*'All Mean values indicate a last glacial age for all samples and appear in stratigraphic order. Median values are all in stratigraphic sequence (except for LF-2a/b) although they are significantly younger than the Mean and the CAM values. The CAM ages are consistent with the Mean and Median values in indicating a last glacial age (MIS3) for all samples. The CAM series data are all in stratigraphic order (except for the LF-2a/b sample), and the absolute values are younger than the means. The MAM ages "lean" towards the younger end of the timescale, which is to be expected, because they are the minimum possible ages, not the most likely ages'.*

*New reference:*
*Galbraith, R. F., Roberts, R. G., Laslett, G. M., Yoshida, H. & Olley, J. M. (1999): Optical dating of single and multiple grains of quartz from Jinmium Rock Shelter, northern Australia: Part I, experimental design and statistical models. In: Archaeometry, 41: 339 – 364.*

4) The manuscript would be stronger if it had a more critically developed interpretation (see section 4). As presently written, the interpretation read as a "just-so" story (i.e. a self-consistent narrative). The reader is left to wonder if there might be other interpretations that fit the data. In addition, I would expect that if the issue of steady versus unsteady uplift is important, then the interpretation should address this issue as well. *Addressed together with the following 'Comment 5'. We have modified the text at several locations to make it sound less narrative.*

5) The manuscript contains many cases where an argument is advanced on the basis of what might be "likely" or "unlikely" to have occurred (see below for partial list). My sense is that these terms are used to indicate what the authors view as reasonable and unreasonable aspects of their interpretations. Unfortunately, this phrasing tends to make the argumentation sound weak. I would recommend stating the interpretation in direct terms and to avoid personal assessments of whether the ideas are likely or unlikely. That judgment is probably best left to the reader.

p. 10, line 20: The initiation of deposition of the upper-fan (Fig. 10a) is likely to have occurred post ~50 kyr BP and prior to 45 kyr *Done.*

p. 10, line 23: We argue that upper-fan deposition is unlikely to have started as early *Done.*

p. 11, line 14: "Marine trimming of the lower-fan surface and deposits is unlikely to have occurred *Done.*

p. 11, line 16: "The Holocene high sea-level stand is the most likely candidate period *Done.*

*We have modified the sentences indicated above, and also the caption of Figure 5, to make our interpretation sound more solid (and less narrative).*

6) The term casual "marine bench" is widely used throughout the paper. The term "marine terrace" or "marine abrasional surface" is probably more precise. Also, it would be useful to know if there is direct evidence that these "flats" were cut in the surf zone, as opposed to being fluvial straths. I suspect that this distinction was made by considering the context of the flat relative to the coast and adjacent river channels. Nonetheless, it would probably help to know if the marine versus fluvial origin of these flats included evidence based on fossils or deposits.

*Done. We have deliberately chosen the non-genetic term "abrasion surface" for this feature as it describes well its morphology without dictating a mode of origin. While there are no marine deposits or fossils associated with the abrasion surface, we conclude from its geometry (and that of the nearby features) that it is marine in origin. Firstly, a cliff incised in the upper fan deposits about 50 m behind the current coastline lies entirely parallel with the present coastal cliff. The older coast-parallel cliff is truncated in the west by an alluvial entrenchment cliff of the Klados River at a high angle (almost orthogonal) to it. The river entrenchment cliff defines the course of the river at a time following deposition of the upper fan and it is barely conceivable under any circumstance that such a steep, unobstructed river course might take such a right angle deviation required to be responsible for erosion of that cliff. Further, as the abrasion surface incised in the upper fan deposits is sub-horizontal across the entire beach-front, there is little or no possibility that this cliff and the abrasion surface could have been eroded by the Klados River. The relationship between the presently active Klados River entrenchment cliff and coastal cliff in the lower fan deposits is entirely analogous. We conclude that the abrasion surface represents a bench in the upper fan deposits that was cut in the surf zone in front of the contemporaneous marine cliff.*

7) The title includes the noun "eustatism" as a modifier. I would suggest using the adjective, "eustatic". For example, "Distinct phases of eustatic and tectonics forcing for late Quaternary landscape evolution in southwest Crete".
*Done.*

**Recommendations for Figures**

Caption for Figure 1: "Numbered arrows show geodetically-derived convergence rates between the African and Eurasian plates and their azimuths at selected sites (after Reilinger et al., 2010)." >>> "Labelled arrows show geodetically-derived site velocities (mm/a) relative to a fixed Nubia plate (Reilinger et al., 2010)."
*Done.*

Figures 3, 5, 6: Change "upper fan surface" and "lower fan surface" to "upper fan tread" and "lower fan tread". Surface is vague. Tread clearly indicates the uppermost limit of an inset geomorphic unit.
*Done.*

Figure 4:
1) Change labels as such: profiles > profile (Fig. 5 shows only one profile for 2) The label "marine cliff profile" should refer to Fig. 5b (not 5a).
*profiles > profile Done!*
*However, reference to Fig. 5a is correct (as each of the two profiles in Figure 4 corresponds to those presented in Figure 5a –and not 5b)!*

Figure 6a:
1) My understanding is that the "upper fan gravels" are inset into the "lower fan gravels". It might help to use a different contact line to indicate an inset relationship. Perhaps something like TTTTTTTTT (like a normal fault contact) with the barbs pointing to the inset unit.
*Done.*

2) In the lower panel, the "marine bench" is shown as grading into the "upper fan gravels". This relationship is inconsistent with the description in the text. The line work needs to be modified (or the text rewritten) to resolve this conflict.
*Done.*

3) The term "marine bench" is a bit confusing. Does it refer to the wave-cut unconformity (marine erosion) or does it refer to a depositional unit below the unconformity? The position of the label suggests that it is depositional unit. Is there evidence that this unit was a marine deposit?
*We have changed the label of the feature from marine bench to "abrasion surface". We also changed the position of the label so that it sits on top of the feature. Please also see our response to your 6th comment, that the feature is erosional and it does not refer to the deposits below (they are remnants of the upper fan deposits, and they are alluvial, not terrestrial).*

Figure 6b: This figure is a bit confusing and incomplete.
1) Unit labeled "rockfall debris post-dating upper fan deposition" lies within the dashed line that encircles the upper and lower fan deposits. I recommend placing the dashed line below this "rockfall debris" unit, to make it clear that it is not part of the two fan units.
*Done.*

2) I recommend showing here the "Older Quaternary deposits" unit in the lower left side of this image (see Figure 6c).
*Done.*
3) The unconformity at the base of the fan units is not highlighted, and the "beach bench" unit is not labeled.
*The unconformity at the base of the fan units is represented here by the onlapping of both fan units against the basement rocks (dashed white line). We have now replaced the term 'marine bench' with 'marine terrace' or 'marine abrasion surface' throughout the manuscript for consistency (as requested by the reviewer earlier).*

Figure 6c: The "Older Quaternary deposits" are not dated, so it is best to call them "older deposits" or "older sediments".
*Done. They are now called 'older deposits'.*

Figure 7
1) What are the numbers on the right axis of these plots? I suspect that they show the cumulative distribution in terms of "Cumulative Grains". This right axis need a label.
*Figure 7 is now replaced according to the new age calculations (see response to Reviewer's comment 3. The right axis is labelled 'cumulative IRSL ages'.*

2) Left axis is labeled "kernel density estimate" but the units are not specified. Of course, density plots do not require any units since they are only used to indicate a relative sense where the location of the probability mass of the dated grains. If scaling is needed, then I recommend, for simplicity, to leave this axis blank. If the authors do insist on labeling this axis, then it should be called "probability density" and the units should be shown (dP/dTau where P is probability and tau is age; could use density units of grains/ka or %/ka).
*Despite the concern of the reviewer, we confirm that the labelling of the left axis as 'kernel density' is correct.*

Table 1.
1) The columns showing locations should be labeled "Longitude" and "Latitude". Easting and northing refer distances, as used for UTM coordinates.
   *Done.*

2) The term "mode" (in the sense of mode of the probability density distribution) should be used instead of "KDE max
   *We removed the KDEmax column completely from Table 1. See our response to Reviewer's 3rd comment above.*

[revised manuscript text omitted]

Figure 1

[Figure]

Figure 2

[Figure]

Figure 3

[Figure]

Figure 4

[Figure]

Figure 5a

[Figure]

Figure 5b

[Figure]

Figure 6a

[Figure]

Figure 6b

[Figure]

5 Figure 6c

**IRSL age distributions**

[Figure]

Figure 7

[Figure]

Figure 8

[Figure]

Figure 9

[Figure]

Figure 10

[Figure]

Figure 11

Table 1

| Sample (depth) | Lab. no. (aliquot no.) | U [ppm] (a) | Th [ppm] (a) | K [%] (a) | Cosmic dose rate [mGy/ka] (b) | Water cont. measured [%] (c) | Water cont. estimated [%] (d) | Dose rate ($D_r$) [Gy/ka] (e) | Equivalent dose ($D_e$) [Gy] / IRSL age [ka] Mean | Median | CAM (f) | MAM (g) | Standard deviation [%] [ka] | Standard error (h) [%] [ka] | Over-dispersion (i) [%] |
|---|---|---|---|---|---|---|---|---|---|---|---|---|---|---|---|
| UF-1 (0.3 m) | HUB-0423 (12) | 0.46 ± 0.02 | 0.63 ± 0.06 | 0.11 ± 0.01 | 195 ± 20 | 1,1 | 3 ± 2 | 0.99 ± 0.07 | 41.7 Gy / **42.1 ka** | 36.0 Gy / **36.4 ka** | 38.5 ± 4.4 Gy / **38.8 ± 5.2 ka** | 27.7 ± 5.0 Gy / **27.9 ± 5.4 ka** | 42.7% / **18.0 ka** | 12,3% / **5.2 ka** | 39.8% |
| UF-2 (1.1 m) | HUB-0424 (7) | 1.06 ± 0.04 | 2.36 ± 0.13 | 0.53 ± 0.01 | 173 ± 17 | 5,1 | 5 ± 2 | 1.67 ± 0.11 | 41.7 Gy / **25.0 ka** | 38.6 Gy / **23.1 ka** | 41.2 ± 2.4 Gy / **24.7 ± 2.1 ka** | - / **-** | 17.5% / **4.4 ka** | 6,6% / **1.7 ka** | 15.0% |
| RB-1 (28.0 m) | HUB-0425 (11) | 0.84 ± 0.02 | 0.62 ± 0.05 | 0.15 ± 0.01 | 17 ± 2 | 1,6 | 3 ± 2 | 0.96 ± 0.06 | 51.3 Gy / **53.4 ka** | 43.4 Gy / **45.2 ka** | 46.3 ± 6.2 Gy / **48.0 ± 7.1 ka** | 34.5 ± 6.2 Gy / **35.8 ± 6.8 ka** | 49.4% / **26.4 ka** | 14,9% / **8.0 ka** | 44.1% |
| LF-1a/b (0.28 m) | HUB-0426 (10) | 0.67 ± 0.02 | 0.4 ± 0.05 | 0.09 ± 0.01 | 194 ± 19 | 0,3 | 3 ± 2 | 1.01 ± 0.07 | 39.9 Gy / **39.5 ka** | 29.1 Gy / **28.8 ka** | 34.7 ± 5.8 Gy / **34.3 ± 6.2 ka** | 29.0 ± 5.2 Gy / **28.6 ± 5.5 ka** | 58.2% / **23.0 ka** | 18,4% / **7.3 ka** | 52.8% |
| LF-2a/b (0.24 m) | HUB-0427 (13) | 0.73 ± 0.02 | 0.67 ± 0.04 | 0.1 ± 0.01 | 195 ± 20 | 1,3 | 3 ± 2 | 1.07 ± 0.07 | 44.0 Gy / **41.2 ka** | 42.8 Gy / **40.0 ka** | 42.6 ± 3.2 Gy / **39.9 ± 4.0 ka** | 39.4 ± 5.4 Gy / **36.9 ± 5.6 ka** | 26.2% / **10.8 ka** | 7,3% / **3.0 ka** | 26.6% |

(a) Uranium, thorium, and potassium contents were determined via high resolution gamma ray spectrometry (HPGe detector).
U-238: U-234 (53.2 keV), Th-234 (63.3 keV), Ra-226 (186.1 keV), Pb-214 (295.2 keV, 351.9 keV), Bi-214 (609.3 keV, 1120.3 keV, 1764.5 keV), Pb-210 (46.5 keV).
Th-232: Ac-228 (338.3 keV, 911.2 keV, 969.0 keV), Pb-212 (238.6 keV), Bi-212 (727.3 keV), Tl-208 (583.2 keV).
K-40: 1461.0 keV.
U-238 and Th-232: The arithmetic means of the activities of the above mentioned natural daughter products were used (± standard error).
The internal K content of the potassium feldspar was set to 12.5 ± 0.5 % (Huntley & Baril 1997).
(b) Cosmic dose rates were estimated regarding geographic position (35°N, 24°E), altitude and sampling depth.
(c) Water content of sediment samples in % of dry mass (oven dried for 24 h at 105 °C).
(d) Water content used for dose rate calculation.
(e) For coarse grain potassium feldspar an a-value of 0.15 ± 0.05 was assumed (Balescu & Lamothe 1994).
(f) Central Age Model (CAM) according Galbraith et al. (1999)
(g) Minimum Age Model (MAM) according Galbraith et al. (1999). Sigma b was set to 0.25.
(h) Standard error of the mean: standard deviation devided by the square root of the number of measured aliquots.
(i) The Overdispersion describes the variation of the equivalent dose in addition to the expected error. It is given by the CAM.